# Simplified Mamba with Disentangled Dependency Encoding for Long-Term Time Series Forecasting

## Abstract

Recent advances in deep learning have led to the development of numerous models for Long-term Time Series Forecasting (LTSF). However, most approaches still struggle to comprehensively capture reliable and informative dependencies inherent in time series data. In this paper, we identify and formally define three critical dependencies essential for improving forecasting accuracy: the order dependency and semantic dependency in the time dimension as well as cross-variate dependency in the variate dimension. Despite their significance, these dependencies are rarely considered holistically in existing models. Moreover, improper handling of these dependencies can introduce harmful noise that significantly impairs forecasting performance. To address these challenges, we explore the potential of Mamba for LTSF, highlighting its three key advantages to capture three dependencies, respectively. We further empirically observe that nonlinear activation functions used in vanilla Mamba are redundant for semantically sparse time series data. Therefore, we propose SAMBA, a Simplified Mamba with disentangled dependency encoding. Specifically, we first eliminate the nonlinearity of vanilla Mamba to make it more suitable for LTSF. Along this line, we propose a disentangled dependency encoding strategy to endow Mamba with efficient cross-variate dependency modeling capability while minimizing the interference between time and variate dimensions. We also provide rigorous theory as a justification for our design. Extensive experiments on nine real-world datasets demonstrate the effectiveness of SAMBA over state-of-the-art forecasting models.

## 1 Introduction

Long-term Time Series Forecasting (LTSF) plays a critical role in various real-world applications, such as early disaster warning and long-term energy scheduling (Zhou et al. (2021); Wu et al. (2021)). Accurate LTSF often relies on extracting reliable dependencies from longer historical time horizons. For example, periodic and trend patterns are frequently leveraged in existing literature to guide model design and interpret results (Wu et al. (2021)). However, the success of simple linear model (Zeng et al. (2023)) and patching strategy in Transformer (Nie et al. (2022)) suggests that only capturing periodic and trend patterns may be insufficient for accurate forecasting. In this paper, we explore the predictability of time series by decomposing dependencies into three key aspects: the **order dependency** and **semantic dependency** in time dimension, as well as the **cross-variate dependency**. Specifically, order dependency represents the sequential nature among time points, semantic dependency captures the high-level temporal variation beyond surface numerical values, and cross-variate dependency refers to the inter-variate relationships in multi-variate time series data. This decomposition offers a new perspective for model design in LTSF.

In fact, most existing LTSF models fail to comprehensively consider both order and semantic dependencies. For example, linear models (e.g., DLinear (Zeng et al. (2023))) rely on point-wise mappings to preserve order dependency but are too simplistic to capture complex semantic dependency. In contrast, Transformer-based models are effective in learning semantic dependency through the self-attention mechanism (Vaswani et al. (2017)) and show improved performance when local temporal semantics is enriched using patching strategies(Nie et al. (2022)). However, they struggle with perceiving temporal order due to the permutation-invariant nature of self-attention, even with

positional encodings (Zeng et al. (2023)). Therefore, capturing both order and semantic dependencies simultaneously remains an underexplored problem in LTSF.

Besides the order and semantic dependencies, cross-variate dependency is also crucial in many real-world scenarios, such as traffic flow, where significant correlations among variates have been observed (Han et al. (2024); Liu et al. (2020)). However, existing approaches that utilize cross-variate dependency (Channel Dependent, CD) frequently underperform compared to methods that treat each variate independently (Channel-Independent, CI) (Nie et al. (2022)). This phenomenon arises because previous CD approaches often encounter issues with over-smoothing and difficulty in fitting individual variate series (Chen et al. (2024)). Although a recent method proposes to perform CI modeling first before CD modeling to alleviate the issues (Liu et al. (2023)), our findings, from both empirical and theoretical, indicate that the less relevant temporal dependencies extracted from other variates can still harm the informativeness of the embedding produced for the target variate sequence. Consequently, effectively leveraging cross-variate dependency while maintaining the unique temporal dynamics and characteristics of each variate continues to be a challenging problem.

Recently, Mamba (Gu & Dao (2023)) has emerged as a strong contender to the transformer architecture in the field of natural language processing (NLP) (Park et al. (2024)). Given the similar sequential structure of time series and text data, Mamba also demonstrates great potential in solving LTSF task. Through extensive experiments, we identify that Mamba presents three major advantages over Transformer for LTSF: (i) Mamba uses State Space Models (SSMs) to process sequences recursively, inherently capturing order dependency(Gu et al. (2021b)). (ii) Its selection mechanism, similar to self-attention, enables it to focus on or ignore particular inputs, making it well-suited for learning semantic dependency. (iii) Mamba's near-linear complexity provides an efficiency advantage in encoding cross-variate dependency, which are typically computationally expensive when dealing with large amounts of variates.

Despite these advantages, directly adapting Mamba for LTSF is still non-trivial due to two key challenges: (i) Mamba was initially designed for NLP tasks, where semantic dependency is of primary importance. In contrast, LTSF typically emphasizes order dependency due to the sparsity of information in time series as opposed to natural language (Zeng et al. (2023)). (ii) Although Mamba can effectively capture both order and semantic dependencies, it lacks mechanisms for modeling cross-variate dependency, which is crucial for multivariate forecasting. Simply incorporating cross-variate dependency into Mamba using a CD strategy may exacerbate the issue by introducing irrelevant correlations, undermining the model's effectiveness to handle individual variate series properly.

To overcome these challenges, we propose SAMBA, a Simplified Mamba with disentangled dependency encoding specifically tailored for LTSF. First, we remove the nonlinearities in vanilla Mamba, which we find to be redundant and prone to overfitting in the context of semantically sparse time series data, thereby improving its generalization ability. Furthermore, we introduce a theoretically sound disentangled encoding strategy that explicitly separates cross-time and cross-variate dependencies to avoid mutual interference between the two dimensions. We conduct extensive experiments on real-world forecasting benchmarks, demonstrating the superiority of SAMBA over the state-of-the-art methods. In particular, SAMBA proves universally effective across datasets with varying degrees of cross-variate dependency. Our contributions are summarized as follows.

- We identify and formally define three critical dependencies in time series data to guide the design of the LTSF models.

- Our comprehensive analysis reveals two insights: (i) Compared to Linear model and Transformers, Mamba can effectively capture both order and semantic dependencies. (ii) Directly applying MLP, Transformer, and Mamba to LTSF can lead to overfitting issues due to the presence of nonlinearities. To mitigate this, we propose a simplified Mamba by removing nonlinearities and incorporating a theoretically sound disentangled encoding strategy that appropriately integrates cross-variate dependency to the model, enhancing the model's global representation and predictive capabilities.

- We empirically demonstrate that SAMBA achieves state-of-the-art performance across mainstream benchmarks. Our analysis also highlights the general utility of disentangled encoding, offering a universal strategy for developing future LTSF models.

## 2 RELATED WORKS

Deep learning has made significant advances in the field of natural language processing (Devlin et al. (2018)) and speech recognition (Dong et al. (2018)), inspiring researchers to repurpose these models for time series forecasting (Zhou et al. (2021); Wu et al. (2021)). The mainstream methods for LTSF currently include linear models and Transformer-based models. In particular, because of the strong modeling capability of the transformer, numerous works have utilized transformers to achieve superior performance. However, applying Transformers to long sequence forecasting presents two major challenges: 1) computational overhead due to quadratic space-time complexity and 2) the difficulty of capturing long-term dependencies. Early approaches concentrated on modifying the architectural design to reduce complexity. For example, LogTrans (Li et al. (2019)) introduced convolutional self-attention to capture local information while utilizing LogSparse techniques to reduce complexity. Informer (Zhou et al. (2021)) leveraged a self-attention distillation mechanism to extend the input length and proposed ProbSparse self-attention to optimize computation efficiency. Furthermore, Autoformer (Wu et al. (2021)) selected a decomposed structural framework to disintegrate time and replaced traditional self-attention mechanisms with autocorrelation mechanisms to reduce complexity.

Recently, linear models have demonstrated superior performance with fewer parameters and higher efficiency, prompting researchers to question whether transformers are suitable for long sequence forecasting (Zeng et al. (2023)). Researchers have begun exploring how to let transformers exhibit their powerful performance in scenarios like natural language processing and speech based on the inherent properties of time series. For example, considering the distribution shift problem inherent in time series, methods like Non-stationary transformers (Liu et al. (2022b)) have applied techniques to stabilize and reduce the inherent distribution shift of time series. Given the insufficiency of semantic information at individual time points, the PatchTST (Nie et al. (2022)) model, by patching to extract local information, addresses the issue of insufficient short-time series semantic content, thus enriching the semantic content of each token.

Recent research shows that utilizing the CI strategy to achieve promising results (Nie et al. (2022)), inspiring researchers to explore a new encoding method that can consider both intra-variate and inter-variate interactions (Chen et al. (2023)). Crossformer (Zhang & Yan (2022)) adopted a cross-variate encoding method and a two-stage attention mechanism that considers both time and variate relationships. iTransformer (Liu et al. (2023)) applied a data inversion method, considering the time series corresponding to each variate as a token and then capturing the relationships between variates using self-attention mechanisms. CARD (Wang et al. (2024)) explicitly models both cross-time dependency and cross-variate dependency, where the latter directly mixes these dependencies at each model layer. As a result, CARD still suffers from the issue mentioned in the introduction, which is the inappropriate mixing of cross-time and cross-variate dependencies.

Unlike previous work, we explore a new architecture more suitable for LTSF—Mamba (Gu & Dao (2023)), with modifications to better fit the LTSF scenarios. Moreover, different from previous approaches to serially encoding time and variate relationships, we propose a parallel encoding method that decouples cross-time and cross-variate dependencies. This disentanglement ensures that the encoding process of each dependency does not adversely affect each other, leading to state-of-the-art (SOTA) performance on datasets with varying degrees of variate relationships.

## 3 PROBLEM FORMULATION

The LTSF aims to learn a mapping function $\mathcal{F}(\cdot)$ that forecasts the temporal evolution of $N$ variates in the future $S$ time steps based on the observations in the past $T$ time steps, *i.e.*,

$$\mathcal{F} : \mathbb{R}^{N \times T} \to \mathbb{R}^{N \times S}, \ \mathbf{X} = (\mathbf{x}_1, \dots, \mathbf{x}_T) \mapsto \mathbf{Y} = (\mathbf{x}_{T+1}, \dots, \mathbf{x}_{T+S}), \tag{1}$$

where $\mathbf{x}_t = (c_t^1, \dots, c_t^N) \in \mathbb{R}^N$ denotes the states of $N$ variates at time step $t$.

Accurate LTSF depends on effectively capturing the following three types of dependencies:

(1) *Order dependency:* It refers to the temporal ordering relationships among sequentially observed data points of a variate. Formally, given observations $c_t$ and $c_s$ where $t < s$, we omit the superscript $i$ of $c_t$ for brevity. The order dependency is significant for prediction if

$$I(c_t; c_s \mid t, s) > I(c_t; c_s), \tag{2}$$

where $I(\cdot; \cdot)$ $(I(\cdot; \cdot \mid \cdot))$ denotes the (conditional) mutual information (Cover (1999)). For example, the increasing trend of daily temperatures in the past week helps predict that the trend will likely continue in the future.

(2) *Semantic dependency:* It refers to the latent semantic relationships between historical and future data points of a variate. It goes beyond the superficial temporal ordering information, which is more stable across temporal contexts, and requires more expressive models to extract. Formally, the semantic dependency is significant for prediction if there exists a permutation-invariant (nonlinear) function $\mathcal{S}(\cdot)$ that maps inputs to semantic space, *s.t.*,

$$H(\boldsymbol{c}_{T+1:T+S}|\mathcal{O}(\boldsymbol{c}_{1:T}), \mathcal{S}(\boldsymbol{c}_{1:T})) < H(\boldsymbol{c}_{T+1:T+S}|\mathcal{O}(\boldsymbol{c}_{1:T})). \tag{3}$$

where $H(\cdot|X)$ denotes the conditional entropy of a random variate given variate $X$ and $\mathcal{O}(\cdot)$ includes additional order dependencies *w.r.t.* numerical inputs. For instance, periodic patterns facilitate more precise forecasting of seasonal temperature patterns on top of the potential temporal trends.

(3) *Cross-variate dependency:* It refers to the complex relationships between variate $i$ and $j$. The cross-variate dependency is significant for predicting variate $i$ if

$$I(\boldsymbol{c}_{1:T}^i; \boldsymbol{c}_{T+1:T+S}^i \mid \boldsymbol{c}_{1:T}^j) > I(\boldsymbol{c}_{1:T}^i; \boldsymbol{c}_{T+1:T+S}^i). \tag{4}$$

For instance, exploiting co-evolving temperature patterns between the target region and adjacent regions could improve the prediction in the target region.

## 4 EMPIRICAL EXPLORATION OF MAMBA FOR LTSF

An effective LTSF model should be able to take advantage of the three types of dependencies defined in Section 3. We first conduct in-depth ablation experiments to evaluate three models: two classical LTSF models, Linear model and Transformer, and the third, Mamba, in capturing these dependencies. The details of the implementation of the following experiments are provided in Appendix B.4.

### 4.1 WHY MAMBA? A DEEP DIVE INTO ITS SUITABILITY FOR LTSF

To validate Mamba's suitablity for LTSF, we evaluate its capability to capture both order dependency and semantic dependency. The evaluation of Mamba's performance in capturing cross-variate dependency is provided in Appendix C. To assess the models' ability to capture order dependency, we first make the following assumption.

**Assumption 1.** The performance of an effective order dependency learner is significantly influenced by the order of the input sequence (Zeng et al. (2023)).

According to Assumption 1, a greater performance variation indicates a greater reliance on order dependency. Our results in Table 1 show that both the Linear model and Mamba surpass the Transformer in terms of capturing order dependency. This is because the Linear model and Mamba that use the SSM-based approach are permutation-variant, i.e., regarding the order. Their sensitivity to sequence order is stronger than that of the Transformer, which relies on the permutation-invariant self-attention mechanism. This increased sensitivity accounts for the greater performance decrease observed for the linear model and Mamba, as illustrated in Table 1.

Table 1: Order dependency analysis on ETTm1 dataset (Zhou et al. (2021)). Based on Assumption 1, we compare each model's performance variation before and after randomly shuffling the temporal order of testing data points. O.MSE and O.MAE are evaluated in the original test set. S.MSE and S.MAE are evaluated in the shuffling test set.

| Models | Linear Model | | | | Mamba | | | | Transformer | | | |
|---|---|---|---|---|---|---|---|---|---|---|---|---|
| Metrics | O.MSE | S.MSE | O.MAE | S.MAE | O.MSE | S.MSE | O.MAE | S.MAE | O.MSE | S.MSE | O.MAE | S.MAE |
| 96 | 0.383 | 0.988 | 0.400 | 0.697 | 0.517 | 0.922 | 0.508 | 0.688 | 0.643 | 0.884 | 0.575 | 0.643 |
| 192 | 0.413 | 0.986 | 0.415 | 0.697 | 0.575 | 0.931 | 0.546 | 0.699 | 0.805 | 1.01 | 0.664 | 0.730 |
| 336 | 0.441 | 0.987 | 0.435 | 0.698 | 0.730 | 0.957 | 0.634 | 0.703 | 0.882 | 1.12 | 0.737 | 0.817 |
| 720 | 0.497 | 0.992 | 0.469 | 0.704 | 0.873 | 0.973 | 0.704 | 0.723 | 0.928 | 1.12 | 0.752 | 0.80 |
| Avg. Drop | - | 127.97% | - | 62.55% | - | 40.37% | - | 17.60% | - | 22.40% | - | 6.55% |

To assess the model's ability to capture semantic dependency, we introduce the second assumption.

Table 2: Results of the linear model, Mamba, and Transformer w/ or w/o patching on ETTm1 dataset.

| Models | Linear Model | | | | Mamba | | | | Transformer | | | |
|---|---|---|---|---|---|---|---|---|---|---|---|---|
| | w/o patching | | w/ patching | | w/o patching | | w/ patching | | w/o patching | | w/ patching | |
| Metrics | MSE | MAE | MSE | MAE | MSE | MAE | MSE | MAE | MSE | MAE | MSE | MAE |
| 96 | 0.383 | 0.400 | 0.366 | 0.388 | 0.517 | 0.508 | 0.341 | 0.377 | 0.643 | 0.575 | 0.364 | 0.394 |
| 192 | 0.413 | 0.415 | 0.400 | 0.404 | 0.575 | 0.546 | 0.378 | 0.399 | 0.805 | 0.664 | 0.394 | 0.404 |
| 336 | 0.441 | 0.435 | 0.429 | 0.425 | 0.730 | 0.634 | 0.413 | 0.421 | 0.882 | 0.737 | 0.429 | 0.430 |
| 720 | 0.497 | 0.469 | 0.485 | 0.460 | 0.873 | 0.704 | 0.474 | 0.465 | 0.928 | 0.752 | 0.468 | 0.4600 |
| Avg. | 0.434 | 0.469 | 0.420 | 0.419 | 0.674 | 0.598 | 0.402 | 0.416 | 0.815 | 0.682 | 0.414 | 0.422 |

**Assumption 2.** Patching enhances the semantic dependency of a sequence (Nie et al. (2022)).

For example, a single temperature data point is insufficient to illustrate a time pattern, but a continuous set of temperature data over a morning period can reveal valuable insights into the day's climate conditions. Based on Assumption 2, we compare each model's performance variation before and after applying the patching strategy to the input sequence. A more pronounced performance shift post-patching suggests a stronger reliance on semantic dependency.

The experimental results in Table 2 show a significant improvement in the Transformer after patching compared to the linear model after patching, demonstrating its stronger semantic learning capability. The Linear model, due to its limited expressive power, struggles to effectively capture semantic dependency, further validating the rationale behind our experimental approach. In addition, compared to the Linear model, Mamba with the patching strategy also exhibits notable improvements and achieves the best performance, which verifies its strength in capturing semantic dependency.

In summary, our evaluation results above indicate that Mamba is the only model capable of simultaneously capturing order and semantic dependencies, making it particularly well-suited for LTSF, where both order and semantic dependencies are critical.

## 4.2 THE PITFALLS OF NONLINEARITY: OVERFITTING IN DEEP LTSF MODELS

The success of Linear models has prompted researchers to reconsider the utility of more complex architectures, such as Transformers (Zeng et al. (2023)). Initially designed for NLP tasks, these models emphasize semantic dependency over order dependency and incorporate multiple nonlinear activation functions. Consequently, we hypothesize that these nonlinear activation functions may increase the risk of overfitting in time series data, which possess lower information density compared to natural language.

To test this hypothesis, we carry out ablation studies using nonlinear activation functions across multiple backbone models: MLP, Transformer, and Mamba, along with their respective variants that incorporate a patching strategy. As shown in Table 3, including nonlinear activation functions negatively impacts the model's performance. Interestingly, removing these functions results in notable performance improvements. Figure 1 further illustrates this trend, showing that MLP, Mamba, and Transformer exhibit varying levels of overfitting, with the nonlinear versions overfitting more than their linear counterparts. This observation implies that while nonlinear activation functions enhance a model's capacity for semantic dependency learning, they may simultaneously impair its ability to generalize from temporal patterns in time series data.

Our analysis further reveals that the adverse effects of nonlinear activation functions are most pronounced in Transformer architectures, with Mamba and MLP following. Notably, this order is inversely related to the model's ability to capture order dependency. This suggests that a model's

Table 3: Ablation study on nonlinear activation function on ETTm1 dataset, we report the average performance. 'Original' means vanilla model, '-n' means removing the nonlinear activation function and 'Patch+' means using patching.

| Models | MLP | | Mamba | | Patch+Mamba | | Transformer | | Patch+Transformer | |
|---|---|---|---|---|---|---|---|---|---|---|
| Metric | MSE | MAE | MSE | MAE | MSE | MAE | MSE | MAE | MSE | MAE |
| Original | 0.415 | 0.421 | 0.674 | 0.598 | 0.402 | 0.412 | 0.815 | 0.682 | 0.414 | 0.422 |
| Original-n | 0.406 | 0.411 | 0.635 | 0.585 | 0.399 | 0.414 | 0.653 | 0.600 | 0.406 | 0.417 |
| **Improvement** | 2.17% | 2.37% | 5.79% | 2.17% | 0.75% | -0.48% | 19.88% | 12.02% | 1.93% | 1.18% |

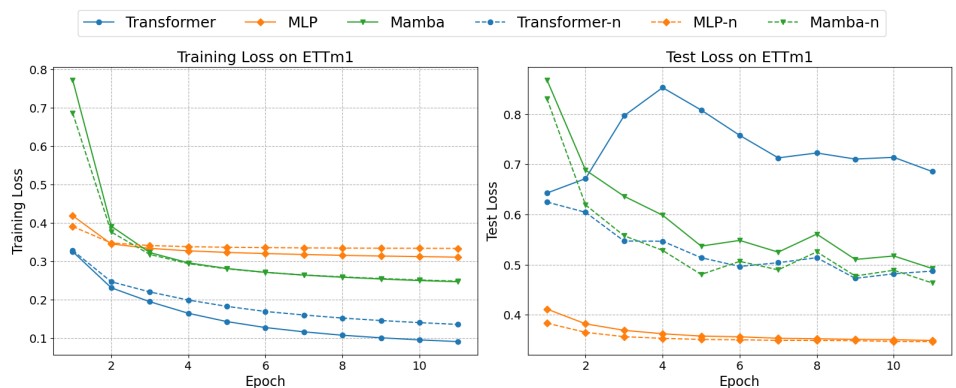

Figure 1: Training curves of three models and their variants with nonlinear activation removed.

ability to capture order dependency can partially mitigate the overfitting introduced by nonlinear activation functions, thereby enhancing its capacity to generalize temporal patterns.

Furthermore, after applying the patching strategy, we observe that removing nonlinear activation functions results in fewer performance variability in Mamba and Transformer models. The patching strategy increases the semantic density of the time series data, allowing nonlinear models to better utilize their capacity without overfitting to sparse information. However, despite the ability of patching to mitigate overfitting to some extent, removing nonlinear activation functions remains overall beneficial, leading to more stable and generalizable models.

## 5 SAMBA

As aforementioned, adapting Mamba to LTSF presents two non-trivial challenges: over-reliance on semantic dependency and the difficulty in leveraging cross-variate dependency without compromising the individual characteristics of each variate. To address these challenges, we analyze the training dynamic of Mamba in Section 4.2 and identify an overfitting issue caused by nonlinear activation function. Inspired by this, we proposed SAMBA, illustrated in Figure 2, with a disentangled dependency encoding strategy to avoid mutual interference between time and variate dimensions.

### 5.1 SIMPLIFYING MAMBA FOR LTSF

Although attention mechanism excel in learning semantic dependency, its inherent permutation invariance constrains its ability to accurately capture order dependency, despite the use of opsitional encoding (Zeng et al. (2023)). Compared to Transformer, Mamba's strength in LTSF lies in its use of selective SSM that is similar to self-attention but processes time series recursively. This approach enables Mamba to capture order dependency and semantic dependency simultaneously, as

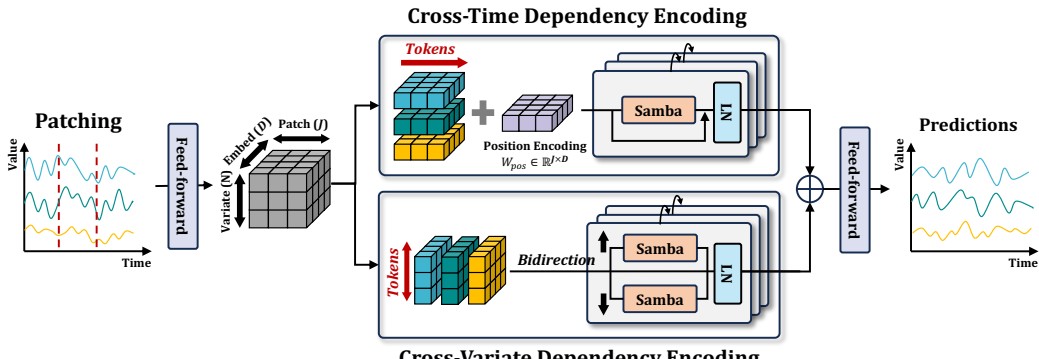

Figure 2: The overall framework of SAMBA.

demonstrated in Section 4.1. Formally, selective SSM can be described as:

$$h_t = \overline{\mathbf{A}}h_{t-1} + \overline{\mathbf{B}}\boldsymbol{x}_t, \ \boldsymbol{y}_t = \mathbf{C}\boldsymbol{h}_t.$$

where $\overline{\mathbf{A}} \in \mathbb{R}^{N \times N}$, and $\overline{\mathbf{B}}, \mathbf{C} \in \mathbb{R}^{N \times D}$ are learnable parameters that map input sequence $\boldsymbol{x}_t \in \mathbb{R}^D$ to output sequence $\boldsymbol{y}_t \in \mathbb{R}^D$ through an hidden state $\boldsymbol{h}_t \in \mathbb{R}^N$. In particular, $\overline{\mathbf{A}}$ and $\overline{\mathbf{B}}$ are the discretized forms of $\mathbf{A}$ and $\mathbf{B}$ using $\boldsymbol{\Delta}$ for seamless intergration deep learning. The discretization functions for the input time series are defined as:

$$\overline{\mathbf{A}} = \exp\left(\boldsymbol{\Delta}\mathbf{A}\right), \ \overline{\mathbf{B}} = (\boldsymbol{\Delta}\mathbf{A})^{-1}(\exp\left(\boldsymbol{\Delta}\mathbf{A}\right) - \mathbf{I}) \cdot \boldsymbol{\Delta}\mathbf{B}.$$

Based on discrete SSM that has been theoretically proven to capture long-range order dependency (Gu et al. (2021a)), Mamba introduces a selection mechanism that makes $\mathbf{B}, \mathbf{C}$ and $\boldsymbol{\Delta}$ data dependent, allowing it to capture semantic dependency as well:

$$\mathbf{B} = \text{Linear}_N(\boldsymbol{x}_t), \quad \mathbf{C} = \text{Linear}_N(\boldsymbol{x}_t), \quad \boldsymbol{\Delta} = \text{softplus}(\text{Linear}_N(\boldsymbol{x}_t)).$$

In addition to the slective SSM, each Mamba layer consists of two branches. The left branch show in Figure 3 is expressed as:

$$\boldsymbol{x}_t' = \text{SelectiveSSM}(\sigma(\text{Conv1D}(\text{Linear}(\boldsymbol{x}_t)))).$$

This branch is designed to efficiently capture both order and semantic dependencies. Based on our analysis in Section 4.2, we propose a SAMBA block that removes the nonlinear activation function between the Conv1D and SSM layers to mitigate the overfitting issue. Our rationale is that 1D convolution, similar to a linear layer, is sufficient to learn a powerful time series representation.

The remaining part is the implementation of a gating mechanism and residual operation, which can be written as:

$$\boldsymbol{y} = \text{LayerNorm}(\boldsymbol{x}_t' \otimes (\sigma(\text{Linear}(\boldsymbol{x}_t))) + \boldsymbol{x}_t).$$

Notice that we retain the nonlinear activation within the gating mechanism (right in Figure 3) to ensure the stability and robustness of the learning process ( Gu & Dao (2023); Chung et al. (2014)).

## 5.2 DISENTANGLED DEPENDENCY ENCODING

Previous CD approaches encode cross-time dependency (order and semantic dependencies) and cross-variate dependency sequentially. However, this can introduce less relevant cross-time dependency from other variates, which negatively impact the informativeness of the embedding for the target variate. To address this, we propose a theoretically sound disentangled encoding strategy that processes cross-time and cross-variate dependencies in parallel. By aggregating information from both dimensions and mapping them into a unified space, our strategy ensures that these dependencies are fully leveraged while minimizing the negative interference between them.

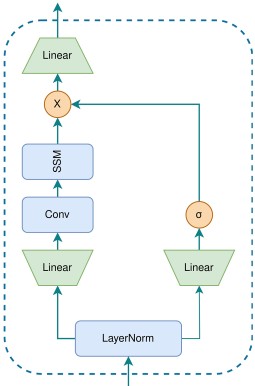

Figure 3: The framework of a SAMBA block.

**Dependency-specific Embedding.** The intricate interactions within multivariate time series make disentangling cross-time and cross-variate dependencies a complex challenge. Analyzing the inputs of Figure 2, we observe that different variates exhibit various periodicities and value ranges but show evident regularity in local changes. This insight motivated us to consider dependencies from the sub-series level. Consequently, we divide the $i$-th variate time series $\mathbf{c}^i$ into several patches $\mathbf{C}_p^i \in \mathbb{R}^{J \times P}$, where $J$ is the number of patches and $P$ refers to the length of each patch. Subsequently, we use a linear projection $\mathbf{W}_p \in \mathbb{R}^{P \times D}$ to map each patch into the $D$ dimensional embedding space as $\mathbf{E}^i \in \mathbb{R}^{J \times D}$. Thereafter, we concatenate the embeddings of the patch into a tensor $\mathbf{E} \in \mathbb{R}^{N \times J \times D}$, where $N$ denotes the number of variates. To separately capture the cross-time and cross-variate dependencies, we disentangle the tensor along the temporal dimension as $\mathbf{E}_{time}^i \in \mathbb{R}^{J \times D}$ and along the variate dimension as $\mathbf{E}_{var}^i \in \mathbb{R}^{N \times D}$.

**Cross-time Dependency Modeling.** To accurately capture cross-time dependency, we independently model each variate. First, we introduce the learnable additive position encoding $\mathbf{W}_{\text{pos}} \in \mathbb{R}^{J \times D}$ to

enhance the order information of the patches, modifying the embedding as $\mathbf{E}_{time}^i \leftarrow \mathbf{E}_{time}^i + \mathbf{W}_{\text{pos}}$, where $\mathbf{E}_{time}^i \in \mathbb{R}^{J \times D}$. Subsequently, we process each channel independently through a single SAMBA block, equipped with residual connections and Layer Normalization (LN) (Ba et al. (2016)): $\mathbf{E}_{time}^i = \text{LN}(\text{SAMBA}(\mathbf{E}_{time}^i) + \mathbf{E}_{time}^i)$. This process effectively extracts the order dependency and semantic dependency of each channel, producing a patch representation $\mathbf{E}_{time} \in \mathbb{R}^{N \times J \times D}$ enriched with temporal information.

**Cross-variate Dependency Modeling.** Existing methods that introduce cross-variate dependency often conflate temporal and variate information, reducing the distinguishability of each channel (Liu et al. (2023)). This can negatively impact model performance. To overcome this limitation, we design an adjustable bidirectional SAMBA module to capture complex cross-variate dependency in a disentangled way without blurring the distinction between the time and variate dimensions. The input sequence is $\mathbf{E}_{var}^i \in \mathbb{R}^{N \times D}$. Each input sequence through the module will generate a forward result $\mathbf{E}_{fo}^i$ and a backward result $\mathbf{E}_{ba}^i$, maintaining the disorder of time series in variate dimensions. Thereafter, we employ a flexible aggregation operation to integrate the forward and backward information from the variate dimensions of time series sequences and use layer normalization to increase the convergence: $\mathbf{E}_{var}^i = \text{LN}(\alpha \mathbf{E}_{fo}^i + \beta \mathbf{E}_{ba}^i + \mathbf{E}_{var}^i)$ where $\alpha$ and $\beta$ are hyperparameters. This strategy produces patch representations $\mathbf{E}_{var} \in \mathbb{R}^{J \times N \times D}$ that encompass complex cross-variate dependency.

**Unified Representation for Forecasting.** After disentangling and separately capturing cross-time and cross-variate dependencies, we concatenate the corresponding representations and utilize a Feed-Forward Network (FFN) to aggregate and map them into a unified space: $\mathbf{E}_o = \text{FFN}(\mathbf{E}_{time} || \mathbf{E}_{var}) \in \mathbb{R}^{N \times J \times D}$. In this way, we obtain patch representations that capture three key dependencies without introducing harmful interference. Finally, we use a flatten layer with a linear head to map these patch representations $\mathbb{E}_o$ to the final prediction result $\mathbf{Z} \in \mathbb{R}^{N \times S}$.

**Theoretical Discussions.** Finally, we theoretically analyze the sufficient expressiveness of our disentanglement strategy compared to LSTF models that alternatively encode cross-time and cross-variate dependencies. To begin with, let $\Phi : \mathbb{R}^{J \times D} \to \mathbb{R}^{J \times D}$ and $\Psi : \mathbb{R}^{N \times D} \to \mathbb{R}^{N \times D}$ denote a cross-time encoder and a cross-variate encoder, respectively. With a little abuse of notation, a broadcasting mechanism will be applied when encoding 3D tensor inputs. Then our disentangled model can be denoted as $\mathcal{F}_d(\cdot) = \text{FFN}(\Phi(\cdot) || \Psi(\cdot))$, which encodes the two types of dependencies in parallel. Based on Theorem 3.5 in Gao & Ribeiro (2022) and the evidence from Liu et al. (2023), we focus on comparing our model with *time-then-variate* LSTF models, which can be defined as $\mathcal{F}_{ttv}(\cdot) = \text{FFN}(\Psi(\Phi(\cdot)))$. Without loss of generality, we only consider the single-layer setting. We further assume that the cross-time encoder $\Phi(\cdot)$ can be decomposed into $\Phi(\cdot) = \phi(\mathcal{O}(\cdot), \mathcal{S}(\cdot)) + \mathcal{Z}(\cdot)$, where $\mathcal{O}(\cdot)$ and $\mathcal{S}(\cdot)$ extract order and semantic dependency respectively, $\mathcal{Z}(\cdot)$ is the noisy component, and $\phi(\cdot)$ is a mapping. Let $\mathbf{c}^i := \mathbf{c}_{T+1:T+S}^i$ for simplicity. Our main theorem is as follows.

**Theorem 1.** *For any variate $i$, $1 \leq i \leq N$, if (1) $I(\mathbf{c}^i; \mathcal{Z}(\mathbf{E})^j | \mathcal{O}(\mathbf{E})^j, \mathcal{S}(\mathbf{E})^j) = 0, \forall j, 1 \leq j \leq N$ and (2) the dependencies of different variates satisfy $I(\mathbf{c}^i; \mathcal{S}(\mathbf{E})^{-i} | \Phi(\mathbf{E})^i, \mathcal{O}(\mathbf{E})^{-i}) \leq \epsilon$ for some $\epsilon \geq 0$, where $\mathcal{O}(\mathbf{E})^{-i}(\mathcal{S}(\mathbf{E})^{-i})$ denotes the order(semantic) dependencies of variates except from the $i$-th one, then the informativeness of the representations output by the disentangled model and the time-then-variate model satisfy $H(\mathbf{c}^i | \mathcal{F}_d(\mathbf{E})^i) \leq H(\mathbf{c}^i | \mathcal{F}_{ttv}(\mathbf{E})^i) + \epsilon$.*

Please refer to Appendix A for detailed proof. Intuitively, given the assumption that cross-time encoding cannot extract more informative semantic dependencies from the embeddings of other variates for forecasting the future of the target variate, our disentanglement strategy can extract sufficient dependencies while reducing interference between cross-time and cross-variate encoding processes. More discussions on the rationality of the assumption can be found in Appendix A.2

# 6 EXPERIMENTS

We comprehensively evaluate the performance and efficiency of the proposed SAMBA in LTSF and analyze the effectiveness of each component. Meanwhile, we also extend our disentangled strategy to other models, validating the universality of the proposed framework. We briefly list the datasets and baselines below. More details are provided in Appendix B.

**Datasets.** We conduct experiments on nine real-world datasets following Liu et al. (2023): (1) ECL, (2) ETTh1, (3) ETTh2, (4) ETTm1, (5) ETTm2, (6) Exchange, (7) Traffic, (8) Weather, (9) Solar-Energy.

**Baselines.** We carefully use 13 popular LTSF forecasting models as our baselines and we cite their performance from Liu et al. (2023) if applicable. Our baselines include (1) *Pretrained Language model-based model:* FTP (Zhou et al. (2023)); (2) *Transformer-based models:* Autoformer (Wu et al. (2021)), FEDformer (Zhou et al. (2022)), Stationary (Liu et al. (2022b)), Crossformer (Zhang & Yan (2022)), PatchTST (Nie et al. (2022)), iTransformer (Liu et al. (2023)), CARD (Wang et al. (2024)); (3) *Linear-based models:* DLinear (Zeng et al. (2023)), TiDE (Das et al. (2023)), RLinear (Li et al. (2023)); and (4) *TCN-based models:* SCINet (Liu et al. (2022a)), TimesNet (Wu et al. (2022)).

## 6.1 LONG-TERM TIME SERIES FORECASTING

Table 4 presents the multivariate long-term forecasting results. In general, SAMBA outperforms all baseline methods. Specifically, for the ETT dataset, which exhibits weak cross-variate dependency, models using the CD strategy, such as iTransformer and Crossformer, underperform the models that use the CI strategy (e.g. PatchTST). However, SAMBA, which introduces cross-variate dependency through a disentangled encoding strategy, demonstrates superior performance over all models that use the CI strategy. For datasets with significant cross-variate dependency, such as Weather, ECL, and Traffic, SAMBA also performs comparable to or superior to the SOTA iTransformer.

Table 4: Multivariate forecasting results with prediction lengths $S \in \{96, 192, 336, 720\}$ and fixed lookback length $T = 96$. Results are averaged from all prediction lengths. Full results are listed in Appendix N.2.

| Models | SAMBA (Ours) | | CARD (2024) | | FTP (2023) | | iTransformer (2023) | | RLinear (2023) | | PatchTST (2022) | | Crossformer (2022) | | TiDE (2023) | | TimesNet (2022) | | DLinear (2023) | | SCINet (2022a) | | FEDformer (2022) | | Stationary (2022b) | | Autoformer (2021) | |
|---|---|---|---|---|---|---|---|---|---|---|---|---|---|---|---|---|---|---|---|---|---|---|---|---|---|---|---|---|
| Metric | MSE | MAE | MSE | MAE | MSE | MAE | MSE | MAE | MSE | MAE | MSE | MAE | MSE | MAE | MSE | MAE | MSE | MAE | MSE | MAE | MSE | MAE | MSE | MAE | MSE | MAE | MSE | MAE |
| ECL | **0.172** | **0.268** | 0.204 | 0.291 | 0.210 | 0.291 | 0.178 | 0.270 | 0.219 | 0.298 | 0.205 | 0.290 | 0.244 | 0.334 | 0.251 | 0.344 | 0.192 | 0.295 | 0.212 | 0.300 | 0.268 | 0.365 | 0.214 | 0.327 | 0.193 | 0.296 | 0.227 | 0.338 |
| ETTh1 | 0.443 | **0.432** | 0.500 | 0.474 | 0.450 | 0.439 | 0.454 | 0.447 | 0.446 | 0.434 | 0.469 | 0.454 | 0.529 | 0.522 | 0.541 | 0.507 | 0.458 | 0.450 | 0.456 | 0.452 | 0.747 | 0.647 | **0.440** | 0.460 | 0.570 | 0.537 | 0.496 | 0.487 |
| ETTh2 | **0.363** | **0.392** | 0.398 | 0.416 | 0.385 | 0.411 | 0.383 | 0.407 | 0.374 | 0.398 | 0.387 | 0.407 | 0.942 | 0.684 | 0.611 | 0.550 | 0.414 | 0.427 | 0.559 | 0.515 | 0.954 | 0.723 | 0.437 | 0.449 | 0.526 | 0.516 | 0.450 | 0.459 |
| ETTm1 | **0.378** | **0.394** | 0.409 | 0.407 | 0.392 | 0.401 | 0.407 | 0.410 | 0.414 | 0.407 | 0.387 | 0.400 | 0.513 | 0.496 | 0.419 | 0.419 | 0.400 | 0.406 | 0.403 | 0.407 | 0.485 | 0.481 | 0.448 | 0.452 | 0.481 | 0.456 | 0.588 | 0.517 |
| ETTm2 | **0.276** | **0.322** | 0.288 | 0.332 | 0.285 | 0.331 | 0.288 | 0.332 | 0.286 | 0.327 | 0.281 | 0.326 | 0.757 | 0.610 | 0.358 | 0.404 | 0.291 | 0.333 | 0.350 | 0.401 | 0.571 | 0.537 | 0.305 | 0.349 | 0.306 | 0.347 | 0.327 | 0.371 |
| Exchange | 0.356 | **0.401** | 0.395 | 0.421 | 0.368 | 0.406 | 0.360 | 0.403 | 0.378 | 0.417 | 0.367 | 0.404 | 0.940 | 0.707 | 0.370 | 0.413 | 0.416 | 0.443 | **0.354** | 0.414 | 0.750 | 0.626 | 0.519 | 0.429 | 0.461 | 0.454 | 0.613 | 0.539 |
| Traffic | **0.422** | **0.276** | 0.481 | 0.321 | 0.511 | 0.334 | 0.428 | 0.282 | 0.626 | 0.378 | 0.481 | 0.304 | 0.550 | 0.304 | 0.760 | 0.473 | 0.620 | 0.336 | 0.625 | 0.383 | 0.804 | 0.509 | 0.610 | 0.376 | 0.624 | 0.340 | 0.628 | 0.379 |
| Weather | **0.249** | 0.278 | **0.249** | **0.276** | 0.267 | 0.287 | 0.258 | 0.278 | 0.272 | 0.291 | 0.259 | 0.281 | 0.259 | 0.315 | 0.271 | 0.320 | 0.259 | 0.287 | 0.265 | 0.317 | 0.292 | 0.363 | 0.309 | 0.360 | 0.288 | 0.314 | 0.338 | 0.382 |
| Solar-Energy | **0.229** | **0.253** | 0.245 | 0.277 | 0.269 | 0.304 | 0.233 | 0.262 | 0.369 | 0.356 | 0.270 | 0.307 | 0.641 | 0.639 | 0.347 | 0.417 | 0.301 | 0.319 | 0.330 | 0.401 | 0.282 | 0.375 | 0.291 | 0.381 | 0.261 | 0.381 | 0.885 | 0.711 |

## 6.2 ABLATION EXPERIMENT

To verify the rationale for removing the non-linear activation function and the disentangled encoding strategy, we choose our proposed SAMBA as the SOTA benchmark and develop three variant models. Figure 4 shows that SAMBA consistently outperforms other variants. Excluding either the nonlinear activation function or any branch of the disentangled encoding strategy results in a performance decline, demonstrating the effectiveness of our architecture and strategy. After incorporating the patch technique, the impact of the nonlinear activation function decreased. This occurs because the patch technique enriches the information available for each token, thereby enhancing the capture of semantic dependency. Additionally, our analysis shows that removing the cross-variate dependency encoding branch leads to a more significant performance degradation on the Traffic dataset compared to other datasets. This result is intuitive, as variates in traffic scenarios typically exhibit strong interdependencies (Han et al. (2024)).

## 6.3 EVALUATING THE UNIVERSALITY OF DISENTANGLED ENCODING STRATEGY

We utilize Transformer and their variants, including Informer (Zhou et al. (2021))and PatchTST (Nie et al. (2022)), to validate the universality of our proposed disentangled encoding strategy. Table 5 has demonstrated our strategy is effective not only for Mamba but also for Transformer-based models ( Full results can be found in Appendix N.1.). The significant improvements indicate that these models have not properly modeled and introduced cross-variate dependency, whereas our strategy effectively mitigates these issues. Therefore, this disentangled encoding strategy is a model-agnostic method and could potentially be integrated with other models.

## 6.4 EFFICIENCY ANALYSIS OF SAMBA

In general, the time complexity of SAMBA is $O(D^2 \cdot NJ + D \cdot NJ \log(NJ))$ where $D$ is the dimension of token embedding, $T$ is the length of tokens, $N$ is the number of variates, $J$ is the number

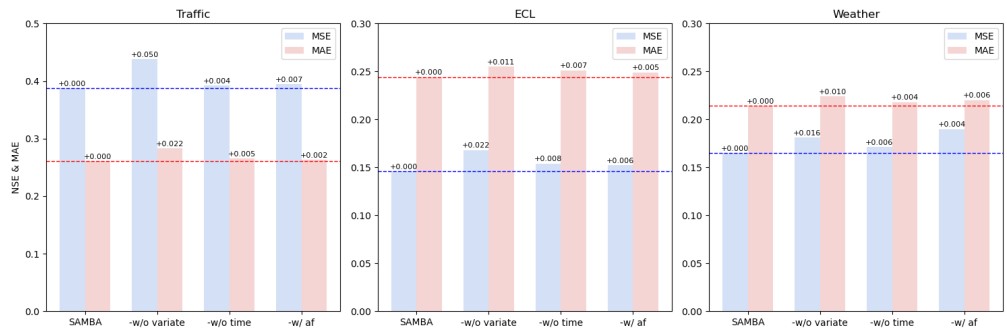

Figure 4: Ablation study of removing nonlinear activation function and disentangled encoding strategy in SAMBA. 4 cases are include: 1) SAMBA; 2) SAMBA without time encoding; 3) SAMBA without variate encoding; 4) Mamba with disentangled encoding.

of patches ( The detailed analysis process is given in Appendix D). This validates that SAMBA maintains the *near linear* time complexity *w.r.t.* of the patch number and the variate number, a key merit of Mamba, while also inheriting the efficiency of the patching strategy.

In addition, we provide a detailed efficiency comparison between SAMBA and the baseline models. As shown in Table 6 and , SAMBA achieves both a faster training speed and a smaller memory usage compared to many SOTA transformer-based models, such as PatchTST and Crossformer, which also employ attention mechanisms in temporal dimensions. Furthermore, SAMBA exhibits a much smaller increase in memory consumption and training time as input lengths grow, underscoring its superior overall efficiency.

Table 5: Performance gains of our disentangled encoding strategy on MAE and MSE.

| Models Metric | | Transformer | | PatchTST | | Informer | |
|---|---|---|---|---|---|---|---|
| | | MSE | MAE | MSE | MAE | MSE | MAE |
| ETTh1 | Original | 0.997 | 0.797 | 0.469 | 0.454 | 1.060 | 0.791 |
| | +disentangled | **0.481** | **0.477** | **0.450** | **0.439** | **0.489** | **0.485** |
| | Promotion | 51.8% | 42.2% | 4.05% | 3.31% | 53.9% | 38.7% |
| ETTm1 | Original | 0.773 | 0.656 | 0.387 | 0.400 | 0.870 | 0.696 |
| | +disentangled | **0.454** | **0.461** | **0.383** | **0.396** | **0.470** | **0.459** |
| | Promotion | 41.3% | 29.7% | 1.03% | 1.00% | 46.0% | 34.1% |
| Weather | Original | 0.657 | 0.572 | 0.259 | 0.281 | 0.634 | 0.548 |
| | +disentangled | **0.258** | **0.279** | **0.248** | **0.278** | **0.271** | **0.330** |
| | Promotion | 60.7% | 51.2% | 4.25% | 1.08% | 57.3% | 39.8% |

Table 6: Efficiency Analysis: The GPU memory (MiB) and speed (running time, s/iter) of each model on ETTm1 dataset. Mem means memory footprint.

| Input Length | 96 | | 336 | | 720 | |
|---|---|---|---|---|---|---|
| Models | Mem | Speed | Mem | Speed | Mem | Speed |
| SAMBA | 448 | 0.0096 | 684 | 0.0145 | 1028 | 0.0268 |
| PatchTST | 790 | 0.0095 | 1980 | 0.0222 | 3264 | 0.0398 |
| iTransformer | 630 | 0.0113 | 632 | 0.0116 | 634 | 0.0121 |
| DLinear | 338 | 0.0043 | 342 | 0.0049 | 344 | 0.0055 |
| TimesNet | 930 | 0.0653 | 1446 | 0.1099 | 2212 | 0.243 |
| Crossformer | 1736 | 0.0502 | 2304 | 0.0664 | 3322 | 0.1031 |
| FEDFormer | 2336 | 0.2021 | 2772 | 0.2262 | 3616 | 0.2365 |
| Autoformer | 2502 | 0.0517 | 4408 | 0.0982 | 7922 | 0.1716 |

# 7 CONCLUSIONS AND FUTURE WORK

In this paper, we identify and rigorously define three crucial dependencies in time series: order dependency, semantic dependency, and cross-variate dependency. Based on these dependencies, we examine the three limitations of existing time series forecasting models: (1) The inability to capture order and semantic dependencies simultaneously. (2) The presence of overfitting issues. (3) The improper introduction of cross-variate dependency. We further empirically verify the superiority of Mamba as an alternative to linear and Transformer models, despite its redundant nonlinear activation functions. Building on these insights, we introduce SAMBA, a model derived from a simplified Mamba with a theoretically guaranteed disentangled encoding strategy. Experimentally, SAMBA achieves SOTA performance, validating the rationality and efficacy of our method design grounded in rich empirical findings. Our efforts further demonstrate the great potential of Mamba in time series modeling. In the future, we aim to delve deeper into diverse time series tasks, pushing the boundaries of analysis with Mamba.

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

# A  THEORETICAL DISCUSSIONS

## A.1  PROOF FOR THEOREM 1

Without loss of generality, we only consider the setting with two variates, *i.e.*, $N = 2$, and treat the 1st variate as the target variable. For simplicity, we introduce the notation $\Psi\left((\mathbf{A}, \mathbf{B})^T\right) := \Psi(\mathbf{A}, \mathbf{B})$. Then, we can express the outputs of $\mathcal{F}_d(\cdot)$ and $\mathcal{F}_{ttv}(\cdot)$ *w.r.t.* the target variate as follows:

$$\mathcal{F}_d(\mathbf{E})^1 = \text{FFN}\left(\Phi(\mathbf{E}^1) \| \Psi\left(\mathbf{E}^1, \mathbf{E}^2\right)^1\right), \tag{5}$$

$$\mathcal{F}_{ttv}(\mathbf{E})^1 = \text{FFN}\left(\Psi\left(\Phi(\mathbf{E}^1), \Phi(\mathbf{E}^2)\right)^1\right). \tag{6}$$

Next, we prove the informativeness of output representation from $\mathcal{F}_d$ compared to that of $\mathcal{F}_{ttv}$. Based on the information loss property of data processing, we have

$$
\begin{aligned}
H(\mathbf{c}^1 | \mathcal{F}_{ttv}(\mathbf{E})^1) &= H\left(\mathbf{c}^1 | \text{FFN}\left(\Psi\left(\Phi(\mathbf{E}^1), \Phi(\mathbf{E}^2)\right)^1\right)\right) \\
&\geq H\left(\mathbf{c}^1 | \text{FFN}\left(\Psi\left(\Phi(\mathbf{E}^1), \Phi(\mathbf{E}^2)\right)^1\right)\right) \\
&= H\left(\mathbf{c}^1 | \text{FFN}\left(\Psi\left(\Phi(\mathbf{E}^1), \phi(\mathcal{O}(\mathbf{E}^2), \mathcal{S}(\mathbf{E}^2)) + \mathcal{Z}(\mathbf{E}^2)\right)^1\right)\right) \\
&\geq H\left(\mathbf{c}^1 | \text{FFN}\left(\Psi\left(\Phi(\mathbf{E}^1), \mathcal{O}(\mathbf{E}^2)\right)^1\right), \mathcal{S}(\mathbf{E}^2), \mathcal{Z}(\mathbf{E}^2)\right).
\end{aligned}
\tag{7}
$$

Given the assumption (1) on the noisy components, we have

$$H(\mathbf{c}^1 | \mathcal{F}_{ttv}(\mathbf{E})^1) \geq H\left(\mathbf{c}^1 | \text{FFN}\left(\Psi\left(\Phi(\mathbf{E}^1), \mathcal{O}(\mathbf{E}^2)\right)^1\right), \mathcal{S}(\mathbf{E}^2)\right). \tag{8}$$

Furthermore, by combining assumption (2) on the semantic dependencies, we have

$$
\begin{aligned}
H(\mathbf{c}^1 | \mathcal{F}_{ttv}(\mathbf{E})^1) &\geq H\left(\mathbf{c}^1 | \text{FFN}\left(\Psi\left(\Phi(\mathbf{E}^1), \mathcal{O}(\mathbf{E}^2)\right)^1\right), \mathcal{S}(\mathbf{E}^2)\right) \\
&= H\left(\mathbf{c}^1 | \text{FFN}\left(\Psi\left(\Phi(\mathbf{E}^1), \mathcal{O}(\mathbf{E}^2)\right)^1\right)\right) \\
&\quad - I\left(\mathbf{c}^1; \mathcal{S}(\mathbf{E}^2) | \text{FFN}\left(\Psi\left(\Phi(\mathbf{E}^1), \mathcal{O}(\mathbf{E}^2)\right)^1\right)\right) \\
&= H\left(\mathbf{c}^1 | \text{FFN}\left(\Psi\left(\Phi(\mathbf{E}^1), \mathcal{O}(\mathbf{E}^2)\right)^1\right)\right) - I(\mathbf{c}^1; \mathcal{S}(\mathbf{E}^2) | \Phi(\mathbf{E}^1), \mathcal{O}(\mathbf{E}^2)) \\
&\geq H\left(\mathbf{c}^1 | \text{FFN}\left(\Psi\left(\Phi(\mathbf{E}^1), \mathcal{O}(\mathbf{E}^2)\right)^1\right)\right) - \epsilon \\
&\geq H\left(\mathbf{c}^1 | \text{FFN}\left(\Phi(\mathbf{E}^1) \| \Psi\left(\Phi(\mathbf{E}^1), \mathcal{O}(\mathbf{E}^2)\right)^1\right)\right) - \epsilon \\
&\geq H\left(\mathbf{c}^1 | \text{FFN}\left(\Phi(\mathbf{E}^1) \| \mathcal{O}(\mathbf{E}^2)\right)\right) - \epsilon.
\end{aligned}
\tag{9}
$$

For $\mathcal{F}_d$, though the cross-variate dependency encoding $\Psi(\cdot)$ does not explicitly model the order dependencies in $\mathbf{E}^2$, it aggregates the patch embedding of the 2nd variate to the target variate at each time step. This enables the $\text{FFN}(\cdot)$ layer, which transforms the input with linear mapping, to further capture the order dependencies. Moreover, according to the empirical findings in Table 1 as well as previous works Zeng et al. (2023); Li et al. (2023), we assume that linear mapping is not worse than other complex cross-time encoders such as Transformers and Mamba in terms of order dependency modeling. Therefore, combining inequality (9), we have

$$H(Y_1 | F_d(\mathbf{E})^1) = H(Y_1 | \text{FFN}\left(\Phi(\mathbf{E}^1) \| \Psi\left(\mathbf{E}^1, \mathbf{E}^2\right)^1\right)) \tag{10}$$

$$= H\left(Y_1 | \text{FFN}\left(\Phi(\mathbf{E}^1)\right) \| \text{FFN}\left(\Psi\left(\mathbf{E}^1, \mathbf{E}^2\right)^1\right)\right) \tag{11}$$

$$\leq H(Y_1 | \text{FFN}\left(\Phi(\mathbf{E}^1)\right) \| \mathcal{O}(\mathbf{E}^2)) \tag{12}$$

$$\leq H(Y_1 | \text{FFN}\left(\Phi(\mathbf{E}^1) \| \mathcal{O}(\mathbf{E}^2)\right)) \tag{13}$$

$$\leq H(Y_1 | F_{ttv}(\mathbf{E})^1) + \epsilon. \tag{14}$$

## A.2 RATIONALITY OF THE ASSUMPTION (2) IN THEOREM 1

We interpret the rationality of the assumption from two perspectives: (1) When the lookback window is short, like the often used 1-hour window in traffic prediction task Wu et al. (2021), there exist low-density semantic dependencies within each variate's observations, while the order dependencies from other variates could lead to a direct impact on the future of the target variate due to continuity in physical laws, *e.g.*, the increasing traffic volume spreads to neighboring road segments; (2) Even when the lookback window becomes longer, the widely adopted patching strategy compresses the most useful local semantics into a $D$-dim feature vector, which is attached to each time step of embedding $\mathbf{E}$. Consequently, the semantic dependencies extracted along the temporal dimension of $\mathbf{E}$ usually become less informative.

## B IMPLEMENTATION DETAILS

### B.1 DATASET DESCRIPTIONS

We perform tests on seven actual-world datasets to assess the capabilities of our new SAMBA model. The datasets we use are:

**Electricity Transformer Time Series (ETT)**    This dataset, as reported by Zhou et al. (2021), spans from July 2016 to July 2018 and monitors seven aspects of an electricity transformer's operation. It is divided into four subsets, with ETTh1 and ETTh2 offering hourly readings and ETTm1 and ETTm2 providing data every quarter of an hour.

**Weather**    This dataset encompasses 21 different weather parameters. It draws information from the Max Planck Biogeochemistry Institute's Weather Station, with readings taken every ten minutes throughout 2020.

**Electricity Consumption Log (ECL)**    This dataset catalogs hourly electricity usage for 321 different clients.

**Traffic**    This dataset gathers information on road occupancy rates. It uses data from 862 sensors positioned on freeways in the San Francisco Bay area, with hourly updates from January 2015 to December 2016.

**Exchange**    This dataset collects daily exchange rate panel data from 8 countries between 1990 and 2016 and covers a long time span of foreign exchange market data across multiple countries. These datasets provide a diverse and comprehensive set of data to thoroughly evaluate the performance of our proposed SAMBA model across various real-world scenarios.

**Solar-Energy**    This dataset records the solar power production data from 137 photovoltaic (PV) plants in 2006. The data is sampled every 10 minutes, capturing detailed variations in power generation at each plant.

### B.2 BASELINE METHODS AND IMPLEMENTATIONS

We briefly describe the selected baselines:

**CARD** (Wang et al. (2024)) use a patching strategy and two transformer encoders to encode cross-time and cross-variate dependencies. Later, directly mixes cross-time and cross-variate information at each model layer. In this paper, we directly utilize the model from their GitHub repository and apply the same training loss function as in the Time Series Library GitHub repository to ensure fairness.

**FTP** (Zhou et al. (2023)) uses GPT2 to forecast time series and is implemented by their provided GitHub repository. .

**iTransformer** (Liu et al. (2023)) only inverts the time series and, without making any modifications to the transformer architecture, uses the Feed-Forward Network layer to capture cross-time dependency

Table 7: **Detailed dataset descriptions**. *Dim* refers to the number of variates present within each dataset.*Dataset Size* is indicative of the overall count of time points that are categorized into three segments: training, validation, and testing. *Prediction Length* denotes the future time points to be predicted and four prediction settings are included in each dataset. *Frequency* represents the rate at which time points are sampled, essentially detailing the time interval between successive data points.

| Task | Dataset | Dim | Prediction Length | Dataset Size | Frequency | Information |
|------|---------|-----|-------------------|--------------|-----------|-------------|
| Low-dim | ETTh1 | 7 | {96, 192, 336, 720} | (8545, 2881, 2881) | Hourly | Electricity |
| | ETTh2 | 7 | {96, 192, 336, 720} | (8545, 2881, 2881) | Hourly | Electricity |
| | ETTm1 | 7 | {96, 192, 336, 720} | (34465, 11521, 11521) | 15min | Electricity |
| | ETTm2 | 7 | {96, 192, 336, 720} | (34465, 11521, 11521) | 15min | Electricity |
| | Exchange | 8 | {96, 192, 336, 720} | (5120, 665, 1422) | Daily | Economy |
| High-dim | Weather | 21 | {96, 192, 336, 720} | (36792, 5271, 10540) | 10min | Weather |
| | Solar-Energy | 137 | {96, 192, 336, 720} | (366001, 5161, 10417) | 10min | Energy |
| | ECL | 321 | {96, 192, 336, 720} | (18317, 2633, 5261) | Hourly | Electricity |
| | Traffic | 862 | {96, 192, 336, 720} | (12185, 1757, 3509) | Hourly | Transportation |

and employs the transformer to capture cross-variate dependency. In this paper, we implement it by using the Time Series Library GitHub repository.

**RLinear** (Li et al. (2023)) uses a single-layer linear model with RevIN has been added for time series forecasting. In this paper, we implement it by using the Time Series Library GitHub repository.

**PatchTST** (Nie et al. (2022)) employs of patches and channel-independent strategy without altering the transformer architecture. In this paper, we implement it by using the Time Series Library GitHub repository.

**Crossformer** (Zhang & Yan (2022)) is a Transformer-based model designed to enhance multivariate time series forecasting by leveraging cross-variate dependency. In this paper, we implement it by using the Time Series Library GitHub repository.

**TiDE** (Das et al. (2023)) is an MLP-based encoder-decoder architecture for long-term time-series forecasting that excels in efficiency and performance by incorporating dense MLPs for encoding past time-series data and decoding future predictions, while also managing covariates and nonlinear relationships. In this paper, we implement it by using the Time Series Library GitHub repository.

**TimesNet** (Wu et al. (2022)) is a task-general foundation model for time series analysis that excels in capturing complex temporal variations by transforming 1D time series data into 2D tensors and utilizing a parameter-efficient inception block within its TimesBlock architecture. In this paper, we implement it by using the Time Series Library GitHub repository.

**DLinear** (Zeng et al. (2023)) uses a single-layer linear model with The seasonal term and trend term decoupled has been added for time series forecasting. In this paper, we implement it by using the Time Series Library GitHub repository.

**SCINet** (Liu et al. (2022a)) is a novel neural network architecture for time series modeling and forecasting that employs a hierarchical downsample-convolve-interact structure. In this paper, we implement it by using the Time Series Library GitHub repository.

**FEDformer** (Zhou et al. (2022)) is a Transformer-based architecture that employs seasonal-trend decomposition integrated with frequency-enhanced blocks to seize cross-time dependencies for forecasting. In this paper, we implement it by using the Time Series Library GitHub repository.

**Stationary** (Liu et al. (2022b)) is a Transformer-based model that uses De-stationary Attention mechanism to recover the intrinsic non-stationary information into temporal dependencies by approximating distinguishable attentions learned from unstationarized series. In this paper, we implement it by using the Time Series Library GitHub repository.

**Autoformer** (Wu et al. (2021)) is a Transformer-based model that utilizes a decomposition architecture coupled with an Auto-Correlation mechanism to effectively capture cross-time dependencies, thereby enhancing forecasting capabilities. In this paper, we implement it by using the Time Series Library GitHub repository.

### B.3 EXPERIMENT DETAILS

All experiments are implemented using PyTorch on a single NVIDIA A40 GPU. We optimize the model using ADAM with an initial learning rate in the set $\{10^{-3}, 5 \times 10^{-4}, 10^{-4}\}$ and L2 loss. The batch size is uniformly set to $\{32 - 128\}$, depending on the maximum GPU capacity. To ensure that all models can be fitted effectively, the number of training epochs is fixed at 10 with an early stopping mechanism. By default, our SAMBA encodes both the time and variate dimensions using a single-layer SAMBA encoding, with a patch length of 16 and a stride of 8.

The baseline models we use for comparison are implemented based on the Time Series Library (Wu et al. (2022)) repository, and they are run with the hyperparameter settings they recommend. For CARD, we download the model from https://github.com/wxie9/CARD/tree/main, but utilize the same training loss as the other models (i.e. the the Time Series Library GitHub repository recommend) for a fair comparison. Additionally, the full results of the predictions come from the outcomes reported by the iTransformer (Liu et al. (2023)), ensuring fairness.

### B.4 IMPLEMENTATION DETAILS OF SECTION 4

For all experiments in Section 4, we fixed the lookback length to 96, adhering to the configuration recommended by TimesNet(Wu et al. (2022)). The Linear, MLP, Transformer, Mamba, and PatchTST models are all implemented based on the TimesNet repository. For the Linear model, we base it on the DLinear model, removing the decomposition module to achieve the simplest linear forecasting model and applying the recommended parameters of DLinear. For the MLP, we add an extra linear layer and a ReLU activation function to the linear model, placing the activation function between the two linear layers and setting the hidden layer dimension uniformly to 128. The Transformer model is fully implemented based on the TimesNet repository, with the number of layers set to three to better highlight the impact of the nonlinear activation function, using the recommended hyperparameters. We utilize the Mamba model as implemented by (Gu & Dao (2023)), setting the dimension expansion factor to 2, the convolution kernel size to 4, the state expansion factor to 16 by default, and the expand factor to 4 by default. For PatchMamba, we incorporate a patch strategy akin to PatchTST on top of the aforementioned Mamba. PatchLinear, after patching the time series, averages all values within the patch to reduce dimensionality.

For the implementation that removes the nonlinear activation function, as the Linear model does not contain a nonlinear activation function, we opt to replace it with an MLP and eliminate the nonlinear activation function between the two linear layers. For the simplified variant of the Transformer, we remove the nonlinear activation function within the FFN of the Encoder, which facilitates comparison with PatchTST, given that PatchTST is a model based on the Transformer's encoder. For Mamba, we remove the nonlinear activation function between the 1D convolutional layer and the SSM.

## C WHY CHOOSE MAMBA TO ENCODE CROSS-VARIATE DEPENDENCY?

### C.1 THE RATIONALE OF USING MAMBA TO ENCODE UNORDER VARIATES SEQUENCE

Using recursive models (e.g., RNN, Mamba) to process unordered sequences is feasible. For instance, GraphSAGE (Hamilton et al. (2017)) uses LSTM to aggregate unordered node neighbors. To comprehensively consider the interactions between variates, we adopt a bidirectional Mamba to model cross-variate dependency.

### C.2 ADVANTAGE OF USING MAMBA TO ENCODE CROSS-VARIATE DEPENDENCY

Compared to the Transformer, Mamba models cross-variate dependency more efficiently. To compare the efficiency of Transformer and Mamba in modeling cross-variate dependency, we implement the branch of our proposed disentangled encoding strategy that models variate relationships using PatchTST (PatchTST$_v$) and SAMBA (SAMBA$_v$) respectively. To better demonstrate the efficiency differences in encoding variates, we evaluate two models on the Traffic and ECL datasets, which have the highest number of variates in our experiments. Table 8 shows that, compared to the Transformer, Mamba has a faster training speed and a smaller GPU memory footprint. As a result, Mamba is able to encode cross-variate dependency more efficiently.

Table 8: The GPU memory (MiB) and speed (s/iter) of each model

| Models | SAMBA$_v$ | | PatchTST$_v$ | |
|--------|------|-------|------|-------|
| Metric | MiB | s/iter | MiB | s/iter |
| Traffic | 4900 | 0.1245 | 9784 | 0.1658 |
| ECL | 3786 | 0.0981 | 5118 | 0.1264 |

## C.3 REPLACING MAMBA WITH TRANSFORMER TO ENCODE CROSS-VARIATE DEPENDENCY

To better evaluate the superiority of Mamba encoding in capturing cross-variate dependencies, we conducted the following experiment. We design a variant, SAMBA$_t$, where we keep the other modules of SAMBA unchanged and replace the SAMBA implementation with the Transformer implementation in the branch of our proposed disentangled encoding strategy that encodes cross-variate dependency. Then, we select two representative datasets: ETTm1 (with fewer variates) and Traffic (with more variates) to evaluate the performance of SAMBA and SAMBA$_t$. Table 9 shows that SAMBA performed better.

Table 9: Performance Comparison

| Models | SAMBA | | SAMBA$_t$ | |
|--------|------|------|------|------|
| Metric | MSE | MAE | MSE | MAE |
| ETTm1 | 0.315 | 0.357 | 0.324 | 0.365 |
| Traffic | 0.388 | 0.261 | 0.411 | 0.277 |

## D COMPUTATIONAL COMPLEXITY ANALYSIS

First, the time complexity of each Mamba block is $O(D^2T + DT \log T)$ when adopting the convolution-based implementation ( Fu et al. (2022); Gu & Dao (2023)), where $D$ is the dimension of token embeddings and $T$ is the length of tokens. As a SAMBA block only removes the nonlinear activation before the selection mechanism, it possesses the same time complexity.

Next, we consider the time complexity of the disentangled modeling. (i) The linear projection in the tokenization process takes time $O(NJPD)$. (ii) In the CI learning, the token length is the variate number $N$, thereby taking time $O(N(D^2J + DJ \log J))$. (iii) In the CD learning, the dimension of token embeddings becomes the patch number $J$, while the token length becomes the hidden dimension $D$. Considering the bidirectional design, it takes time $O(2J(D^2N + DN \log N))$. (iv) The FFN-based aggregation operation and the final prediction head take time $O(NJD^2 + NJDS)$.

In general, the time complexity of SAMBA is $O(NJPD + N(D^2J + DJ \log J) + 2J(D^2N + DN \log N) + (NJD^2 + NJDS)) = O(D^2 \cdot NJ + D \cdot NJ \log(NJ))$. This validates that SAMBA maintains the *near linear* time complexity *w.r.t.* of the patch number and the variate number, a key merit of Mamba, while also inheriting the efficiency of the patching strategy.

## E ADDITIONAL ABLATIONAL EXPERIMENT

To evaluate the effectiveness of our proposed simplified architecture and disentangled encoding strategy, we introduce three variants and use SAMBA as the baseline. In Figure 4, we demonstrate the effectiveness of removing the nonlinear activation function and the disentangled encoding strategy on the Traffic, ECL, and Weather datasets. However, due to page limitations, we provide additional dataset results and detailed analysis. Figure 5 shows the results in Solar-Energy, ETTm1, and ETTm2 datasets.

In Figures 4 and 5, SAMBA consistently shows better performance compared to other variants, mainly due to the removal of the nonlinear activation function, which mitigates overfitting, and the disentangled encoding strategy, which effectively captures both cross-time and cross-variate dependencies. Although the influence of the nonlinear activation function decreased after introducing the patch technique, this is because the patch technique increases the information available for each token, enhancing semantic dependency. Nonlinear activation functions typically capture this enhanced dependency.

In datasets with a smaller number of variates, such as ETTm1 and ETTm2, the performance impact of removing cross-time dependency encoding (-w/o time) is greater than that of removing cross-variate dependency encoding (-w/o variate), indicating that cross-time dependency plays a more important role in these datasets. However, as the number of variates increases (e.g., 21 variables in Weather, 137 in Solar-Energy, 321 in ECL, and 862 in Traffic), the -w/o variate variant performs noticeably worse than -w/o time, suggesting that cross-variate dependency has a more significant impact in larger datasets.

Further evidence supporting this conclusion is seen in the ETTm1 and ETTm2 datasets, where SAMBA shows only marginal improvements over the -w/o variate variant, indicating weaker cross-variate dependency, consistent with previous studies supporting CI strategies (Nie et al. (2022)). In contrast, on datasets like Weather, Solar-Energy, ECL, and Traffic, SAMBA significantly outperforms the -w/o variate variant, highlighting the crucial role of cross-variate dependency in multivariate time series forecasting. Previous studies(Liu et al. (2023); Zhang & Yan (2022)), which jointly encoded cross-time and cross-variate dependencies, mistakenly mixed information from the time and variate dimensions, compromising the unique physical meaning of each channel. Our proposed SAMBA, by adopting a disentangled encoding strategy, effectively captures both types of dependencies, leading to enhanced performance on complex multivariate datasets.

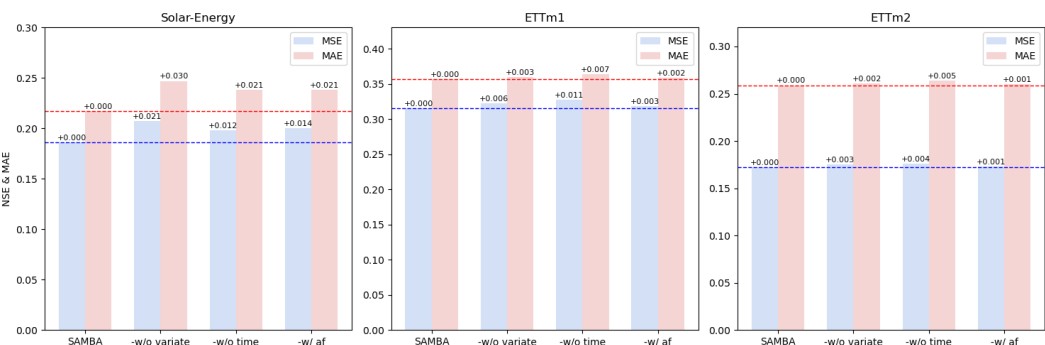

Figure 5: Ablation study of removing nonlinear activation function and disentangled encoding strategy in SAMBA. 4 cases are include: 1) SAMBA; 2) SAMBA without time encoding; 3) SAMBA without variate encoding; 4) Mamba with disentangled encoding.

## F  ADDITIONAL EXPERIMENTS TO VERIFY MAMBA'S ADVANTAGES

In the empirical exploration of Section 4, we arrive at two conclusions:

- Compared to the Linear Model and Transformer, Mamba effectively captures both order dependency and semantic dependency simultaneously.
- Removing the nonlinear activation function in Mamba is beneficial.

To further validate these conclusions comprehensively, we conducted experiments on additional datasets.

### F.1  ORDER DEPENDENCY

We additionally included the SOTA LTSF model, iTransformer, as one of our baselines. It is worth noting that iTransformer encodes temporal relationships using a Linear Model and then uses a Transformer encoder to capture cross-variate dependencies. Ideally, it is expected to be sensitive to order as well. The supplementary results align with the conclusions drawn in Section 4 and the expectations of iTransformer.

Linear Model, Mamba, and iTransformer effectively capture order dependency, exhibiting performance degradation on shuffled ETTm1 and Exchange datasets. In contrast, Transformer shows no performance degradation on Exchange and even achieves improvements, a phenomenon consistent with observations in DLinear, indicating that the self-attention mechanism in Transformer fails to capture order dependency.

Additionally, we further validated our conclusions on multivariate datasets Traffic. Significant performance degradation is observed for all models on these datasets, which can be attributed to the strong cross-variate dependencies present. Shuffling disrupts not only order dependency but also cross-variate dependency.This results in Mamba's performance degradation being greater than that of the Linear Model, as the Linear Model does not account for cross-variate dependency. On the other hand, despite being affected by cross-variate dependency, the performance degradation of the transformer is still less than that of the Linear Model, indicating its weaker ability to capture order dependency.

Overall, the additional experimental results support our conclusion that Mamba can effectively capture order dependency, whereas the transformer, due to the permutation invariance of self-attention, cannot achieve this effectively.

Table 10: Additional experiments for analyzing order dependency.The input sequence length is 96 for all baselines. *Avg.Drop* means the average performance degradation.

| Models | | | | | Linear Model | | | | Mamba | | | | Transformer | | | | iTransformer | | | |
|---|---|---|---|---|---|---|---|---|---|---|---|---|---|---|---|---|---|---|---|---|
| Metric | | | O.MSE | S.MSE | O.MAE | S.MAE | O.MSE | S.MSE | O.MAE | S.MAE | O.MSE | S.MSE | O.MAE | S.MAE | O.MSE | S.MSE | O.MAE | S.MAE |
| ETTm1 | 96 | | 0.383 | 0.988 | 0.400 | 0.697 | 0.517 | 0.922 | 0.508 | 0.688 | 0.643 | 0.884 | 0.575 | 0.643 | 0.345 | 0.892 | 0.378 | 0.610 |
| | 192 | | 0.413 | 0.986 | 0.415 | 0.697 | 0.575 | 0.931 | 0.546 | 0.699 | 0.805 | 1.01 | 0.664 | 0.730 | 0.383 | 0.903 | 0.395 | 0.617 |
| | 336 | | 0.441 | 0.987 | 0.435 | 0.698 | 0.730 | 0.957 | 0.634 | 0.703 | 0.882 | 1.12 | 0.737 | 0.817 | 0.423 | 0.923 | 0.420 | 0.630 |
| | 720 | | 0.497 | 0.992 | 0.469 | 0.704 | 0.873 | 0.973 | 0.704 | 0.723 | 0.928 | 1.12 | 0.752 | 0.800 | 0.489 | 0.932 | 0.456 | 0.641 |
| Avg. Drop | | | - | 127.97% | - | 62.55% | - | 40.37% | - | 17.60% | - | 22.40% | - | 6.55% | - | 122.56% | - | 51.5% |
| Exchange | 96 | | 0.0832 | 0.210 | 0.201 | 0.332 | 1.260 | 1.401 | 0.915 | 0.943 | 0.730 | 0.738 | 0.782 | 0.722 | 0.0869 | 0.242 | 0.207 | 0.358 |
| | 192 | | 0.179 | 0.325 | 0.299 | 0.414 | 1.398 | 1.626 | 1.040 | 1.060 | 1.304 | 1.284 | 0.913 | 0.949 | 0.179 | 0.374 | 0.301 | 0.450 |
| | 336 | | 0.338 | 0.521 | 0.418 | 0.534 | 1.835 | 1.921 | 1.111 | 1.141 | 1.860 | 1.862 | 1.090 | 1.085 | 0.331 | 0.555 | 0.417 | 0.557 |
| | 720 | | 0.903 | 1.167 | 0.714 | 0.822 | 3.940 | 4.023 | 1.687 | 1.697 | 3.860 | 3.865 | 1.684 | 1.685 | 0.856 | 1.202 | 0.698 | 0.841 |
| Avg. Drop | | | - | 47.89% | - | 28.80% | - | 6.38% | - | 1.85% | - | -0.06% | - | -0.63% | - | 63.33% | - | 35.89% |
| Traffic | 96 | | 0.656 | 1.679 | 0.403 | 0.882 | 0.634 | 1.859 | 0.363 | 0.967 | 0.788 | 1.846 | 0.467 | 0.918 | 0.396 | 1.945 | 0.271 | 0.976 |
| | 192 | | 0.609 | 1.665 | 0.382 | 0.876 | 0.637 | 2.387 | 0.375 | 1.120 | 0.850 | 1.715 | 0.483 | 0.878 | 0.416 | 1.751 | 0.279 | 0.916 |
| | 336 | | 0.615 | 1.671 | 0.385 | 0.880 | 0.674 | 2.428 | 0.386 | 1.127 | 0.836 | 1.676 | 0.480 | 0.861 | 0.430 | 1.801 | 0.287 | 0.938 |
| | 720 | | 0.656 | 1.697 | 0.405 | 0.882 | 0.732 | 1.751 | 0.414 | 0.916 | 0.859 | 1.746 | 0.490 | 0.880 | 0.559 | 1.978 | 0.375 | 0.971 |
| Avg. Drop | | | - | 164.67% | - | 123.49% | - | 214.72% | - | 168.53% | - | 109.51% | - | 84.22% | - | 315.05% | - | 213.61% |

## F.2 SEMANTIC DEPENDENCY

Benefiting from the approach of aggregating time steps into subseries-level patches, where patches in time series data enhance locality and capture comprehensive semantic information that cannot be achieved at the point level, patching has been widely adopted as a method to enhance the semantics of time series (Nie et al. (2022); Zhang & Yan (2022); Liu et al. (2023)). As shown in Tabel 11, on datasets like Exchange and Traffic, which exhibit underlying dynamics and shifting multivariate effects, patching demonstrates significant performance improvements, consistent with observations from PatchTST (Nie et al. (2022)). However, we observe a performance drop when applying patching on the Solar-Energy dataset, which might be attributed to the excessive presence of zero values in Solar-Energy, disrupting the continuity and stationarity of the sequence.

Table 11: Results of the linear model, Mamba, and Transformer w/ or w/o patching.

| Models | Linear Model | | Patch+Linear Model | | Mamba | | Patch+Mamba | | Transformer | | Patch+Transformer | |
|---|---|---|---|---|---|---|---|---|---|---|---|---|
| Metric | MSE | MAE | MSE | MAE | MSE | MAE | MSE | MAE | MSE | MAE | MSE | MAE |
| **Exchange** 96 | 0.0832 | 0.201 | 0.0823 | 0.207 | 1.260 | 0.915 | 0.0871 | 0.207 | 0.989 | 0.782 | 0.0861 | 0.204 |
| 192 | 0.179 | 0.299 | 0.165 | 0.302 | 1.398 | 1.040 | 0.176 | 0.298 | 1.265 | 0.913 | 0.183 | 0.303 |
| 336 | 0.338 | 0.418 | 0.285 | 0.401 | 1.835 | 1.111 | 0.327 | 0.413 | 1.860 | 1.090 | 0.332 | 0.417 |
| 720 | 0.903 | 0.714 | 0.799 | 0.685 | 3.940 | 1.687 | 0.853 | 0.694 | 3.860 | 1.684 | 0.854 | 0.697 |
| Avg | 0.376 | 0.408 | 0.333 | 0.399 | 2.254 | 1.188 | 0.361 | 0.403 | 1.993 | 1.117 | 0.364 | 0.405 |
| **Traffic** 96 | 0.656 | 0.403 | 0.867 | 0.523 | 0.634 | 0.363 | 0.430 | 0.275 | 0.788 | 0.467 | 0.469 | 0.305 |
| 192 | 0.609 | 0.382 | 0.827 | 0.511 | 0.637 | 0.375 | 0.443 | 0.281 | 0.850 | 0.483 | 0.476 | 0.309 |
| 336 | 0.615 | 0.385 | 0.839 | 0.515 | 0.674 | 0.386 | 0.457 | 0.288 | 0.836 | 0.480 | 0.491 | 0.314 |
| 720 | 0.656 | 0.405 | 0.867 | 0.523 | 0.732 | 0.414 | 0.493 | 0.306 | 0.859 | 0.490 | 0.526 | 0.332 |
| Avg | 0.634 | 0.394 | 0.850 | 0.518 | 0.669 | 0.385 | 0.456 | 0.288 | 0.833 | 0.480 | 0.491 | 0.315 |
| **Solar-Energy** 96 | 0.326 | 0.346 | 0.357 | 0.439 | 0.190 | 0.248 | 0.204 | 0.243 | 0.201 | 0.269 | 0.218 | 0.264 |
| 192 | 0.363 | 0.364 | 0.373 | 0.446 | 0.224 | 0.292 | 0.237 | 0.265 | 0.233 | 0.289 | 0.250 | 0.284 |
| 336 | 0.402 | 0.378 | 0.395 | 0.454 | 0.229 | 0.315 | 0.254 | 0.277 | 0.232 | 0.294 | 0.271 | 0.300 |
| 720 | 0.402 | 0.368 | 0.393 | 0.445 | 0.223 | 0.295 | 0.254 | 0.278 | 0.216 | 0.280 | 0.271 | 0.295 |
| Avg | 0.373 | 0.364 | 0.380 | 0.446 | 0.217 | 0.288 | 0.237 | 0.266 | 0.220 | 0.283 | 0.252 | 0.286 |

## F.3 NONLINEAR ACTIVATION FUNCTION

We tested the impact of nonlinear activation functions on the Exchange and Traffic datasets. As shown in Table 12:

1. For Mamba, removing the nonlinear activation function is always beneficial, regardless of whether patching is applied. This supports the design of our proposed SAMBA block.

2. For Transformer, removing the nonlinear activation function significantly improves performance on datasets with relatively limited semantics. Additionally, even on datasets with relatively strong semantics, such as Traffic, removing the activation function consistently proves advantageous. This indicates that the Transformer architecture includes excessive, unnecessary nonlinearity. The use of patches mitigates this issue, aligning with our observations in Section. 4, as patching enhances the semantics of time series, thereby allowing the nonlinear activation function to contribute more effectively.

3. For MLP, removing the nonlinear activation function also yields significant improvements on datasets with limited semantics. However, the expected performance drop on semantically strong datasets like Traffic aligns with expectations. Without the activation function, MLP behaves more like a linear model, lacking the capacity to capture semantic dependencies.

In summary, the above results suggest that appropriately leveraging nonlinearity is beneficial for handling semantic dependencies. However, existing complex models (e.g., Transformer and Mamba) are prone to overfitting. In particular, for **Mamba**, removing nonlinear activation functions is consistently advantageous.

To further analyze the role of activation functions, we evaluated the training curves of models on the Exchange and Traffic datasets. Consistent with the analysis in Section 4, removing nonlinear activation functions alleviates the overfitting issues observed during training.

Table 12: Ablation study on nonlinear activation function, we report the average performance. 'Original' means vanilla model, '-n' means removing the nonlinear activation function and 'Patch+' means using patching.

| Models | MLP | | Patch+MLP | | Mamba | | Patch+Mamba | | Transformer | | Patch+Transformer | |
|---|---|---|---|---|---|---|---|---|---|---|---|---|
| Metric | MSE | MAE | MSE | MAE | MSE | MAE | MSE | MAE | MSE | MAE | MSE | MAE |
| **ETTm1** Original | 0.415 | 0.421 | 0.424 | 0.427 | 0.674 | 0.598 | 0.402 | 0.412 | 0.815 | 0.682 | 0.414 | 0.422 |
| Original-n | 0.406 | 0.411 | 0.411 | 0.412 | 0.635 | 0.585 | 0.399 | 0.414 | 0.653 | 0.600 | 0.406 | 0.417 |
| **Improvement** | 2.17% | 2.37% | 3.07% | 3.39% | 5.79% | 2.17% | 0.75% | -0.48% | 19.88% | 12.02% | 1.93% | 1.18% |
| **Exchange** Original | 0.398 | 0.419 | 0.403 | 0.448 | 2.255 | 1.189 | 0.361 | 0.403 | 1.994 | 1.117 | 0.364 | 0.405 |
| Original-n | 0.374 | 0.407 | 0.321 | 0.399 | 2.122 | 1.143 | 0.357 | 0.400 | 1.194 | 0.895 | 0.362 | 0.404 |
| **Improvement** | 6.03% | 2.86% | 20.19% | 10.93% | 5.90% | 3.79% | 1.02% | 0.74% | 40.13% | 19.87% | 0.58% | 0.25% |
| **Traffic** Original | 0.554 | 0.366 | 0.718 | 0.384 | 0.669 | 0.385 | 0.456 | 0.288 | 0.833 | 0.480 | 0.491 | 0.315 |
| Original-n | 0.621 | 0.400 | 0.652 | 0.413 | 0.658 | 0.381 | 0.453 | 0.285 | 0.829 | 0.479 | 0.499 | 0.322 |
| **Improvement** | -12.27% | -9.28% | 9.16% | -7.68% | 1.57% | 0.98% | 0.55% | 0.87% | 0.42% | 0.16% | -1.78% | -2.30% |

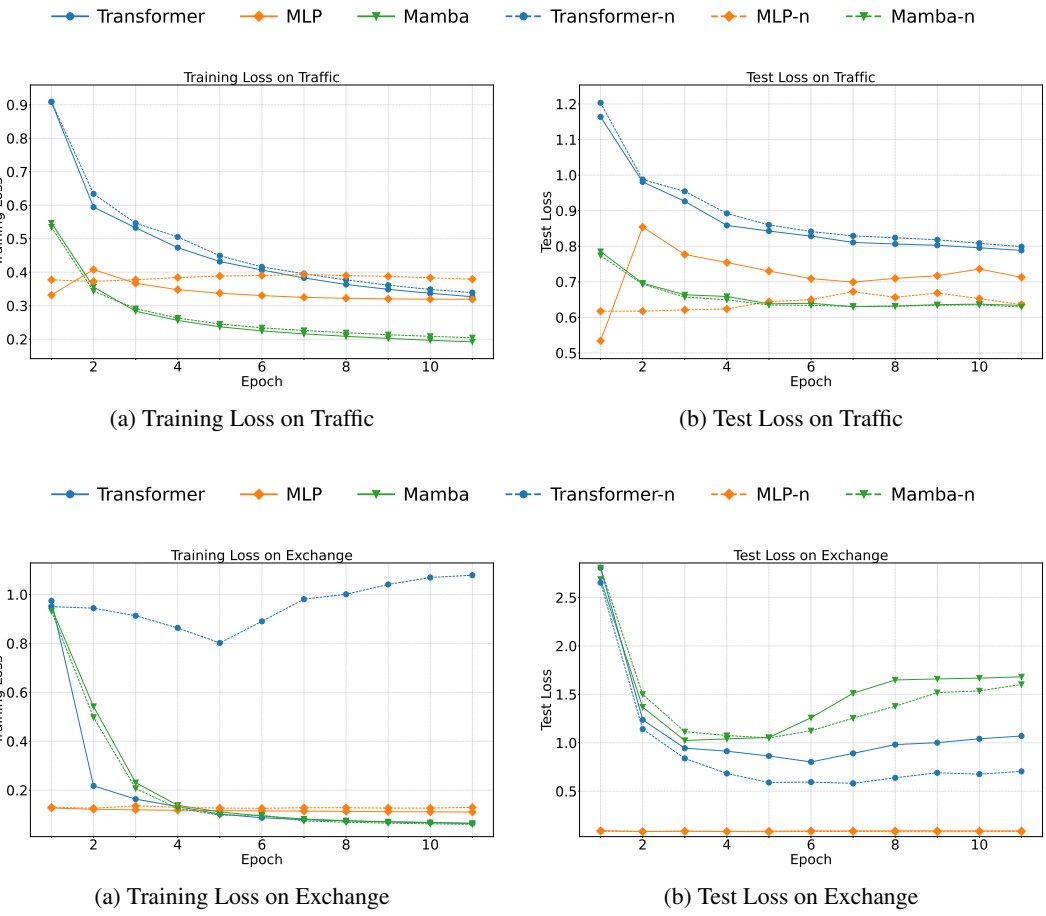

(a) Training Loss on Traffic  (b) Test Loss on Traffic

(a) Training Loss on Exchange  (b) Test Loss on Exchange

# G ADDITIONAL EFFICIENCY ANALYSIS

In the main text, we provide a detailed efficiency comparison between SAMBA and the baseline models on the ETTm1 dataset. Here we provide the quantitive results on Traffic dataset in Table 13. It

should be noted that that SAMBA achieves both a faster training speed and a smaller memory usage compared to many SOTA transformer-based models, such as PatchTST and Crossformer, which also employ attention mechanisms in temporal dimensions. Furthermore, SAMBA exhibits a much smaller increase in memory consumption and training time as input lengths grow, underscoring its superior overall efficiency.

Table 13: Efficiency Analysis: The GPU memory (MiB) and speed (running time, s/iter) of each model on Traffic dataset. Mem means memory footprint.

| Input Length | 96 | | 336 | | 720 | |
|---|---|---|---|---|---|---|
| Models | Mem | Speed | Mem | Speed | Mem | Speed |
| SAMBA | 2235 | 0.0403 | 2275 | 0.0711 | 2311 | 0.1232 |
| PatchTST | 3065 | 0.0658 | 12299 | 0.2382 | 25023 | 0.4845 |
| iTransformer | 3367 | 0.0456 | 3389 | 0.0465 | 3411 | 0.0482 |
| DLinear | 579 | 0.0057 | 619 | 0.0082 | 681 | 0.0139 |
| TimesNet | 6891 | 0.2492 | 7493 | 0.4059 | 8091 | 0.6289 |
| Crossformer | 21899 | 0.1356 | 40895 | 0.1369 | 69711 | 0.1643 |
| FEDFormer | 1951 | 0.1356 | 1957 | 0.1369 | 2339 | 0.1643 |
| Autoformer | 1489 | 0.0309 | 1817 | 0.0362 | 2799 | 0.0457 |

## H  HYPERPARAMETER SENSITIVITY

To verify the robustness of SAMBA, we evaluate three important hyper-parameters of SAMBA: the hidden dimension $d_{\text{model}}$ that determines the model's capacity to learn complex representations from the set $\{64, 128, 256, 512, 1024\}$, the state expansion factor $d_{\text{state}}$ that controls the internal expansion of state representations from the set $\{2, 4, 8, 16, 32, 64, 128\}$, and the convolutional kernel size $d_{\text{conv}}$ that influences how much information is aggregated across time steps in $\{2, 4\}$.

- Based on Figure 8, we can observe that the optimal hidden dimension varies across different datasets. For the Weather dataset, the best hidden dimension is 512, while for ETTm1 and ETTh1, it is 128. This indicates that datasets with higher dimensionality tend to require larger hidden dimensions to capture more complex information in the time series. Notably, the performance was relatively stable across all values, suggesting that SAMBA is robust to variations in the hidden dimension.

- Figure 8 shows that the MSE score does not significantly vary with different choices of state expansion factor $d_{\text{state}}$, indicating the robustness of our model to the state expansion factor $d_{\text{state}}$ hyperparameter.

- Figure 8 shows that SAMBA maintains similar MSE values regardless of kernel size, pointing to the robustness of the model when varying this parameter. This stability implies that SAMBA's time-series dependency ability are not heavily dependent on kernel size.

- Figure 8 demonstrates that the MSE does not vary significantly with changes in patch size, indicating that our model is robust to the patch size hyperparameter. Experimental results suggest that the optimal patch size depends on the specific dataset, but 8, 16 generally yield favorable results.

## I  INCREASING LOOKBACK LENGTH

It is intuitive to expect that using a longer lookback window would enhance forecasting performance. However, previous research has shown that most transformer-based models cannot leverage the increased receptive field to improve predictive accuracy effectively (Nie et al. (2022)). As a result, SOTA transformer-based models, such as iTransformer and PatchTST, have improved their design to more effectively utilize a longer lookback window, leading to improved forecasting performance. In evaluating SAMBA, we explore two key questions: (1) Can SAMBA effectively take advantage of a

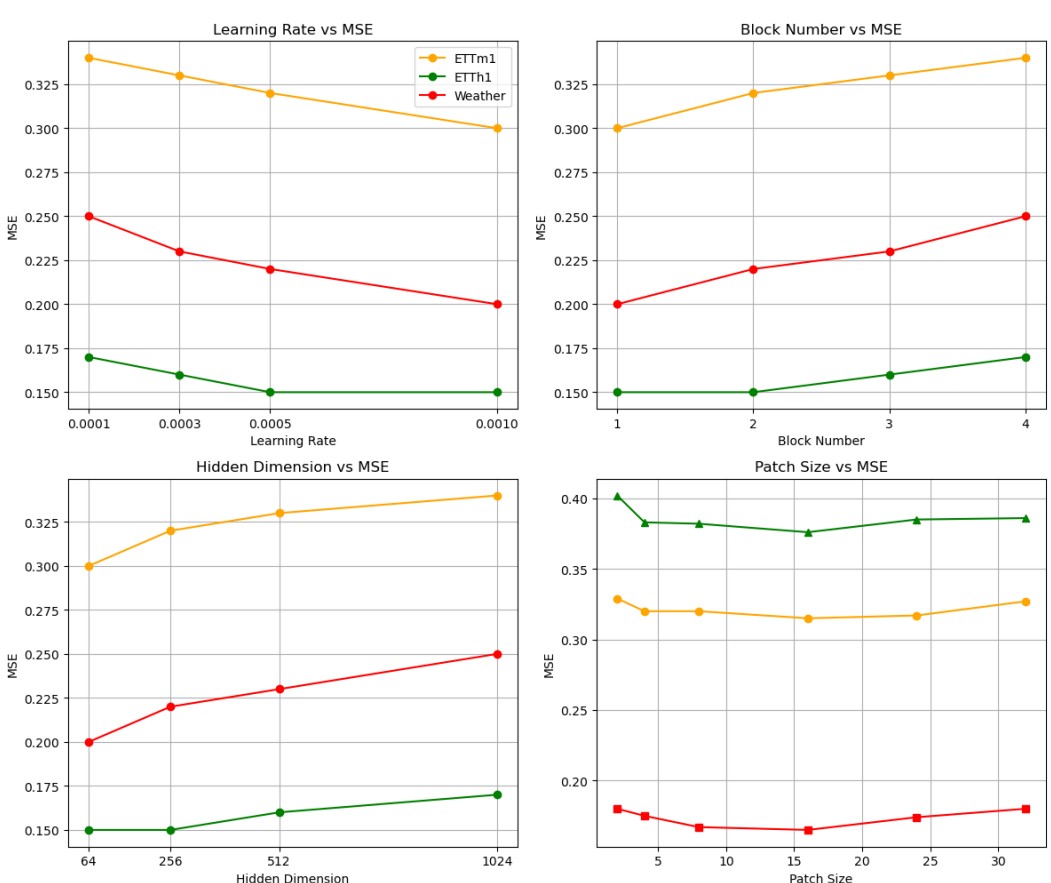

Figure 8: Hyperparameter sensitivity with respect to the hidden dimension, the state expansion factor, the convolutional kernel, and the patch size. The lookback window length $T = 96$ and the forecast window length $S = 96$

longer lookback window to achieve more accurate forecasting? (2) If so, how does its performance compare to other models that can also utilize a longer lookback window?

We conduct experiments to evaluate SAMBA's ability to leverage a longer lookback window. For comparison, we select four baseline models—DLinear, PatchTST, and iTransformer. These models have been proven to utilize extended a lookback window effectively (Liu et al. (2023); Nie et al. (2022)). As shown in Figure 9, we observe a steady decrease in MSE as the lookback window increases, confirming SAMBA's capacity to learn from longer temporal contexts. Moreover, while both SAMBA and the baseline models can effectively take advantage of a longer look-back window, SAMBA consistently demonstrates superior overall performance.

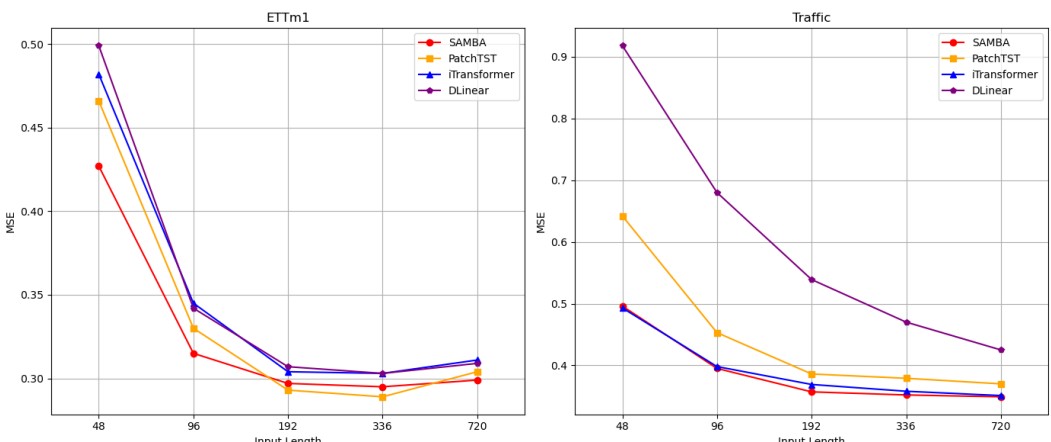

Figure 9: Forecasting results on simulated time series to Verify SAMBA's ability to capture cyclical dependency,

## J  ABLATION STUDY ON SAMBA'S ENCODING COMPONENT AND ITS SUPERIORITY OVER SELF-ATTENTION

To demonstrate that SAMBA's superior performance is not solely due to its disentangled encoding strategy but also because Samba (Simplified Mamba) outperforms the self-attention mechanism, we provide detailed ablations experimental results by replacing the cross-time encoder and cross-variate encoder.

Table 14 shows the experimental results. SAMBA, which utilizes Samba to capture both temporal and cross-variate dependencies, consistently exhibits the best performance across different datasets. For example, replacing the variate encoder with self-attention leads to an increase in the average MSE from 0.378 to 0.390 for ETTm1 and similar performance degradation is observed for other datasets. This highlights the limitations of self-attention in handling temporal dependency and cross-variate dependency and reinforces the effectiveness of SAMBA's architecture.

These findings underscore Samba's superiority in encoding both cross-time and cross-variate dependencies. The performance decline when replacing Samba with self-attention reveals the limitations of self-attention in these contexts. Furthermore, these results are consistent with our earlier observations in Section 4.1 and Appendix C, which demonstrated SAMBA's ability to effectively model order dependency, semantic dependency, and cross-variate dependency, making it particularly well-suited for LTSF applications that require handing of complex temporal and variate dependency.

## K  iTRANSFORMER AND FTP WITH DISENTANGLED ENCODING STRATEGY

We apply our proposed disentangled encoding strategy to both iTransformer and FTP, two SOTA LTSF baselines. The results shown in the table 15 indicate the effectiveness of our strategy on both iTransformer and FTP, demonstrating its universality. Specifically, FTP achieves an approximately

Table 14: Results of ablation study on SAMBA's encoding component. We apply different components on the respective dimension to learn temporal dependency (Temporal) and cross-variate dependency (Variate).

| Design | Temporal | Variate | Prediction Lengths | ETTm1 MSE | MAE | ECL MSE | MAE | Weather MSE | MAE | Solar-Energy MSE | MAE |
|--------|----------|---------|--------------------|-----------|-----|---------|-----|-------------|-----|------------------|-----|
| **SAMBA** | **Samba** | **Samba** | 96 | 0.315 | 0.357 | 0.146 | 0.244 | 0.165 | 0.214 | 0.186 | 0.217 |
| | | | 192 | 0.360 | 0.383 | 0.162 | 0.258 | 0.214 | 0.255 | 0.230 | 0.251 |
| | | | 336 | 0.389 | 0.405 | 0.177 | 0.274 | 0.271 | 0.297 | 0.253 | 0.270 |
| | | | 720 | 0.448 | 0.440 | 0.202 | 0.297 | 0.346 | 0.347 | 0.247 | 0.274 |
| | | | Avg | **0.378** | **0.394** | **0.172** | **0.268** | **0.249** | **0.278** | **0.229** | **0.253** |
| Replace | Samba | Attention | 96 | 0.323 | 0.364 | 0.151 | 0.246 | 0.172 | 0.216 | 0.205 | 0.247 |
| | | | 192 | 0.375 | 0.391 | 0.167 | 0.259 | 0.218 | 0.257 | 0.250 | 0.264 |
| | | | 336 | 0.398 | 0.410 | 0.183 | 0.278 | 0.278 | 0.303 | 0.261 | 0.278 |
| | | | 720 | 0.462 | 0.448 | 0.212 | 0.304 | 0.354 | 0.351 | 0.257 | 0.278 |
| | | | Avg | 0.390 | 0.403 | 0.178 | 0.272 | 0.256 | 0.282 | 0.243 | 0.267 |
| | Attention | Samba | 96 | 0.320 | 0.361 | 0.148 | 0.245 | 0.165 | 0.212 | 0.191 | 0.224 |
| | | | 192 | 0.364 | 0.386 | 0.165 | 0.260 | 0.217 | 0.261 | 0.240 | 0.259 |
| | | | 336 | 0.394 | 0.408 | 0.180 | 0.277 | 0.279 | 0.305 | 0.252 | 0.270 |
| | | | 720 | 0.461 | 0.450 | 0.204 | 0.299 | 0.358 | 0.356 | 0.253 | 0.277 |
| | | | Avg | 0.385 | 0.401 | 0.174 | 0.270 | 0.255 | 0.284 | 0.234 | 0.258 |
| | Attention | Attention | 96 | 0.323 | 0.362 | 0.154 | 0.248 | 0.169 | 0.212 | 0.203 | 0.244 |
| | | | 192 | 0.362 | 0.383 | 0.165 | 0.258 | 0.222 | 0.260 | 0.239 | 0.264 |
| | | | 336 | 0.394 | 0.403 | 0.178 | 0.272 | 0.281 | 0.303 | 0.249 | 0.277 |
| | | | 720 | 0.454 | 0.438 | 0.211 | 0.302 | 0.353 | 0.350 | 0.251 | 0.279 |
| | | | Avg | 0.383 | 0.396 | 0.177 | 0.270 | 0.256 | 0.281 | 0.236 | 0.266 |

| Models | SAMBA MSE | MAE | FTP MSE | MAE | FTP w/D MSE | MAE | iTransformer MSE | MAE | iTransformer w/D MSE | MAE |
|--------|-----------|-----|---------|-----|-------------|-----|------------------|-----|----------------------|-----|
| ECL | **0.146** | 0.244 | 0.190 | 0.273 | 0.173 | 0.260 | 0.150 | 0.242 | 0.148 | **0.239** |
| ETTh1 | **0.376** | 0.400 | 0.380 | 0.396 | 0.377 | **0.395** | 0.391 | 0.407 | 0.385 | 0.404 |
| ETTh2 | **0.288** | **0.340** | 0.305 | 0.354 | 0.304 | 0.453 | 0.299 | 0.349 | 0.294 | 0.345 |
| ETTm1 | **0.315** | **0.357** | 0.335 | 0.367 | 0.320 | 0.361 | 0.345 | 0.379 | 0.344 | 0.375 |
| ETTm2 | **0.172** | 0.259 | 0.176 | 0.262 | 0.174 | 0.258 | 0.184 | 0.270 | 0.179 | 0.265 |
| Traffic | **0.388** | **0.261** | 0.498 | 0.330 | 0.457 | 0.300 | 0.398 | 0.272 | 0.397 | 0.271 |
| Weather | **0.165** | **0.211** | 0.186 | 0.227 | 0.177 | 0.217 | 0.178 | 0.218 | 0.173 | 0.214 |

Table 15: Forecasting of input-96-predict-96 results. Here, the iTransformer results are reproduced on our own, which is a little different from its officially reported results.

Figure 10: Forecasting results on simulated time series to Verify SAMBA's ability to capture cyclical dependency.

7% improvement in datasets with a larger number of variables, such as weather, ECL, and traffic. However, SAMBA still achieves the overall optimal performance, indicating the superiority of the proposed SAMBA block.

## L   HANDLE COMPLEX PERIODIC SIGNAL

The ability to handle complex cyclical signals is an important capability of the model. We create cyclical signals and train SAMBA on them. Figure 10 demonstrates that SAMBA can effectively capture the periodicity of the signals, leading to accurate predictions.

## M  Error Bars

We evaluate the robustness of SAMBA with 5 different sets of random seeds. For each seed, we compute the MSE and MAE scores presented in Table 16. The small variances observed indicate that our model is robust to the choice of random seeds.

Table 16: Robustness of SAMBA performance. The results are obtained from five random seeds.

| Dataset | ETTh1 | | ETTm1 | | Weather | |
|---|---|---|---|---|---|---|
| Horizon | MSE | MAE | MSE | MAE | MSE | MAE |
| 96 | 0.376±0.001 | 0.400±0.002 | 0.315±0.000 | 0.357±0.001 | 0.165±0.003 | 0.211±0.002 |
| 192 | 0.432±0.003 | 0.429±0.001 | 0.360±0.000 | 0.383±0.001 | 0.271±0.002 | 0.297±0.000 |
| 336 | 0.477±0.001 | 0.447±0.002 | 0.389±0.005 | 0.371±0.003 | 0.271±0.007 | 0.297±0.008 |
| 720 | 0.488±0.003 | 0.471±0.002 | 0.448±0.005 | 0.440±0.004 | 0.346±0.010 | 0.347±0.007 |

## N  Full Results

### N.1  Full Results of Evaluating the Universality of Disentangled Encoding

We utilize Transformer and their variants, including Informer (Zhou et al. (2021))and PatchTST (Nie et al. (2022)), to validate the universality of our proposed disentangled encoding strategy. Table 5 shows the average results due to the limited pages. Hence, we provide the full results in Table 17. It demonstrates that our strategy is effective not only for Mamba but also for Transformer-based models. Significant improvements indicate that these models did not properly model and introduce cross-variate dependency, while our strategy effectively alleviates these issues. Therefore, this disentangled encoding strategy is a model-agnostic method and has the potential to be adopted by other models.

### N.2  Full Forecasting Results

In section 6.1, we provide the average forecasting results due to the limited pages. Table 18 shows the full forecasting results. It demonstrates that our proposed SAMBA achieves SOTA performance (e.g., ETT, Traffic, Solar-Energy) or comparable performance (e.g., Weather, Exchange) overall.

Table 17: Full results of Transformer with our proposed disentangled encoding strategy.

| Models | | | Transformer (2017) | | PatchTST (2022) | | Informer (2021) | | |
|---|---|---|---|---|---|---|---|---|---|
| Metric | | | MSE | MAE | MSE | MAE | MSE | MAE | |
| ETTh1 | Original | 96 | 0.863 | 0.721 | 0.414 | 0.419 | 0.924 | 0.719 | |
| | | 192 | 0.883 | 0.750 | 0.460 | 0.445 | 1.051 | 0.786 | |
| | | 336 | 1.081 | 0.855 | 0.501 | 0.466 | 1.132 | 0.827 | |
| | | 720 | 1.163 | 0.861 | 0.500 | 0.488 | 1.132 | 0.832 | |
| | | Avg | 0.997 | 0.797 | 0.469 | 0.454 | 1.060 | 0.791 | |
| | +Disentangled | 96 | 0.393 | 0.415 | 0.376 | 0.394 | 0.395 | 0.412 | |
| | | 192 | 0.459 | 0.464 | 0.449 | 0.437 | 0.463 | 0.465 | |
| | | 336 | 0.507 | 0.493 | 0.486 | 0.452 | 0.518 | 0.502 | |
| | | 720 | 0.568 | 0.537 | 0.488 | 0.471 | 0.578 | 0.559 | |
| | | Avg | **0.481** | **0.477** | **0.450** | **0.439** | **0.489** | **0.485** | |
| ETTm1 | Original | 96 | 0.548 | 0.524 | 0.329 | 0.367 | 0.618 | 0.556 | |
| | | 192 | 0.631 | 0.587 | 0.367 | 0.385 | 0.738 | 0.630 | |
| | | 336 | 0.804 | 0.691 | 0.399 | 0.410 | 1.012 | 0.776 | |
| | | 720 | 1.110 | 0.832 | 0.454 | 0.439 | 1.113 | 0.820 | |
| | | Avg | 0.773 | 0.656 | 0.387 | 0.400 | 0.870 | 0.696 | |
| | +Disentangled | 96 | 0.408 | 0.434 | 0.323 | 0.362 | 0.456 | 0.469 | |
| | | 192 | 0.419 | 0.435 | 0.362 | 0.383 | 0.448 | 0.442 | |
| | | 336 | 0.470 | 0.476 | 0.394 | 0.403 | 0.467 | 0.450 | |
| | | 720 | 0.518 | 0.499 | 0.454 | 0.438 | 0.509 | 0.475 | |
| | | Avg | **0.454** | **0.461** | **0.383** | **0.396** | **0.470** | **0.459** | |
| Weather | Original | 96 | 0.395 | 0.427 | 0.177 | 0.218 | 0.300 | 0.384 | |
| | | 192 | 0.619 | 0.560 | 0.225 | 0.259 | 0.598 | 0.544 | |
| | | 336 | 0.689 | 0.594 | 0.278 | 0.297 | 0.578 | 0.523 | |
| | | 720 | 0.926 | 0.710 | 0.354 | 0.348 | 1.059 | 0.741 | |
| | | Avg | 0.657 | 0.572 | 0.259 | 0.281 | 0.634 | 0.548 | |
| | +Disentangled | 96 | 0.174 | 0.214 | 0.169 | 0.215 | 0.180 | 0.251 | |
| | | 192 | 0.221 | 0.254 | 0.213 | 0.256 | 0.244 | 0.318 | |
| | | 336 | 0.278 | 0.296 | 0.268 | 0.295 | 0.282 | 0.343 | |
| | | 720 | 0.358 | 0.349 | 0.340 | 0.345 | 0.377 | 0.409 | |
| | | Avg | **0.258** | **0.279** | **0.248** | **0.278** | **0.271** | **0.330** | |

Table 18: Full results of the long-term forecasting task. We compare extensive competitive models under different prediction lengths following the setting of TimesNet (2022). The input sequence length is 96 for all baselines. *Avg* means the average results from all four prediction lengths.

| Models | SAMBA Ours | | CARD (2024) | | FTP (2023) | | iTransformer (2023) | | RLinear (2023) | | PatchTST (2022) | | Crossformer (2022) | | TiDE (2023) | | TimesNet (2022) | | DLinear (2022) | | SCINet (2022a) | | FEDformer (2022) | | Stationary (2022b) | | Autoformer (2021) | |
|---|---|---|---|---|---|---|---|---|---|---|---|---|---|---|---|---|---|---|---|---|---|---|---|---|---|---|---|---|
| Metric | MSE | MAE | MSE | MAE | MSE | MAE | MSE | MAE | MSE | MAE | MSE | MAE | MSE | MAE | MSE | MAE | MSE | MAE | MSE | MAE | MSE | MAE | MSE | MAE | MSE | MAE | MSE | MAE |
| ETTm1 96 | 0.315 | 0.357 | 0.351 | 0.379 | 0.335 | 0.367 | 0.334 | 0.368 | 0.355 | 0.376 | 0.329 | 0.367 | 0.404 | 0.426 | 0.364 | 0.387 | 0.338 | 0.375 | 0.345 | 0.372 | 0.418 | 0.438 | 0.379 | 0.419 | 0.386 | 0.398 | 0.505 | 0.475 |
| ETTm1 192 | 0.360 | 0.383 | 0.386 | 0.393 | 0.366 | 0.385 | 0.377 | 0.391 | 0.391 | 0.392 | 0.367 | 0.385 | 0.450 | 0.451 | 0.398 | 0.404 | 0.374 | 0.387 | 0.380 | 0.389 | 0.439 | 0.450 | 0.426 | 0.441 | 0.459 | 0.444 | 0.553 | 0.496 |
| ETTm1 336 | 0.389 | 0.405 | 0.419 | 0.412 | 0.398 | 0.407 | 0.426 | 0.420 | 0.424 | 0.415 | 0.399 | 0.410 | 0.532 | 0.515 | 0.428 | 0.425 | 0.410 | 0.411 | 0.413 | 0.413 | 0.490 | 0.485 | 0.445 | 0.459 | 0.495 | 0.464 | 0.621 | 0.537 |
| ETTm1 720 | 0.448 | 0.440 | 0.479 | 0.445 | 0.461 | 0.443 | 0.491 | 0.459 | 0.487 | 0.450 | 0.454 | 0.439 | 0.666 | 0.589 | 0.487 | 0.461 | 0.478 | 0.450 | 0.474 | 0.453 | 0.595 | 0.550 | 0.543 | 0.490 | 0.585 | 0.516 | 0.671 | 0.561 |
| ETTm1 Avg | 0.378 | 0.394 | 0.409 | 0.407 | 0.392 | 0.401 | 0.407 | 0.410 | 0.414 | 0.407 | 0.387 | 0.400 | 0.513 | 0.496 | 0.419 | 0.419 | 0.400 | 0.406 | 0.403 | 0.407 | 0.485 | 0.481 | 0.448 | 0.452 | 0.481 | 0.456 | 0.588 | 0.517 |
| ETTm2 96 | 0.172 | 0.259 | 0.185 | 0.270 | 0.176 | 0.262 | 0.180 | 0.264 | 0.182 | 0.265 | 0.175 | 0.259 | 0.287 | 0.366 | 0.207 | 0.305 | 0.187 | 0.267 | 0.193 | 0.292 | 0.286 | 0.377 | 0.203 | 0.287 | 0.192 | 0.274 | 0.255 | 0.339 |
| ETTm2 192 | 0.238 | 0.301 | 0.248 | 0.308 | 0.244 | 0.306 | 0.250 | 0.309 | 0.246 | 0.304 | 0.241 | 0.302 | 0.414 | 0.492 | 0.290 | 0.364 | 0.249 | 0.309 | 0.284 | 0.362 | 0.399 | 0.445 | 0.269 | 0.328 | 0.280 | 0.339 | 0.281 | 0.340 |
| ETTm2 336 | 0.300 | 0.340 | 0.309 | 0.348 | 0.309 | 0.348 | 0.311 | 0.348 | 0.307 | 0.342 | 0.305 | 0.343 | 0.597 | 0.542 | 0.377 | 0.422 | 0.321 | 0.351 | 0.369 | 0.427 | 0.637 | 0.591 | 0.325 | 0.366 | 0.334 | 0.361 | 0.339 | 0.372 |
| ETTm2 720 | 0.394 | 0.394 | 0.411 | 0.402 | 0.412 | 0.410 | 0.412 | 0.407 | 0.407 | 0.398 | 0.402 | 0.400 | 1.730 | 1.042 | 0.558 | 0.524 | 0.408 | 0.403 | 0.554 | 0.522 | 0.960 | 0.735 | 0.421 | 0.415 | 0.417 | 0.413 | 0.433 | 0.432 |
| ETTm2 Avg | 0.276 | 0.322 | 0.288 | 0.332 | 0.285 | 0.331 | 0.288 | 0.332 | 0.286 | 0.327 | 0.281 | 0.326 | 0.757 | 0.610 | 0.358 | 0.404 | 0.291 | 0.333 | 0.350 | 0.401 | 0.571 | 0.537 | 0.305 | 0.349 | 0.306 | 0.347 | 0.327 | 0.371 |
| ETTh1 96 | 0.376 | 0.400 | 0.444 | 0.439 | 0.380 | 0.396 | 0.386 | 0.405 | 0.386 | 0.395 | 0.414 | 0.419 | 0.423 | 0.448 | 0.479 | 0.464 | 0.384 | 0.402 | 0.386 | 0.400 | 0.654 | 0.599 | 0.376 | 0.419 | 0.513 | 0.491 | 0.449 | 0.459 |
| ETTh1 192 | 0.432 | 0.429 | 0.501 | 0.471 | 0.432 | 0.427 | 0.441 | 0.436 | 0.437 | 0.424 | 0.460 | 0.445 | 0.471 | 0.474 | 0.525 | 0.492 | 0.436 | 0.429 | 0.437 | 0.432 | 0.719 | 0.631 | 0.420 | 0.448 | 0.534 | 0.504 | 0.500 | 0.482 |
| ETTh1 336 | 0.477 | 0.437 | 0.529 | 0.485 | 0.480 | 0.437 | 0.487 | 0.458 | 0.479 | 0.446 | 0.501 | 0.466 | 0.570 | 0.546 | 0.565 | 0.515 | 0.491 | 0.469 | 0.481 | 0.459 | 0.778 | 0.659 | 0.459 | 0.465 | 0.588 | 0.535 | 0.521 | 0.496 |
| ETTh1 720 | 0.488 | 0.471 | 0.524 | 0.501 | 0.504 | 0.485 | 0.503 | 0.491 | 0.481 | 0.470 | 0.500 | 0.488 | 0.653 | 0.621 | 0.594 | 0.558 | 0.521 | 0.500 | 0.519 | 0.516 | 0.836 | 0.699 | 0.506 | 0.507 | 0.643 | 0.616 | 0.514 | 0.512 |
| ETTh1 Avg | 0.443 | 0.432 | 0.500 | 0.474 | 0.450 | 0.439 | 0.454 | 0.447 | 0.446 | 0.434 | 0.469 | 0.454 | 0.529 | 0.522 | 0.541 | 0.507 | 0.458 | 0.450 | 0.456 | 0.452 | 0.747 | 0.647 | 0.440 | 0.460 | 0.570 | 0.537 | 0.496 | 0.487 |
| ETTh2 96 | 0.288 | 0.340 | 0.318 | 0.361 | 0.305 | 0.354 | 0.297 | 0.349 | 0.288 | 0.338 | 0.302 | 0.348 | 0.745 | 0.584 | 0.400 | 0.440 | 0.340 | 0.374 | 0.333 | 0.387 | 0.707 | 0.621 | 0.358 | 0.397 | 0.476 | 0.458 | 0.346 | 0.388 |
| ETTh2 192 | 0.373 | 0.390 | 0.399 | 0.409 | 0.389 | 0.408 | 0.380 | 0.400 | 0.374 | 0.390 | 0.388 | 0.400 | 0.877 | 0.656 | 0.528 | 0.509 | 0.402 | 0.414 | 0.477 | 0.476 | 0.860 | 0.689 | 0.429 | 0.439 | 0.512 | 0.493 | 0.456 | 0.452 |
| ETTh2 336 | 0.380 | 0.406 | 0.435 | 0.441 | 0.415 | 0.432 | 0.428 | 0.432 | 0.415 | 0.426 | 0.426 | 0.433 | 1.043 | 0.731 | 0.643 | 0.571 | 0.452 | 0.452 | 0.594 | 0.541 | 1.000 | 0.744 | 0.496 | 0.487 | 0.552 | 0.551 | 0.482 | 0.486 |
| ETTh2 720 | 0.412 | 0.432 | 0.438 | 0.451 | 0.432 | 0.451 | 0.427 | 0.445 | 0.420 | 0.440 | 0.431 | 0.446 | 1.104 | 0.763 | 0.874 | 0.679 | 0.462 | 0.468 | 0.831 | 0.657 | 1.249 | 0.838 | 0.463 | 0.474 | 0.562 | 0.560 | 0.515 | 0.511 |
| ETTh2 Avg | 0.363 | 0.392 | 0.398 | 0.416 | 0.385 | 0.411 | 0.383 | 0.407 | 0.374 | 0.398 | 0.387 | 0.407 | 0.942 | 0.684 | 0.611 | 0.550 | 0.414 | 0.427 | 0.559 | 0.515 | 0.954 | 0.723 | 0.437 | 0.449 | 0.526 | 0.516 | 0.450 | 0.459 |
| ECL 96 | 0.146 | 0.244 | 0.181 | 0.271 | 0.190 | 0.273 | 0.148 | 0.240 | 0.201 | 0.281 | 0.181 | 0.270 | 0.219 | 0.314 | 0.237 | 0.329 | 0.168 | 0.272 | 0.197 | 0.282 | 0.247 | 0.345 | 0.193 | 0.308 | 0.169 | 0.273 | 0.201 | 0.317 |
| ECL 192 | 0.162 | 0.258 | 0.188 | 0.277 | 0.191 | 0.274 | 0.162 | 0.253 | 0.201 | 0.283 | 0.188 | 0.274 | 0.231 | 0.322 | 0.236 | 0.330 | 0.184 | 0.289 | 0.196 | 0.285 | 0.257 | 0.355 | 0.201 | 0.315 | 0.182 | 0.286 | 0.222 | 0.334 |
| ECL 336 | 0.177 | 0.274 | 0.204 | 0.292 | 0.208 | 0.294 | 0.178 | 0.269 | 0.215 | 0.298 | 0.204 | 0.293 | 0.246 | 0.337 | 0.249 | 0.344 | 0.198 | 0.300 | 0.209 | 0.301 | 0.269 | 0.369 | 0.214 | 0.329 | 0.200 | 0.304 | 0.231 | 0.338 |
| ECL 720 | 0.202 | 0.297 | 0.244 | 0.322 | 0.251 | 0.326 | 0.225 | 0.317 | 0.257 | 0.331 | 0.246 | 0.324 | 0.280 | 0.363 | 0.284 | 0.373 | 0.220 | 0.320 | 0.245 | 0.333 | 0.299 | 0.390 | 0.246 | 0.355 | 0.222 | 0.321 | 0.254 | 0.361 |
| ECL Avg | 0.172 | 0.268 | 0.204 | 0.291 | 0.210 | 0.291 | 0.178 | 0.270 | 0.219 | 0.298 | 0.205 | 0.290 | 0.244 | 0.334 | 0.251 | 0.344 | 0.192 | 0.295 | 0.212 | 0.300 | 0.268 | 0.365 | 0.214 | 0.327 | 0.193 | 0.296 | 0.227 | 0.338 |
| Exchange 96 | 0.083 | 0.202 | 0.086 | 0.205 | 0.084 | 0.201 | 0.086 | 0.206 | 0.093 | 0.217 | 0.088 | 0.205 | 0.256 | 0.367 | 0.094 | 0.218 | 0.107 | 0.234 | 0.088 | 0.218 | 0.267 | 0.396 | 0.148 | 0.278 | 0.111 | 0.237 | 0.197 | 0.323 |
| Exchange 192 | 0.176 | 0.298 | 0.182 | 0.303 | 0.174 | 0.295 | 0.177 | 0.299 | 0.184 | 0.307 | 0.176 | 0.299 | 0.470 | 0.509 | 0.184 | 0.307 | 0.226 | 0.344 | 0.176 | 0.315 | 0.351 | 0.459 | 0.271 | 0.315 | 0.219 | 0.335 | 0.300 | 0.369 |
| Exchange 336 | 0.327 | 0.413 | 0.342 | 0.432 | 0.349 | 0.427 | 0.331 | 0.417 | 0.351 | 0.432 | 0.301 | 0.397 | 1.268 | 0.883 | 0.349 | 0.431 | 0.367 | 0.448 | 0.313 | 0.427 | 1.324 | 0.853 | 0.460 | 0.427 | 0.421 | 0.476 | 0.509 | 0.524 |
| Exchange 720 | 0.839 | 0.689 | 0.970 | 0.745 | 0.864 | 0.700 | 0.847 | 0.691 | 0.886 | 0.714 | 0.901 | 0.714 | 1.767 | 1.068 | 0.852 | 0.698 | 0.964 | 0.746 | 0.839 | 0.695 | 1.058 | 0.797 | 1.195 | 0.695 | 1.092 | 0.769 | 1.447 | 0.941 |
| Exchange Avg | 0.356 | 0.401 | 0.395 | 0.421 | 0.368 | 0.406 | 0.360 | 0.403 | 0.378 | 0.417 | 0.367 | 0.404 | 0.940 | 0.707 | 0.370 | 0.413 | 0.416 | 0.443 | 0.354 | 0.414 | 0.750 | 0.626 | 0.519 | 0.429 | 0.461 | 0.454 | 0.613 | 0.539 |
| Traffic 96 | 0.388 | 0.261 | 0.455 | 0.313 | 0.498 | 0.330 | 0.395 | 0.268 | 0.649 | 0.389 | 0.462 | 0.295 | 0.522 | 0.290 | 0.805 | 0.493 | 0.593 | 0.321 | 0.650 | 0.396 | 0.788 | 0.499 | 0.587 | 0.366 | 0.612 | 0.338 | 0.613 | 0.388 |
| Traffic 192 | 0.411 | 0.271 | 0.469 | 0.315 | 0.500 | 0.332 | 0.417 | 0.276 | 0.601 | 0.366 | 0.466 | 0.296 | 0.530 | 0.293 | 0.756 | 0.474 | 0.617 | 0.336 | 0.598 | 0.370 | 0.789 | 0.505 | 0.604 | 0.373 | 0.613 | 0.340 | 0.616 | 0.382 |
| Traffic 336 | 0.428 | 0.278 | 0.482 | 0.319 | 0.512 | 0.332 | 0.433 | 0.283 | 0.609 | 0.369 | 0.482 | 0.304 | 0.558 | 0.305 | 0.762 | 0.477 | 0.629 | 0.336 | 0.605 | 0.373 | 0.797 | 0.508 | 0.621 | 0.383 | 0.618 | 0.328 | 0.622 | 0.337 |
| Traffic 720 | 0.461 | 0.297 | 0.518 | 0.338 | 0.534 | 0.342 | 0.467 | 0.302 | 0.647 | 0.387 | 0.514 | 0.322 | 0.589 | 0.328 | 0.719 | 0.449 | 0.640 | 0.350 | 0.645 | 0.394 | 0.841 | 0.523 | 0.626 | 0.382 | 0.653 | 0.355 | 0.660 | 0.408 |
| Traffic Avg | 0.422 | 0.276 | 0.481 | 0.321 | 0.511 | 0.334 | 0.428 | 0.282 | 0.626 | 0.378 | 0.481 | 0.304 | 0.550 | 0.304 | 0.760 | 0.473 | 0.620 | 0.336 | 0.625 | 0.383 | 0.804 | 0.509 | 0.610 | 0.376 | 0.624 | 0.340 | 0.628 | 0.379 |
| Weather 96 | 0.165 | 0.214 | 0.164 | 0.212 | 0.186 | 0.227 | 0.174 | 0.214 | 0.192 | 0.232 | 0.177 | 0.218 | 0.158 | 0.230 | 0.202 | 0.261 | 0.172 | 0.220 | 0.196 | 0.255 | 0.221 | 0.306 | 0.217 | 0.296 | 0.173 | 0.223 | 0.266 | 0.336 |
| Weather 192 | 0.214 | 0.255 | 0.212 | 0.253 | 0.232 | 0.264 | 0.221 | 0.254 | 0.240 | 0.271 | 0.225 | 0.259 | 0.206 | 0.277 | 0.242 | 0.298 | 0.219 | 0.261 | 0.237 | 0.296 | 0.261 | 0.340 | 0.276 | 0.336 | 0.245 | 0.285 | 0.307 | 0.367 |
| Weather 336 | 0.271 | 0.297 | 0.269 | 0.294 | 0.288 | 0.304 | 0.278 | 0.296 | 0.292 | 0.307 | 0.278 | 0.297 | 0.272 | 0.335 | 0.287 | 0.335 | 0.280 | 0.306 | 0.283 | 0.335 | 0.309 | 0.378 | 0.339 | 0.380 | 0.321 | 0.338 | 0.359 | 0.395 |
| Weather 720 | 0.346 | 0.347 | 0.349 | 0.346 | 0.361 | 0.351 | 0.358 | 0.347 | 0.364 | 0.353 | 0.354 | 0.348 | 0.398 | 0.418 | 0.351 | 0.386 | 0.365 | 0.359 | 0.345 | 0.381 | 0.377 | 0.427 | 0.403 | 0.428 | 0.414 | 0.410 | 0.419 | 0.428 |
| Weather Avg | 0.249 | 0.278 | 0.249 | 0.276 | 0.267 | 0.287 | 0.258 | 0.278 | 0.272 | 0.291 | 0.259 | 0.281 | 0.259 | 0.315 | 0.271 | 0.320 | 0.259 | 0.287 | 0.265 | 0.317 | 0.292 | 0.363 | 0.309 | 0.360 | 0.288 | 0.314 | 0.338 | 0.382 |
| Solar-Energy 96 | 0.186 | 0.217 | 0.210 | 0.249 | 0.244 | 0.293 | 0.203 | 0.237 | 0.322 | 0.339 | 0.234 | 0.286 | 0.310 | 0.331 | 0.312 | 0.399 | 0.250 | 0.292 | 0.290 | 0.378 | 0.237 | 0.344 | 0.242 | 0.342 | 0.215 | 0.249 | 0.884 | 0.711 |
| Solar-Energy 192 | 0.230 | 0.251 | 0.242 | 0.274 | 0.265 | 0.298 | 0.233 | 0.261 | 0.359 | 0.356 | 0.267 | 0.310 | 0.734 | 0.725 | 0.339 | 0.416 | 0.296 | 0.318 | 0.320 | 0.398 | 0.280 | 0.380 | 0.285 | 0.380 | 0.254 | 0.272 | 0.834 | 0.692 |
| Solar-Energy 336 | 0.253 | 0.270 | 0.260 | 0.287 | 0.284 | 0.312 | 0.248 | 0.273 | 0.397 | 0.369 | 0.290 | 0.315 | 0.750 | 0.735 | 0.368 | 0.430 | 0.319 | 0.330 | 0.353 | 0.415 | 0.304 | 0.389 | 0.282 | 0.376 | 0.290 | 0.296 | 0.941 | 0.723 |
| Solar-Energy 720 | 0.247 | 0.274 | 0.268 | 0.296 | 0.281 | 0.312 | 0.249 | 0.275 | 0.397 | 0.356 | 0.289 | 0.317 | 0.769 | 0.765 | 0.370 | 0.425 | 0.338 | 0.337 | 0.356 | 0.413 | 0.308 | 0.388 | 0.357 | 0.427 | 0.285 | 0.295 | 0.882 | 0.717 |
| Solar-Energy Avg | 0.229 | 0.253 | 0.245 | 0.277 | 0.269 | 0.304 | 0.233 | 0.262 | 0.369 | 0.356 | 0.270 | 0.307 | 0.641 | 0.639 | 0.347 | 0.417 | 0.301 | 0.319 | 0.330 | 0.401 | 0.282 | 0.375 | 0.291 | 0.381 | 0.261 | 0.381 | 0.885 | 0.711 |
| 1st Count | 34 | 29 | 4 | 5 | 1 | 3 | 2 | 3 | 2 | 5 | 1 | 3 | 0 | 0 | 0 | 0 | 0 | 0 | 2 | 0 | 0 | 0 | 4 | 0 | 0 | 0 | 0 | 0 |

Table 19: Results of the linear model, Mamba, and Transformer w/ or w/o patching on ETTm1 dataset.

| Models | Linear Model | | | | Mamba | | | | Transformer | | | |
|---|---|---|---|---|---|---|---|---|---|---|---|---|
| | w/o patching | | w/ patching | | w/o patching | | w/ patching | | w/o patching | | w/ patching | |
| Metrics | MSE | MAE | MSE | MAE | MSE | MAE | MSE | MAE | MSE | MAE | MSE | MAE |
| 96 | 0.383 | 0.400 | 0.366 | 0.388 | 0.517 | 0.508 | 0.341 | 0.377 | 0.643 | 0.575 | 0.364 | 0.394 |
| 192 | 0.413 | 0.415 | 0.400 | 0.404 | 0.575 | 0.546 | 0.378 | 0.399 | 0.805 | 0.664 | 0.394 | 0.404 |
| 336 | 0.441 | 0.435 | 0.429 | 0.425 | 0.730 | 0.634 | 0.413 | 0.421 | 0.882 | 0.737 | 0.429 | 0.430 |
| 720 | 0.497 | 0.469 | 0.485 | 0.460 | 0.873 | 0.704 | 0.474 | 0.465 | 0.928 | 0.752 | 0.468 | 0.4600 |
| Avg. | 0.434 | 0.469 | **0.420** | **0.419** | 0.674 | 0.598 | **0.402** | **0.416** | 0.815 | 0.682 | **0.414** | **0.422** |

# O    DISCUSSION ON THE DISENTANGLED ENCODING STRATEGY

We use $x$ layers of SAMBA and $y$ layers of bi-SAMBA (where $x$ and $y$ can be different values) to separately encode temporal dependencies and cross-variable dependencies.

Then, the aggregation formula can be expressed mathematically as follows:

$$\mathbf{E}_o = \text{FFN}(\mathbf{E}_{\text{time}} \| \mathbf{E}_{\text{var}})$$

FFN is a fully connected feed-forward neural network, which can be expressed as:

$$\text{FFN}(\mathbf{x}) = \mathbf{W}_2 \sigma(\mathbf{W}_1 \mathbf{x} + \mathbf{b}_1) + \mathbf{b}_2$$

where $\mathbf{x} = \mathbf{E}_{\text{time}} \| \mathbf{E}_{\text{var}}$ represents the concatenated input vector, $\mathbf{W}_1, \mathbf{W}_2$ are weight matrices, $\mathbf{b}_1, \mathbf{b}_2$ are bias vectors, $\sigma$ is a non-linear activation function (e.g., ReLU).

Since backpropagation involves the derivatives of the output with respect to the inputs, we analyze the derivatives of $\mathbf{E}_o$ with respect to $\mathbf{E}_{\text{time}}$ and $\mathbf{E}_{\text{var}}$.

**Derivative with Respect to $\mathbf{E}_{\text{time}}$**

For the concatenated input:

$$\mathbf{x} = \begin{bmatrix} \mathbf{E}_{\text{time}} \\ \mathbf{E}_{\text{var}} \end{bmatrix}$$

The derivative of $\mathbf{E}_o$ with respect to $\mathbf{E}_{\text{time}}$ is:

$$\frac{\partial \mathbf{E}_o}{\partial \mathbf{E}_{\text{time}}} = \frac{\partial \mathbf{E}_o}{\partial \mathbf{x}} \cdot \frac{\partial \mathbf{x}}{\partial \mathbf{E}_{\text{time}}}$$

Since $\mathbf{x}$ contains $\mathbf{E}_{\text{time}}$ as its first part:

$$\frac{\partial \mathbf{x}}{\partial \mathbf{E}_{\text{time}}} = \mathbf{I}$$

where $\mathbf{I}$ is the identity matrix.

The derivative of the FFN with respect to the input $\mathbf{x}$ is:

$$\frac{\partial \mathbf{E}_o}{\partial \mathbf{x}} = \mathbf{W}_2 \cdot \text{diag}(\sigma'(\mathbf{W}_1 \mathbf{x} + \mathbf{b}_1)) \cdot \mathbf{W}_1$$

Thus:

$$\frac{\partial \mathbf{E}_o}{\partial \mathbf{E}_{\text{time}}} = \mathbf{W}_2 \cdot \text{diag}(\sigma'(\mathbf{W}_1 \mathbf{x} + \mathbf{b}_1)) \cdot \mathbf{W}_1 \cdot \mathbf{P}_{\text{time}}$$

where $\mathbf{P}_{\text{time}}$ is the projection matrix that selects the $\mathbf{E}_{\text{time}}$ part from the concatenated vector.

**Derivative with Respect to $\mathbf{E}_{\text{var}}$**

Similarly, the derivative with respect to $\mathbf{E}_{\text{var}}$ is:

$$\frac{\partial \mathbf{E}_o}{\partial \mathbf{E}_{\text{var}}} = \mathbf{W}_2 \cdot \text{diag}(\sigma'(\mathbf{W}_1 \mathbf{x} + \mathbf{b}_1)) \cdot \mathbf{W}_1 \cdot \mathbf{P}_{\text{var}}$$

where $\mathbf{P}_{\text{var}}$ is the projection matrix that selects the $\mathbf{E}_{\text{var}}$ part from the concatenated vector.

**Summary**

$$\frac{\partial \mathbf{E}_o}{\partial \mathbf{E}_{\text{time}}} = \mathbf{W}_2 \cdot \text{diag}(\sigma'(\mathbf{W}_1 \mathbf{x} + \mathbf{b}_1)) \cdot \mathbf{W}_1 \cdot \mathbf{P}_{\text{time}}$$

$$\frac{\partial \mathbf{E}_o}{\partial \mathbf{E}_{\text{var}}} = \mathbf{W}_2 \cdot \text{diag}(\sigma'(\mathbf{W}_1 \mathbf{x} + \mathbf{b}_1)) \cdot \mathbf{W}_1 \cdot \mathbf{P}_{\text{var}}$$

We can observe that the derivative of $\mathbf{E}_o$ with respect to $\mathbf{E}_{\text{time}}$ involves only $\mathbf{P}_{\text{time}}$, while the derivative with respect to $\mathbf{E}_{\text{var}}$ involves only $\mathbf{P}_{\text{var}}$. Therefore, even during backpropagation, the aggregation operation avoids any entanglement.

# P   FURTHER DISCUSSION BETWEEN ORDER AND SEMANTIC DEPENDENCIES

In this section, we provide a detailed discussion on the differences and connections between order and semantic dependencies. As previously mentioned, order dependency is tied to the direct input sequence and emphasizes local temporal ordering relationships, while semantic dependency abstracts over these sequences to uncover more global or stable latent patterns in time series data. Below is a tangible example illustrating order and semantic dependencies.

*Example 1.*        Suppose we have a sequence of electricity consumption values, $[2, 8, 10, 5, 2, 2, 7, 12, 8, 2]$, exhibiting a clear daily periodic pattern: electricity consumption increases in the morning, peaks during midday, and decreases at night. When considering order dependency, the model observing $[2, 8]$ might predict $10$ as the next value, assuming the upward trend continues. However, if we examine another part of the sequence, $[2, 8, 10, 5, 2, 2, 7, 12]$, order dependency might incorrectly predict the trend will continue upward, while semantic dependency captures the cyclic nature of the data and recognizes this as part of the descending phase of the cycle, predicting that the sequence will likely decrease to $8$ or a lower value.

Beyond periodicity, semantic dependency can also be explained through shape patterns or structural motifs in a time series. Specifically, semantic dependency identifies higher-level, invariant patterns in the data that enable the inference of the sequence's full structure from partial observations, transcending the explicit order of individual values. The following example provides an intuitive illustration.

*Example 2.* Consider a time series, $[0.1, 0.3, 1.0, 0.5, 0.2, 0.1]$, representing the electrical activity of a heartbeat (ECG signal). A typical heartbeat exhibits a distinctive shape: it begins with a small upward bump (P wave), followed by a sharp spike (QRS complex), and ends with a smaller bump (T wave). Semantic dependency enables the model to interpret the observed segment $[0.1, 0.3, 1.0]$ as part of this larger heartbeat structure. The model recognizes the segment as an incomplete heartbeat signal and predicts that it will likely be followed by $[0.5, 0.2, 0.1]$ to complete the full pattern.

Finally, in our formulation, semantic dependency is designed to focus on *intra-variable patterns*, *i.e.*, patterns within a single variable's time series. Interactions between different variables are captured through cross-variate dependency. Such a definition considers the intrinsic properties of each variable independently, enabling a more effective analysis of their distinct contributions.

# Q   DISCUSSION ON THE MAMBA AND ITS ADVANTAGE OF MAMBA OVER TRANSFORMER IN LTSF

## Q.1   MAMBA: FROM CLASSICAL THEORY TO MODERN INNOVATIONS

State Space Model (SSM) originates from control theory and is a mathematical tool used to describe the dynamics of systems. It has been widely applied in computational neuroscience, signal processing, and engineering control. The core idea is to use the state vector to represent the internal dynamics of a system and model the relationship between input and output through input vectors and output matrices.

The continuous-time dynamic system of SSM is represented as:

$$\dot{\boldsymbol{h}}(t) = \mathbf{A}(t)\boldsymbol{h}(t) + \mathbf{B}(t)\mathbf{x}(t),$$

$$\mathbf{y}(t) = \mathbf{C}(t)\boldsymbol{h}(t) + \mathbf{D}(t)\mathbf{U}(t),$$

where $\boldsymbol{h}(t)$ is the state vector, representing the internal state of the system at time $t$, $\dot{\boldsymbol{h}}(t) = \frac{d}{dt}\boldsymbol{h}(t)$, $\boldsymbol{x}(t)$ is the input signal, $\mathbf{y}(t)$ is the output signal, $\mathbf{A}(t), \mathbf{B}(t), \mathbf{C}(t), \mathbf{D}(t)$ are the state matrix, input matrix, output matrix, and direct feedthrough matrix, respectively.

In many practical systems, the direct feedthrough matrix $D(t)$ is often zero, meaning the input signal does not directly influence the output but instead affects the output indirectly through the state vector. This assumption simplifies the equations to:

$$\dot{h}(t) = \mathbf{A}(t)h(t) + \mathbf{B}(t)x(t),$$

$$\mathbf{y}(t) = \mathbf{C}(t)h(t).$$

This simplification makes the equations more concise and easier to work with, providing a strong foundation for further mathematical analysis and engineering applications.

Modern deep learning and computational frameworks are built around discrete-time data, while the original SSM is a continuous-time model. To make it compatible with these frameworks, discretization is required to convert continuous-time equations into discrete-time ones while preserving their dynamic properties.

The discretized SSM equations are:

$$h_t = \bar{\mathbf{A}}h_{t-1} + \bar{\mathbf{B}}x_t,$$

$$y_t = \mathbf{C}x_t,$$

where $\overline{\mathbf{A}} \in \mathbb{R}^{N \times N}$, and $\overline{\mathbf{B}}, \mathbf{C} \in \mathbb{R}^{N \times D}$ are learnable parameters that map input sequence $x_t \in \mathbb{R}^D$ to output sequence $y_t \in \mathbb{R}^D$ through an hidden state $h_t \in \mathbb{R}^N$. In particular, $\overline{\mathbf{A}}$ and $\overline{\mathbf{B}}$ are the discretized forms of $\mathbf{A}$ and $\mathbf{B}$ using $\mathbf{\Delta}$ for seamless intergration deep learning. They are computed using the Zero-Order Hold (ZOH) method:

$$\overline{\mathbf{A}} = \exp\left(\Delta\mathbf{A}\right), \ \overline{\mathbf{B}} = (\mathbf{\Delta}\mathbf{A})^{-1}(\exp\left(\mathbf{\Delta}\mathbf{A}\right) - \mathbf{I}) \cdot \mathbf{\Delta}\mathbf{B}.$$

The discretization provides the following advantages:

- **Compatibility with discrete data:** The discretized equations align with how computers handle sequential data.
- **Parallel computation:** Discretized models can utilize matrix operations and convolutional techniques for parallel processing, significantly improving computational efficiency.
- **Long-sequence modeling:** The discrete model retains the ability to model dynamic systems effectively, even in long-sequence tasks.

Through discretization, SSM adapts to modern computing requirements and becomes a viable tool for deep learning applications.

While Transformer models excel in sequence modeling tasks, their global attention mechanism incurs $O(N^2)$ computational complexity, making it computationally expensive for long sequences. Modern SSM addresses this limitation through the following advancements:

- **Parallelized computation:** To overcome the challenge of parallelizing computations in recursive models, modern SSMs achieve parallelization by reformulating recursive equations into convolutional forms:

$$\mathbf{y} = \mathbf{x} * \mathbf{K},$$

where the convolution kernel $\mathbf{K}$ represents the dynamics over multiple time steps:

$$\mathbf{K} = (\mathbf{C}\bar{\mathbf{B}}, \mathbf{C}\bar{\mathbf{A}}\bar{\mathbf{B}}, \mathbf{C}\bar{\mathbf{A}}^2\bar{\mathbf{B}}, \dots).$$

- **Hardware-friendly optimization:** Models like S4 and S4nd leverage hardware-aware designs, combining parallel scanning and kernel fusion for efficiency.
- **Learnable parameterization:** Modern SSM incorporates learnable parameters for $\bar{\mathbf{A}}, \bar{\mathbf{B}}, \mathbf{C}$, enabling better adaptation to data during training.

However, a significant challenge must be addressed for SSMs to become a strong alternative to Transformers. In the Transformer architecture, contextual information is stored in the attention matrix. In contrast, SSMs lack a similar mechanism, making them less effective in learning context. The emergence of Mamba offers a solution to this critical issue. Mamba introduces the following innovations to tackle this problem:

- **Dynamic selection mechanism:** Mamba introduces data-dependent mechanisms for adjusting input and output matrices and step size dynamically:

$$\mathbf{B} = \text{Linear}_N(\mathbf{x}_t), \quad \mathbf{C} = \text{Linear}_N(\mathbf{x}_t), \quad \Delta = \text{softplus}(\text{Linear}_N(\mathbf{x}_t)).$$

- **Multi-branch structure:** Each Mamba layer includes two branches:
  - A *Selective SSM branch* to capture sequential and semantic dependencies.

  $$\boldsymbol{x}_t' = \text{SelectiveSSM}(\sigma(\text{Conv1D}(\text{Linear}(\boldsymbol{x}_t)))).$$

  - A residual branch to enhance stability and mitigate overfitting using gating mechanisms.

  $$\boldsymbol{y} = \text{LayerNorm}(\boldsymbol{x}_t' \otimes (\sigma(\text{Linear}(\boldsymbol{x}_t))) + \boldsymbol{x}_t).$$

- **Hardware efficiency:** Mamba integrates Hardware-aware Algorithm that allows efficient storage of (intermediate) results through parallel scanning, kernel fusion, and recalculation, making it well-suited for long-sequence tasks.

State Space Model has evolved from a control theory tool to a robust framework for deep learning tasks. Its simplifications and discretization adapt it to modern computing needs, while recent advancements like Mamba extend its capabilities for long-sequence modeling, addressing the computational challenges of Transformer models. These developments establish SSM as a powerful and efficient tool for dynamic system representation in deep learning.

## Q.2    THE ADVANTAGE OF MAMBA OVER TRANSFORMER OVER LTSF

Mamba, based on modern State Space Models (SSMs), offers significant advantages over Transformer models in time series forecasting. This document elaborates on three key aspects that highlight Mamba's superior performance: (i) its ability to inherently capture order dependency through SSMs, (ii) a selection mechanism that enables semantic dependency modeling, and (iii) near-linear complexity for efficient cross-variate dependency encoding.

Mamba leverages SSMs to process sequential data recursively, inherently capturing the order dependency in time series:

- **Recursive Dynamics:** The core of SSM lies in its recursive computation mechanism. Using the state update formula:
$$\mathbf{X}_t = \bar{\mathbf{A}}\mathbf{X}_{t-1} + \bar{\mathbf{B}}\mathbf{U}_t,$$
Mamba updates the state vector step by step, effectively modeling the temporal relationships between different time steps. This is particularly well-suited for capturing the dynamic nature of time series data.

- **Comparison with Transformer:** Transformers rely on self-attention mechanisms to capture global dependencies but lack inherent order modeling capabilities. Positional encodings must be added to compensate for this limitation. In contrast, Mamba's SSM-based recursive framework naturally incorporates order information, making it better equipped for time series forecasting tasks that require long-term dependency modeling.

Mamba incorporates a selection mechanism, similar to self-attention, which allows it to dynamically focus on or ignore specific inputs, thereby improving semantic dependency learning:

- **Mechanism Details:**

- – Mamba dynamically adjusts input weights through linear transformations (e.g., $\mathbf{B} = \text{Linear}_N(\mathbf{x}_t)$) and nonlinear activation functions such as Softplus.
  - – This enables the model to assign different levels of importance to input signals at different time steps, filtering out irrelevant features while retaining significant ones.
- **Semantic Modeling:** In time series forecasting, inputs at different time steps can have varying levels of relevance. Mamba's selection mechanism efficiently learns these semantic dependencies, improving its understanding of complex temporal patterns.

Mamba's near-linear complexity offers a significant efficiency advantage in encoding cross-variate dependencies, which are typically computationally expensive in large-scale time series:

- **Efficient Cross-Variate Modeling:**
  - – Time series forecasting often involves modeling interactions among multiple variables. Mamba achieves this efficiently using the recursive structure of SSMs, where the state update matrix $\bar{\mathbf{A}}$ and input matrix $\bar{\mathbf{B}}$ encode these dependencies.
  - – Unlike Transformers, Mamba does not require explicit computation of all pairwise interactions, reducing computational overhead.
- **Comparison with Transformer:**
  - – Transformer's self-attention mechanism computes interactions across all input dimensions, leading to $O(N^2)$ complexity. This becomes a bottleneck for large-scale time series with many variables.
  - – Mamba's near-linear complexity enables it to handle large amounts of variates efficiently while maintaining competitive performance in cross-variate dependency modeling.

Mamba outperforms Transformer in time series forecasting through three key advantages:

1. **Recursive Processing:** SSMs inherently capture order dependency, making Mamba well-suited for LTSF.
2. **Selection Mechanism:** Similar to self-attention, this mechanism enables dynamic focus on relevant inputs, enhancing semantic understanding.
3. **Near-Linear Complexity:** Mamba efficiently models cross-variate dependencies, offering a computational advantage in handling large-scale time series.

## R  DISCUSSION ON THE DIFFERENCE BETWEEN ORDER AND SEMANTIC DEPENDENCY WITH SEASONAL AND TREND PATTERNS

Order dependency describes the sequential link between consecutive observations in a time series, where each data point is affected by its preceding value(s). On the other hand, semantic dependency refers to the underlying factors that influence time series behavior over extended periods, like high-level semantic in the field of CV and NLP.

In past time series forecasting, researchers consider time seires data from the perspective of seasonal and trend components. However, this view can be considered as a specific example of our proposed dependency view. Trend information in particular represents a type of order dependency, illustrating the sequential progression and causal relationships inherent in time series data. For example, a gradual increase in temperature over time shows how current values are influenced by past observations, establishing a long-term directional trend in the data.

Seasonal patterns can be regards as a type of semantic dependency, encapsulating more abstract and hidden relationships that extend across non-consecutive intervals. For example, repeated cycles

in electricity consumption during the summer are influenced by external latent factors such as temperature and user behavior. These factors are not directly correlated with adjacent time points, but indicate deeper contextual impacts. Although models like RLinear that capture order dependency effectively have demonstrated that straightforward linear models can competently capture cyclical signals (Li et al. (2023)), this is mainly because these strict cyclical patterns are a kind of linear semantic. But the core of model to forecast is to utilize the periodic semantics.

## S  DISCUSSION ON LINEAR MODEL ORDER DEPENDENCY

Regarding the statement "linear models can effectively model order dependency," it is because they inherently preserve the temporal order by linearly regressing historical data points to forecast future values. If the temporal order of the input data points is altered, the trained linear model will perform poorly, which we refer to as'sensitivity.' This argument has been empirically demonstrated in Zeng et al. (2023).

## T  DISCUSSION ON IMPACT OF PROPOSED METHOD

**The impact of real-life applications**    Our model can effectively provide accurate predictions for long-term time series, which is pivotal for making informed decisions and exerts multifaceted impacts on society:

- **Economic Planning**: Governments and enterprises can utilize long-term forecasting outcomes for more efficacious economic strategizing and budget allocation, thereby fostering economic growth and stability.
- **Resource Management**: Long-term time series forecasting aids in the sustainable stewardship of resources, such as in the realms of energy, agriculture, and environmental conservation, where future demands and supplies can be anticipated to prevent resource depletion. As demonstrated in our ETT and ECL dataset, accurate predictions of electricity-related data can assist relevant enterprises and governments in formulating policies to conserve resources.

**Academic Community Impact**    This paper presents three contributions with far-reaching impact.

- We have summarized and, for the first time, formally defined the three types of dependencies present within time series, which will aid the academic community in designing more efficacious models based on these dependencies.
- Furthermore, we have adopted a novel architecture, Mamba, and elucidated its suitability for time series forecasting. We have also highlighted the prevalent overfitting tendencies of past architectures and provided a convenient and efficacious modification approach. These contributions can facilitate the research community in exploring the use of Mamba as a backbone for time series tasks or in seeking other novel architectures that are better suited, thus fostering the flourishing development of time series research.
- We have also proposed a decoupled strategy for simultaneously encoding temporal and variate relationships, thereby reliably introducing cross-variate dependencies and achieving state-of-the-art performance in both low and high-dimensional datasets. This generalizability effectively ensures the effectiveness of our proposed method and its capability for widespread application in real-life scenarios.

## U  DISCUSSION ON THE DIFFERENCE BETWEEN MAMBA AND SAMBA

Figure 11 shows the difference between Mamba and SAMBA. The primary difference between Mamba and SAMBA lies in the removal of the nonlinear activation function between the SSM

and Conv1D layers in SAMBA. This design choice aims to reduce unnecessary nonlinearity while preserving essential semantic dependencies in the time series data.

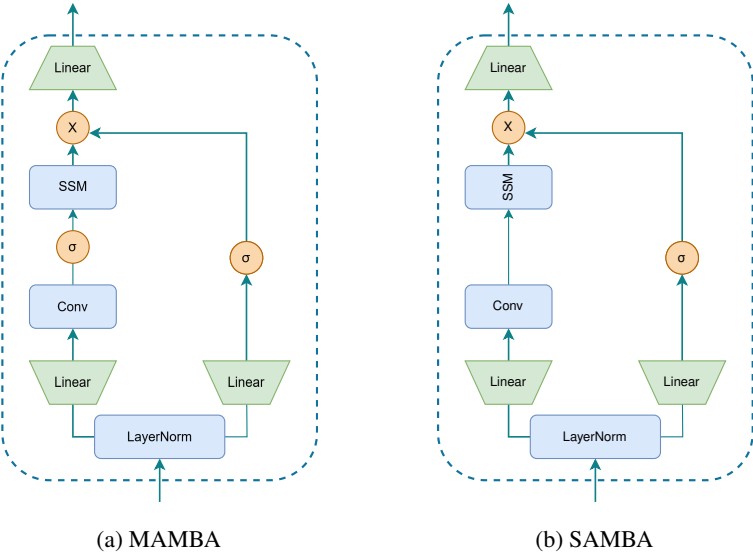

(a) MAMBA        (b) SAMBA

Figure 11: Comparison between MAMBA and SAMBA architectures.

## V   DISCUSSION ON FUTURE WORK

Although our work has made three significant contributions, we have also identified some limitations in our study.

- Limitations of the Work Scope. We have only demonstrated the significant negative impact of nonlinear activation functions on long-term time series forecasting tasks, where their removal leads to noticeable improvements, as well as the effectiveness of the decoupling strategy. However, we have not yet proven that these findings hold true for other tasks such as short-term time series forecasting, classification, etc.

- Exploration of Scaling Laws. A promising direction for current Transformer-based long-term time series forecasting models is to utilize pre-trained language models to verify the scaling laws that have been widely validated in natural language processing and computer vision. Mamba, as a potential alternative model to Transformer, has also been preliminarily proven to conform to scaling laws in the fields of NLP and CV in the current work. However, this paper lacks an exploration of whether Mamba also follows scaling laws in time series, which would affect its potential for continued exploration in the future. Additionally, the decoupling strategy also enables researchers to explore whether scaling laws are satisfied individually in the temporal and variate dimensions.

## W   SHOWCASES

To facilitate a clear comparison between different models, we present additional prediction examples from five key datasets in Figures 12 through 16. These examples are provided by the following models: SAMBA, CARD (Wang et al., 2024), FTP (Zhou et al., 2023), iTransformer (Liu et al., 2023), PatchTST (Nie et al., 2022), and DLinear (Zeng et al., 2023). Among these models, SAMBA demonstrates the most accurate predictions of future series variations, demonstrating superior performance.

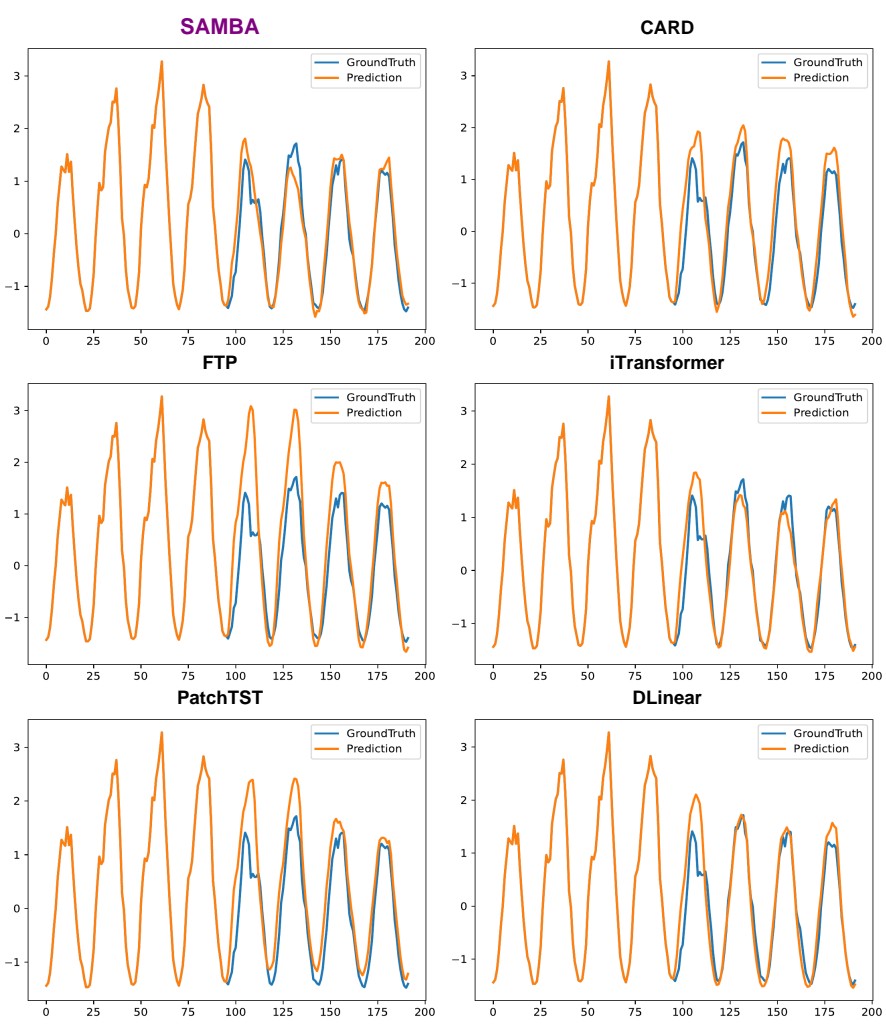

Figure 12: Visualization of input-96-predict-96 results on the Traffic dataset.

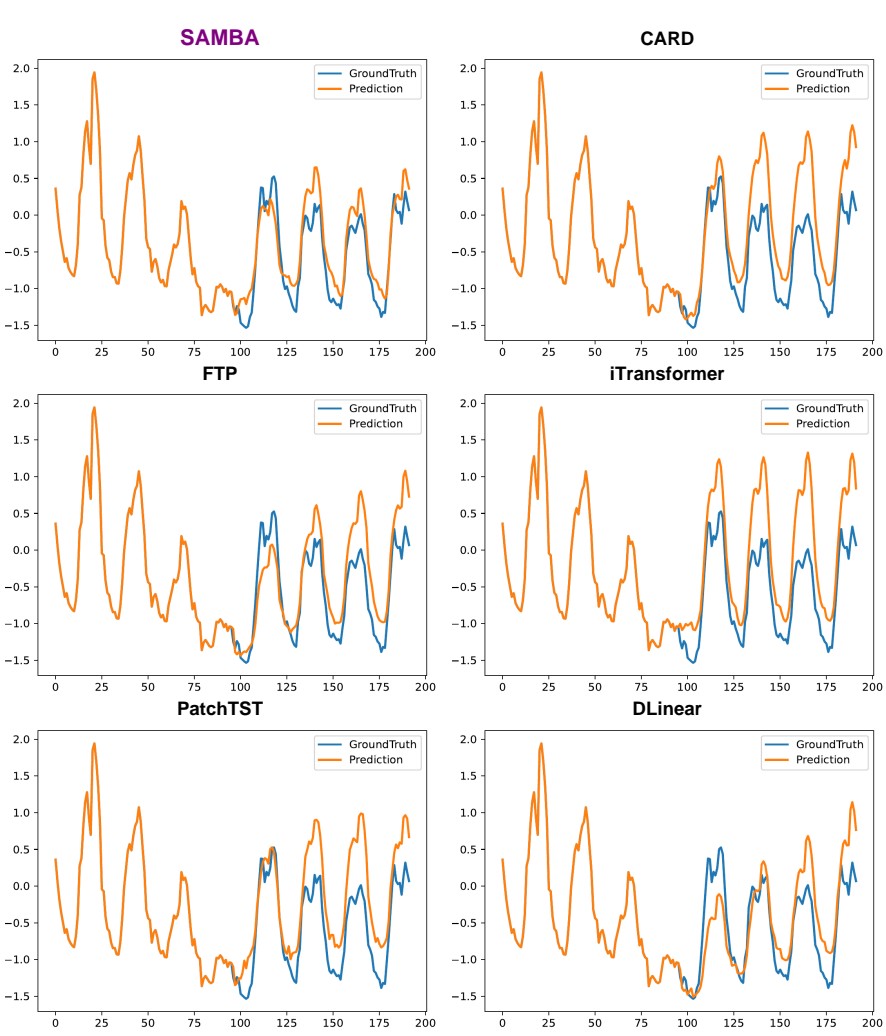

Figure 13: Visualization of input-96-predict-96 results on the ECL dataset.

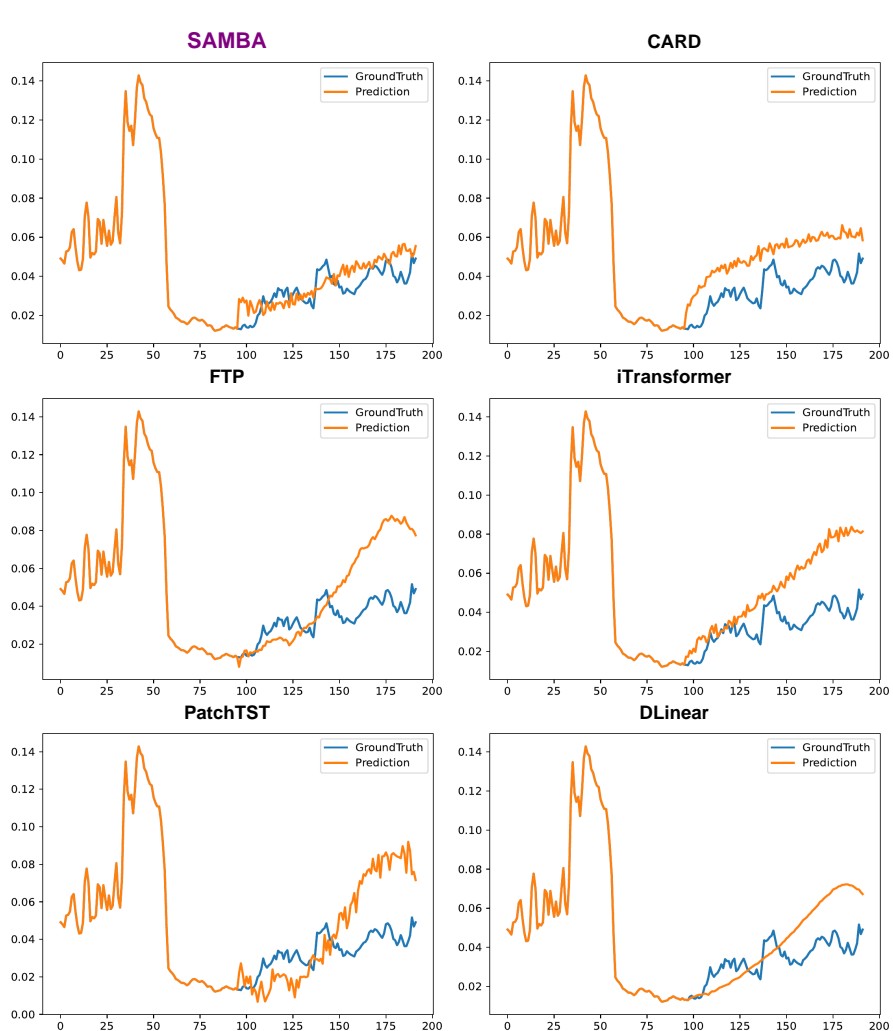

Figure 14: Visualization of input-96-predict-96 results on the Weather dataset.

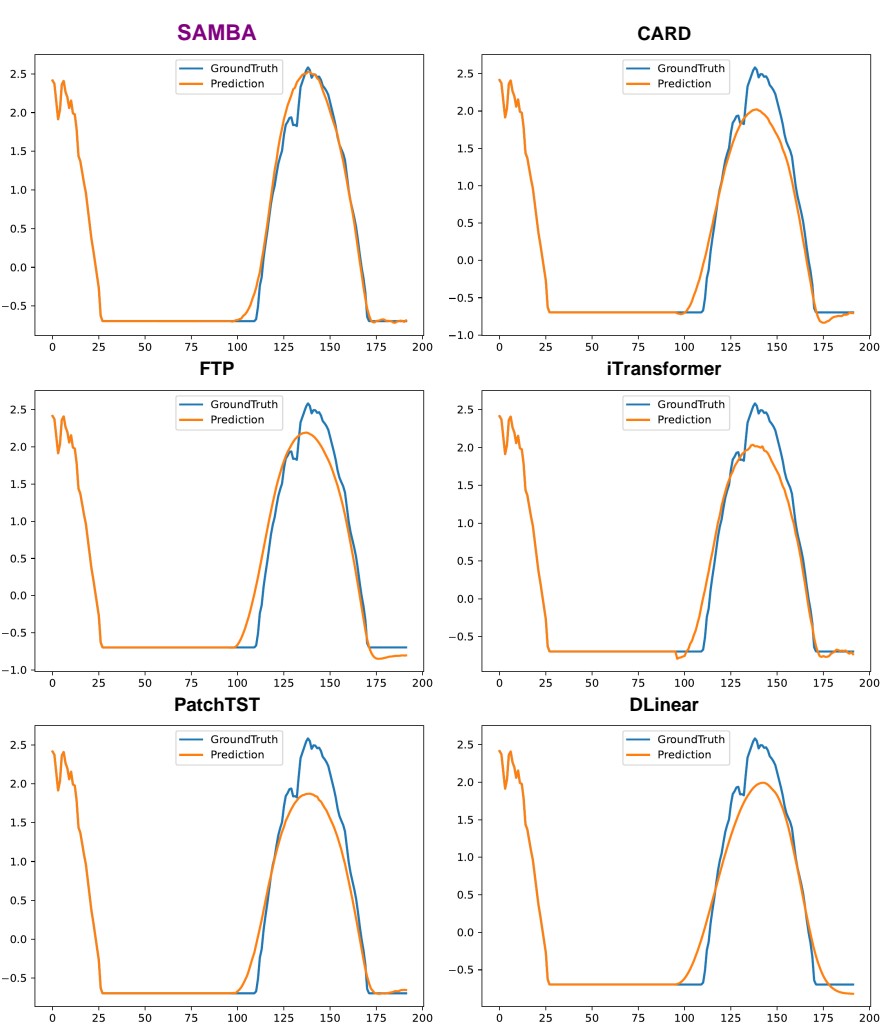

Figure 15: Visualization of input-96-predict-96 results on the Solar-Energy dataset.

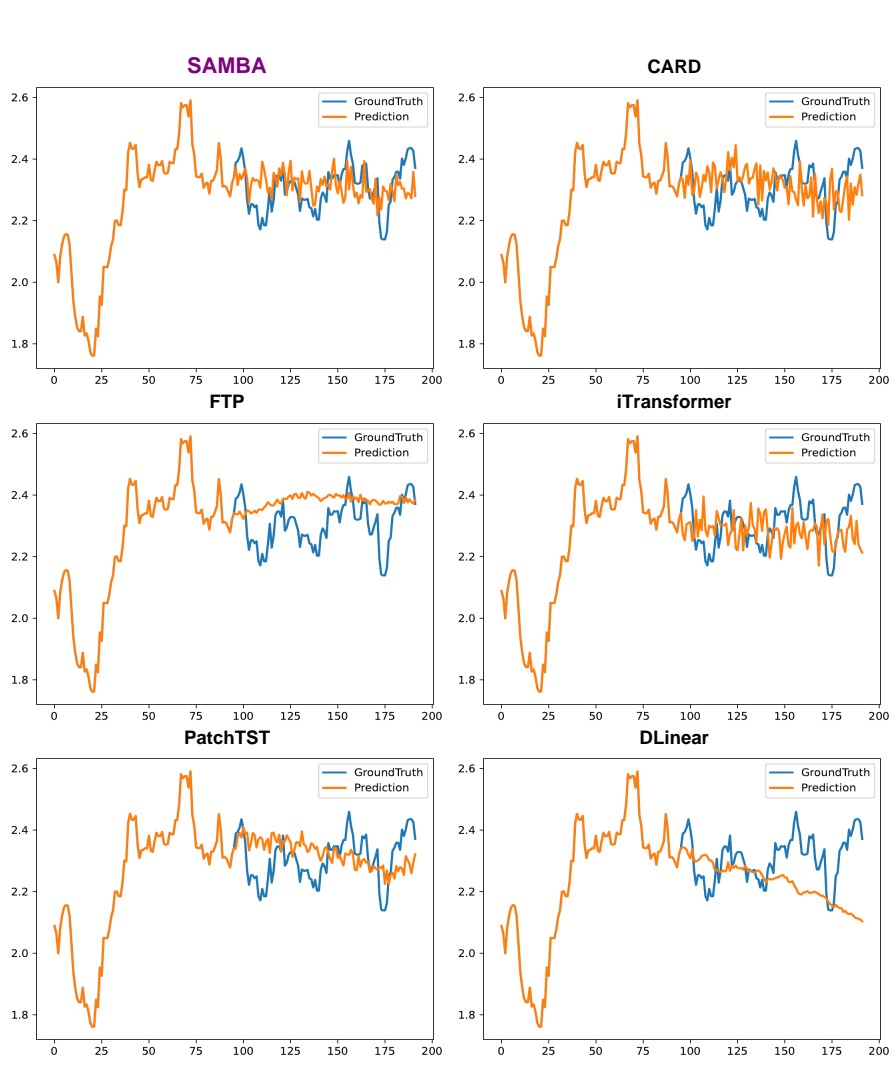

Figure 16: Visualization of input-96-predict-96 results on the Exchange dataset.

