# OpenReview forum: "Simplified Mamba with Disentangled Dependency Encoding for Long-Term Time Series Forecasting"
_ICLR.cc/2025/Conference — Submitted to ICLR 2025_

### Official Review · Reviewer_biFR · 2024-10-16

**Soundness:** 3
**Presentation:** 3
**Contribution:** 3
**Rating:** 6
**Confidence:** 5

**Summary:**

This paper proposes a Mamba-based structure to capture three sources of information from multivariate time series data: order dependency, semantic dependency and cross-variate dependency, and optimizes the Mamba structure by removing nonlinearities and incorporating a theoretically sound disentangled encoding strategy that appropriately integrates cross-variate dependency to the model, enhancing the model’s global representation and predictive capabilities. Experiments on multiple datasets conforms the efficacy of SAMBA structure.

**Strengths:**

The overall idea of applying Mamba for MTSF is novel, and the introduction of 'semantic dependency' is novel to the field of time series. Moreover, theroretical properties of the method are also provided.

**Weaknesses:**

The introduction of Mamba structure and its advantages compared with Transformer structure in MTSF should be in more details.

The explanation of  semantic dependency is not quite informative, and the boundary between order (temporal) dependency and semantic dependency seems unclear. Moreover, I wonder why the semantic dependency here does not include historical values from other covariates.

Moreover, there are many LLM-based methods for MTSF in recent years, and is highly recommended to be included in literature / benchmark methods such as LLM4TS, GPT4TS, CATS etc. I will happily raise my score if these concerns are addressed.

**Questions:**

I have several comments regarding the methodology details:
1. Is the model performance highly dependent on the parameters of patching?
2. The structure of SAMBA block should be directly compared with MAMBA in Fig 3 to show the difference.
3. How does SAMBA blocks achieve disentanglement encoding successully separate cross-time and cross-variate dependencies? It seems to me that such entanglements still exist through sum and back propagation.

---

> ### Author Response · Authors · 2024-11-21
> **Response to Reviewer biFR [Part 1]**
>
> Dear Reviewer biFR,
>
> Thank you for the suggestions regarding the model structure and definition, adding baselines, conducting experiments on the patch length hyperparameter, and disentangling the confusion in encoding have greatly enhanced the comprehensiveness of our experiments and resolved ambiguities. These suggestions significantly improve our paper. **Following your suggestions, we have added new baselines and evaluated SAMBA's sensitivity to patch parameters, reporting 54 new experimental results. We hope these efforts lead to a reassessment of our manuscript.** Below, we provide detailed, point-by-point responses to your concerns.
>
> >[W1] The introduction of Mamba structure and its advantages compared with Transformer structure in MTSF should be in more details.
>
> **[RW1]**  Thank you for your valuable comments. Your suggestions help improve the readability of our manuscript and reduce the difficulty for readers to understand. In the $\underline{\text{Appendix G of revised paper}}$, we start with the classical state-space model theory, introduce the advancements in modern SSM, and then highlight Mamba's contributions, **comprehensively outlining the development trajectory of state-space models.** Additionally, we provide **a detailed explanation of Mamba's advantages over Transformer in the LTSF domain.** We hope our efforts have effectively addressed your concerns.
>
> >[W2-1] The explanation of semantic dependency is not quite informative, and the boundary between order (temporal) dependency and semantic dependency seems unclear.
>
> **[RW2-1]** We appreciate the reviewer's observations about potential limitations. In addition to the formal definitions provided in $\underline{\text{Section 3}}$, we have added a new section in $\underline{\text{Appendix P of revised paper}}$ to explain order dependency and semantic dependency comprehensively. This section is intended to help readers better understand these dependencies and their distinctions.
>
> >[W2-2]  I wonder why the semantic dependency here does not include historical values from other covariates.
>
> **[RW2-2]** Thank you for pointing out this interesting and insightful question. Indeed, historical values from other covariates represent an important dependency. In our definition, **we adopt a more fine-grained perspective to consider the semantics of time series.** LTSF actually involves two types of semantic relationships: **intra-variate patterns,** i.e., patterns within a single variate’s time series, and **inter-variate patterns,** i.e., interactions between different variates, which corresponds to what you refer to as historical values from other covariates. The semantics we define focus on intra-variate patterns, while cross-variate dependency emphasizes inter-variate patterns. Such a definition considers the intrinsic properties of each variate independently, **enabling a more effective analysis of their distinct contributions.**

---

> ### Author Response · Authors · 2024-11-21
> **Response to Reviewer biFR [Part 2]**
>
> >[W3] Moreover, there are many LLM-based methods for MTSF in recent years, and is highly recommended to be included in literature / benchmark methods such as LLM4TS, GPT4TS, CATS etc.
>
> **[RW3]** Thank you for pointing out several impactful and interesting works. Incorporating these as baselines will enhance the comprehensiveness of our manuscript. Indeed, GPT4TS has already been included as a baseline under the name FTP in our work. And unfortunately, we could not find official code implementations for LLM4TS [1], and CATS [2] faces a similar issue, as only a Jupyter Notebook is provided without a formal implementation. We plan to reproduce the aforementioned works in the future and include them as baselines. But due to the tight schedule during rebuttal, we incorporated AutoTimes [3], a recently LLM-based published work on NeurIPS 2024, and used its GPT-2 implementation as a baseline. The experimental results demonstrate that SAMBA still achieve the best performance.
>
> | Models         | SAMBA (Ours)     |                  | AutoTimes (2024) |                  | FTP (2023)       |                  |
> |----------------|------------------|------------------|------------------|------------------|------------------|------------------|
> |                | MSE              | MAE              | MSE              | MAE              | MSE              | MAE              |
> | **ECL**        | 0.172            | 0.268            | 0.188            | 0.275            | 0.210            | 0.291            |
> | **ETTh1**      | 0.443            | 0.432            | 0.464            | 0.451            | 0.450            | 0.439            |
> | **ETTh2**      | 0.363            | 0.392            | 0.403            | 0.417            | 0.385            | 0.411            |
> | **ETTm1**      | 0.378            | 0.394            | 0.392            | 0.409            | 0.392            | 0.401            |
> | **ETTm2**      | 0.276            | 0.322            | 0.289            | 0.334            | 0.285            | 0.331            |
> | **Exchange**   | 0.356            | 0.401            | 0.366            | 0.405            | 0.368            | 0.406            |
> | **Traffic**    | 0.422            | 0.276            | 0.501            | 0.330            | 0.511            | 0.334            |
> | **Weather**    | 0.249            | 0.278            | 0.270            | 0.293            | 0.267            | 0.287            |
> | **Solar-Energy** | 0.229          | 0.253            | 0.256            | 0.297            | 0.260            | 0.304            |
>
>
> *Note: Reported results are the averages over four prediction horizons: 96, 192, 336, and 720.*
>
> >[Q1] Is the model performance highly dependent on the parameters of patching?
>
> **[RQ1]** Thank you to the reviewer for raising the question about the patch length hyperparameter. To address this, we conduct experiments on the ETTm1, ETTh1, and Weather datasets with the patch size from 2 to 32. The results are as follows. **The experimental results indicate that SAMBA is robust to changes in patch length.** The visualization results are included in $\underline{\text{Appendix H of revised paper}}$.
>
> | Dataset   | Metric  | 2      | 4      | 8      | 16     | 24     | 32     |
> |-----------|---------|--------|--------|--------|--------|--------|--------|
> | **ETTm1** | MSE     | 0.329  | 0.320  | 0.320  | 0.315  | 0.317  | 0.327  |
> |           | MAE     | 0.366  | 0.360  | 0.359  | 0.357  | 0.358 | 0.365  |
> | **ETTh1** | MSE     | 0.422  | 0.383  | 0.382  | 0.376  | 0.385  | 0.386  |
> |           | MAE     | 0.427  | 0.409  | 0.401  | 0.400  | 0.404  | 0.406  |
> | **Weather** | MSE   | 0.180  | 0.175 | 0.167  | 0.165  | 0.174  | 0.178  |
> |           | MAE     | 0.223  | 0.220  | 0.212  | 0.214  | 0.222 | 0.223  |
>
>
>
> >[Q2] The structure of SAMBA block should be directly compared with MAMBA in Fig 3 to show the difference.
>
> **[RQ2]** Thank you for the suggestion. In the $\underline{\text{Appendix U of revised paper}}$, we have included a direct comparison between SAMBA and the Mamba block to help readers better understand the differences between the two.

---

> > ### Comment · Reviewer_biFR · 2024-11-25
> >
> > I appreciate the authors for solving my concerns, and I agree that during backpropagation, the aggregation operation avoids any entanglement. The revised paper also seems better organized. I will keep my rating.

---

> > > ### Author Response · Authors · 2024-11-27
> > > **Thanks for your response**
> > >
> > > We sincerely appreciate your positive evaluation and recognition of the contributions of this work. Based on your encouraging comments, we were wondering if you might consider reflecting this in the score, as we would greatly value your support in further acknowledging the impact of this study. And if there are any remaining concerns or points where further clarification could enhance your evaluation, we would be happy to address them in more detail.

---

> ### Author Response · Authors · 2024-11-21
> **Response to Reviewer biFR [Part 3]**
>
> >[Q3] How does SAMBA blocks achieve disentanglement encoding successully separate cross-time and cross-variate dependencies? It seems to me that such entanglements still exist through sum and back propagation.
>
> **[RQ3]** We appreciate the reviewer for pointing this out. This is, in fact, **a key aspect of the disentangled encoding design.** During the disentangled encoding process, we use $x$ layers of SAMBA and $y$ layers of bi-SAMBA (where $x$ and $y$ can be different values) to separately encode cross-time dependency and cross-variate dependency. Regarding the reviewer's concern that entanglements may still exist in the aggregation operation, we will now provide a detailed discussion of how we achieve  disentangled.
>
> First, there is a misunderstanding in the reviewer's concern. We achieve aggregation by concatenating the separately learned temporal and variate representations and inputting them into the FFN, rather than aggregating them through a sum operation.
> The aggregation formula can be expressed as follows:
>
> $
> \mathbf{E}\_o = \text{FFN}(\mathbf{E}\_{\text{time}} || \mathbf{E}\_{\text{var}}).
> $
>
> FFN is a fully connected feed-forward neural network, it can be expressed as:
>
> $
> \text{FFN}(\mathbf{x}) = \mathbf{W}\_2 \sigma(\mathbf{W}\_1 \mathbf{x} + \mathbf{b}\_1) + \mathbf{b}\_2,
> $
>
> where $\mathbf{x} = \mathbf{E}\_{\text{time}} || \mathbf{E}\_{\text{var}}$ represents the concatenated input vector, $\mathbf{W}\_1, \mathbf{W}\_2$ are weight matrices, $\mathbf{b}\_1, \mathbf{b}\_2$ are bias vectors, $\sigma$ is a non-linear activation function (e.g., ReLU).
>
> Since backpropagation involves the derivatives of the output with respect to the inputs, we analyze the derivatives of $\mathbf{E}\_o$ with respect to $\mathbf{E}\_{\text{time}}$ and $\mathbf{E}\_{\text{var}}$.
>
> ### Derivative with Respect to $\mathbf{E}\_{\text{time}}$
>
> For the concatenated input:
> $
> \mathbf{x} = \begin{bmatrix} \mathbf{E}\_{\text{time}} \\ \mathbf{E}\_{\text{var}} \end{bmatrix},
> $
>
> the derivative of $\mathbf{E}\_o$ with respect to $\mathbf{E}\_{\text{time}}$ is:
>
> $
> \frac{\partial \mathbf{E}\_o}{\partial \mathbf{E}\_{\text{time}}} = \frac{\partial \mathbf{E}\_o}{\partial \mathbf{x}} \cdot \frac{\partial \mathbf{x}}{\partial \mathbf{E}\_{\text{time}}}.
> $
>
> Since $\mathbf{x}$ contains $\mathbf{E}\_{\text{time}}$ as its first part:
>
> $
> \frac{\partial \mathbf{x}}{\partial \mathbf{E}\_{\text{time}}} = \mathbf{I},
> $
>
> where $\mathbf{I}$ is the identity matrix.
>
> The derivative of the FFN with respect to the input $\mathbf{x}$ is:
>
> $
> \frac{\partial \mathbf{E}\_o}{\partial \mathbf{x}} = \mathbf{W}\_2 \cdot \text{diag}(\sigma'(\mathbf{W}\_1 \mathbf{x} + \mathbf{b}\_1)) \cdot \mathbf{W}\_1.
> $
>
> Thus:
>
> $
> \frac{\partial \mathbf{E}\_o}{\partial \mathbf{E}\_{\text{time}}} = \mathbf{W}\_2 \cdot \text{diag}(\sigma'(\mathbf{W}\_1 \mathbf{x} + \mathbf{b}\_1)) \cdot \mathbf{W}\_1 \cdot \mathbf{P}\_{\text{time}},
> $
>
> where $\mathbf{P}\_{\text{time}}$ is the projection matrix that selects the $\mathbf{E}\_{\text{time}}$ part from the concatenated vector.
>
> ---
>
> ### Derivative with Respect to $\mathbf{E}\_{\text{var}}$
>
> Similarly, the derivative with respect to $\mathbf{E}\_{\text{var}}$ is:
>
> $
> \frac{\partial \mathbf{E}\_o}{\partial \mathbf{E}\_{\text{var}}} = \mathbf{W}\_2 \cdot \text{diag}(\sigma'(\mathbf{W}\_1 \mathbf{x} + \mathbf{b}\_1)) \cdot \mathbf{W}\_1 \cdot \mathbf{P}\_{\text{var}},
> $
>
> where $\mathbf{P}\_{\text{var}}$ is the projection matrix that selects the $\mathbf{E}\_{\text{var}}$ part from the concatenated vector.
>
> ---
>
> ### Summary
>
> $
> \frac{\partial \mathbf{E}\_o}{\partial \mathbf{E}\_{\text{time}}} = \mathbf{W}\_2 \cdot \text{diag}(\sigma'(\mathbf{W}\_1 \mathbf{x} + \mathbf{b}\_1)) \cdot \mathbf{W}\_1 \cdot \mathbf{P}\_{\text{time}},
> $
>
> $
> \frac{\partial \mathbf{E}\_o}{\partial \mathbf{E}\_{\text{var}}} = \mathbf{W}\_2 \cdot \text{diag}(\sigma'(\mathbf{W}\_1 \mathbf{x} + \mathbf{b}\_1)) \cdot \mathbf{W}\_1 \cdot \mathbf{P}\_{\text{var}}.
> $
>
> We can observe that the derivative of $\mathbf{E}\_o$ with respect to $\mathbf{E}\_{\text{time}}$ involves only $\mathbf{P}\_{\text{time}}$, while the derivative with respect to $\mathbf{E}\_{\text{var}}$ involves only $\mathbf{P}\_{\text{var}}$. Therefore, **even during backpropagation, the aggregation operation avoids any entanglement.** We hope this explanation helps to resolve your doubts. This content also be included in our Appendix O of revised paper.
>
> Overall, We greatly appreciate the reviewer's insights, as they have significantly enhanced the quality of our manuscript.
> We hope that we have addressed your concerns and questions. We look forward to your further feedback and evaluations.
>
> [1] LLM4TS: Aligning Pre-Trained LLMs as Data-Efficient Time-Series Forecasters, Arxiv 2024
>
> [2] CATS: Enhancing Multivariate Time Series Forecasting by Constructing Auxiliary Time Series as Exogenous Variables, ICML 2024
>
> [3] AutoTimes: Autoregressive Time Series Forecasters via Large Language Models, NeurIPS 2024

---

> ### Author Response · Authors · 2024-11-25
> **[**Gentle Reminder**]: Kind Request for Reviewers' Feedback**
>
> Dear Reviewer biFR,
>
> Thank you once again for your valuable and constructive review, which has helped us refine our contribution and clarify its strengths.
>
> We would like to kindly remind you that the discussion deadline is approaching. After this deadline, we may not have the opportunity to respond to your comments.
>
> Additionally, during the rebuttal period, we have supplemented our study with **over 250 experimental results** to enhance the comprehensiveness of our experiments and the reliability of our conclusions. These results address your concerns regarding the experiments and are all included in $\underline{\text{the revised paper, highlighted in blue}}$.
>
> We sincerely appreciate your dedication and look forward to your feedback.
>
> Sincerely,
> ICLR 2025 Conference Submission 767 Authors

---

### Official Review · Reviewer_y2RJ · 2024-10-30

**Soundness:** 2
**Presentation:** 3
**Contribution:** 2
**Rating:** 3
**Confidence:** 4

**Summary:**

The authors propose a simplified Mamba with disentangled dependency encoding for long-term time series forecasting, in which the nonlinearities in vanilla Mamba are removed to improve the generalization ability of the framework and a theoretically sound disentangled encoding strategy is introduced to separate the cross-time and cross-variate dependencies. Experiments demonstrate the effectiveness of the proposed framework.

**Strengths:**

1.The organization of this paper is clear.

2.The definition of three critical dependencies in time series data is novel and interesting.

**Weaknesses:**

1.This paper lacks innovation, as the authors only model interactions along time dimension and variate dimension separately, without the targeted design towards disentangled dependency encoding strategy. The authors should provide a detailed description towards their disentangled dependency encoding strategy.

2.The design of the SAMBA block is also similar to existing works [1, 2]. It seems that the only difference between this work and existing work is the removal of the nonlinear activation function between the Conv1D and SSM layers. Although the authors claim that this approach can mitigate the overfitting issue, Table 3 shows that for the Path+Mamba method, removing the nonlinear activation function actually results in a performance drop. The authors should provide a theoretical analysis about removing the nonlinear activation function can mitigate the overfitting issue and validate their claim on additional datasets (e.g., ECL, Traffic, and Weather datasets).

3.The paper has some weaknesses in the experiments, which are not convincing enough:

(1)The authors claim that they implement the baseline results by using the TimesNet Github repository in Section B.1. However, they also claim that the full results of predictions come from iTransformer in Section B.2, which is confusing. It would be helpful to have an explanation for these differences.

(2) Some tables lack sufficient explanation, making them difficult to understand. For example, in Table 1, what do the bold results mean? Does the transformer refer to the ordinary transformer framework or a state-of-the-art (SOTA) transformer-based framework (e.g., iTransformer)? In addition, in Table 3, why is there no comparison with the Patch+MLP method? The authors should provide a detailed explanation of the notations and abbreviations used in the Tables. In addition, the authors should include comparative experiments with the SOTA transformer-based framework and the Patch+MLP method in Table 1 and Table 3, respectively.

(3) Although the authors add the efficiency comparison between SAMBA and the baseline models, there is no clarification on which dataset(s) the comparison is conducted. In addition, the term 'Mem' in the efficiency comparison is not explained, which makes the experiments regarding efficiency comparison confusing. Please specify which dataset(s) are used for the efficiency comparison, and provide an explanation for the 'Mem' metric used for the efficiency comparison.

4. There are many typos and writing mistakes in the manuscript. For example, on page 22, “Solar-Energy)or” should be "Solar-Energy) or" and "Exchange ) overall" should be "Exchange) overall". The manuscript requires thorough proofreading.

**Questions:**

See Weaknesses.

---

> ### Author Response · Authors · 2024-11-21
> **Response to Reviewer y2RJ [Part 1]**
>
> Dear Reviewer y2RJ,
>
> we would like to sincerely thank you for providing a detailed review and insightful suggestions. **Following your suggestions, we conducted over 500 additional experiments and reported new results for more than 250 of them. We hope these efforts lead to a reassessment of our manuscript.** Below, we provide detailed, point-by-point responses to your concerns.
>
> >[W1] This paper lacks innovation, as the authors only model interactions along time dimension and variate dimension separately, without the targeted design towards disentangled dependency encoding strategy. The authors should provide a detailed description towards their disentangled dependency encoding strategy.
>
> **[RW1]** Thank you for the constructive suggestion. We understand that the reviewer may be concerned about the lack of explicit mechanisms to ensure successful disentanglement, similar to those was done in Disentangled Representation Learning (DRL) literature. DRL focuses on identifying representations of latent variables within the Data Generation Process (DGP), often relying on tailored regularizers grounded in strict assumptions on DGP, which limits its broad applicability.
>
> **Our main objective is to improve the modeling of both cross-time and cross-variate dependencies in LTSF data.** While using DRL to generate latent representations that characterize these two dependencies separately is a meaningful direction, designing a universal approach that ensures identifiability across time series data in diverse domains is very challenging—given the significant DGP differences between datasets. For example, in traffic datasets, the cross-variate dependency originates from the human moving patterns, while the cross-variate dependency in electricity data originates from semantic relations among varied attributes of electricity power.
> **Instead, our approach explores the direction of introducing distinct inductive biases into the encoders to encourage disentanglement.** Specifically, we alter the dimension of sequential modeling within the encoder. We have empirically validated this strategy across various backbone models. Additionally, we provide theoretical guarantees that, under mild assumptions about the two dependencies, our disentangled encoding strategy outperforms the existing time-then-variate encoding approach. We believe our strategy possesses generality and novelty within the LTSF literature.
>
> Finally, apart from the disentangled strategy, we also make other innovative contributions in this paper: (1) Identification and formal definition of three critical dependencies in time series data; (2) In-depth analysis of Mamba’s advantages; (3) Discovering and addressing overfitting issues caused by model non-linearities.
>
> >[W2-1] The design of the SAMBA block is also similar to existing works [1, 2]. It seems that the only difference between this work and existing work is the removal of the nonlinear activation function between the Conv1D and SSM layers.
>
> **[RW2-1]**  Thank you for the reviewer’s suggestions. We indeed focus on studying different modules of the Mamba. However, the existing works on Mamba that remove Conv1D convolutions [1][2] rely on prior findings [3][4], whereas **our proposal to remove non-linear activation functions is based on our empirical analysis and is presented here for the first time.**

---

> ### Author Response · Authors · 2024-11-21
> **Response to Reviewer y2RJ [Part 2]**
>
> >[W2-2] Although the authors claim that this approach can mitigate the overfitting issue, Table 3 shows that for the Path+Mamba method, removing the nonlinear activation function actually results in a performance drop.
>
> **[RW2-2]** Thank you for pointing out this concern. In fact, removing non-linear activation functions is generally beneficial, as confirmed by our ablation experiments. To further address the reviewer’s concern, we conducted large-scale experiments to verify the benefits of removing non-linear activation functions. The table below demonstrates the impact of removing non-linear activation functions on the Exchange dataset. The experimental results show that removing non-linear activation functions is consistently beneficial. As shown in our response to reviewer qdrT [W3-4], this result is also robust to the patch size. More comprehensive experimental results are included in the $\underline{\text{Appendix F.3 of revised paper}}$.
>
> **Table: The effect of nonlinear activation function on the model**
> | Dataset       | T   | Patch+MLP         |            | Patch+MLP-n           |            |Patch+Mamba            |            | Patch+Mamba-n       |            | Patch+Transformer           |   |Patch+ Transformer-n            |            |
> |---------------|-----|----------------|------------|------------|----------------|------------|------------|----------------|------------|------------|----------------|------------|------------|
> |  |     | MSE | MAE  | MSE | MAE            | MSE        | MAE | MSE            | MAE        | MSE | MAE            | MSE        | MAE |
> | **Exchange**   | 96  | 0.0968      | 0.233      | 0.0945       | 0.219          | 0.0871     | 0.207      | 0.0834     | 0.202      | 0.0861      | 0.204          | 0.0866      | 0.205       |
> |           | 192 | 0.269         | 0.392      | 0.156       | 0.285          | 0.176      | 0.298      |  0.174         |0.295      |   0.183     | 0.303          | 0.180      | 0.302      |
> |               | 336 | 0.366        | 0.454     | 0.291      | 0.399          | 0.327      | 0.413     | 0.326          | 0.412      | 0.332      | 0.417          | 0.329     | 0.415      |
> |               | 720 | 0.879          | 0.714     | 0.744      | 0.694          | 0.853     |0.694      | 0.845          | 0.691      | 0.854      | 0.697         | 0.851      | 0.695      |
> |               | Avg | 0.403          | 0.448      | 0.321      | 0.399          | 0.361     |0.403      | 0.357          | 0.400      | 0.364      | 0.405         | 0.362      | 0.404      |
>
> *Note: T means prediction length.`-n` indicates the removal of nonlinear activation functions. Avg means the average results over four prediction lengths: 96, 192, 336, and 720.*
>
> >[W2-3] The authors should provide a theoretical analysis about removing the nonlinear activation function can mitigate the overfitting issue and validate their claim on additional datasets (e.g., ECL, Traffic, and Weather datasets).
>
> **[RW2-3]** Thank you for providing valuable directions for our work.Theoretical analysis will be the focus of our future research. Following your suggestion, we expanded the scope of our experiments to include the Exchange and Traffic datasets. **The table below demonstrates that removing non-linear activation functions benefits both Mamba and Transformer models** (more results are included in the $\underline{\text{Appendix F.3 of revised paper}}$). However, the limited improvements on the Traffic dataset, along with the performance decline of MLP, are attributed to the stronger semantic complexity and the involvement of more non-linear relationships in the Traffic dataset. The simplified architecture of MLP, without non-linear activation functions, is unable to effectively handle such complexity. In contrast, the performance improvements observed for Transformer and Mamba suggest that their inherently complex architectures are already capable of handling these relationships, making non-linear activation functions redundant. Training curves in $\underline{\text{Appendix F.3 of revised paper}}$ further support this conclusion.
>
> | Datsets     | Model        | MLP    |        | Mamba  |        | Transformer |        |
> |--------------|---------------|--------|--------|--------|--------|-------------|--------|
> |              |               | MSE    | MAE    | MSE    | MAE    | MSE         | MAE    |
> | **Exchange** | Original      | 0.398  | 0.419  | 2.255  | 1.189  | 1.994       | 1.117  |
> |              | Original-n    | 0.374  | 0.407  | 2.122  | 1.143  | 1.194       | 0.895  |
> |              | Improvement   | 6.03%  | 2.86%  | 5.90%  | 3.79%  | 40.13%      | 19.87% |
> | **Traffic**  | Original      | 0.554  | 0.366  | 0.669  | 0.385  | 0.833       | 0.480  |
> |              | Original-n    | 0.621  | 0.400  | 0.658  | 0.381  | 0.829       | 0.479  |
> |              | Improvement   | -12.27%| -9.28% | 1.57%  | 0.98%  | 0.42%       | 0.16%  |

---

> ### Author Response · Authors · 2024-11-21
> **Response to Reviewer y2RJ [Part 3]**
>
> [W3] 3.The paper has some weaknesses in the experiments, which are not convincing enough.
>
> >[W3-1] The authors claim that they implement the baseline results by using the TimesNet Github repository in Section B.1. However, they also claim that the full results of predictions come from iTransformer in Section B.2, which is confusing. It would be helpful to have an explanation for these differences.
>
> **[RW3-1]** We apologize for any misunderstanding caused by the phrasing in our paper, and we have revised the ambiguous expressions. However, the results remain accurate. This is because iTransformer is implemented based on the TimesNet GitHub repository. Upon inspection, the TimesNet GitHub repository links to the Time-Series-Library, and iTransformer is implemented using the Time-Series-Library. **Therefore, the results for iTransformer are based on its implementation within the Time-Series-Library.** And we have revised them in the $\underline{\text{Appendix B.2 of revised paper}}$.
>
> >[W3-2-1]  Some tables lack sufficient explanation, making them difficult to understand. For example, in Table 1, what do the bold results mean?
>
> **[RW3-2-1]** Thank you for pointing this out. Initially, the bold text was used solely for emphasis and clarity. We have addressed these issues in our revised paper to avoid any misunderstanding.
>
> >[W3-2-2] Does the transformer refer to the ordinary transformer framework or a state-of-the-art (SOTA) transformer-based framework (e.g., iTransformer)?
>
> **[RW3-2-2]**  Regarding the Transformer, we have already clarified in $\underline{\text{Appendix B.4 of original paper}}$ that it refers to the original Transformer.

---

> ### Author Response · Authors · 2024-11-21
> **Response to Reviewer y2RJ [Part 4]**
>
> >[W3-2-3]  In addition, in Table 3, why is there no comparison with the Patch+MLP method? The authors should include comparative experiments with the SOTA transformer-based framework and the Patch+MLP method in Table 1 and Table 3, respectively.
>
> **[RW3-2-3]** Thank you for the valuable suggestion regarding the comprehensiveness of our experiments. Following your suggestion, we included the latest SOTA model, iTransformer, which uses a Linear Model to encode temporal relationships and is therefore expected to be sensitive to time order. The results are shown in the table below. **The following experimental results demonstrate that both Linear and Mamba effectively capture order dependency.** The additional results for Patch+MLP regarding Table 3 have already been presented in [RW2-2]. More comprehensive results for the above two tables are included in the $\underline{\text{Appendix F of revised paper}}$.
> | Datasets     | Prediction Length | Linear Model |        |       |         | Mamba |        |        |          | Transformer |      |      |         | iTransformer|      |       |             |
> |--------------|--------------------|--------------|--------|-------|---------|-------|--------|--------|----------|-------------|------|------|---------|-------------|------|-------|-------------|
> |              |                    | O.MSE        | S.MSE  | O.MAE | S.MAE   | O.MSE | S.MSE  | O.MAE  | S.MAE    | O.MSE       | S.MSE| O.MAE| S.MAE   | O.MSE       | S.MSE| O.MAE | S.MAE       |
> | **ETTm1**    | 96                 | 0.383        | 0.988  | 0.400 | 0.697   | 0.517 | 0.922  | 0.508  | 0.688    | 0.643       | 0.884| 0.575| 0.643   | 0.345       | 0.892| 0.378 | 0.610       |
> |              | 192                | 0.413        | 0.986  | 0.415 | 0.697   | 0.575 | 0.931  | 0.546  | 0.699    | 0.805       | 1.01 | 0.664| 0.730   | 0.383       | 0.903| 0.395 | 0.617       |
> |              | 336                | 0.441        | 0.987  | 0.435 | 0.698   | 0.730 | 0.957  | 0.634  | 0.703    | 0.882       | 1.12 | 0.737| 0.817   | 0.423       | 0.923| 0.420 | 0.630       |
> |              | 720                | 0.497        | 0.992  | 0.469 | 0.704   | 0.873 | 0.973  | 0.704  | 0.723    | 0.928       | 1.12 | 0.752| 0.800   | 0.489       | 0.932| 0.456 | 0.641       |
> |              | **Avg. Drop**      | -            | 127.97%| -     | 62.55%  | -     | 40.37% | -      | 17.60%   | -           | 22.40%|-     | 6.55%   | -           | 122.56%|-     | 0.515%      |
> | **Exchange** | 96                 | 0.0832       | 0.210  | 0.201 | 0.332   | 1.260 | 1.401  | 0.915  | 0.943    | 0.730       | 0.738| 0.782| 0.722   | 0.0869      | 0.242| 0.207 | 0.358       |
> |              | 192                | 0.179        | 0.325  | 0.299 | 0.414   | 1.398 | 1.626  | 1.040  | 1.060    | 1.304       | 1.284| 0.913| 0.949   | 0.179       | 0.374| 0.301 | 0.450       |
> |              | 336                | 0.338        | 0.521  | 0.418 | 0.534   | 1.835 | 1.921  | 1.111  | 1.141    | 1.860       | 1.862| 1.090| 1.085   | 0.331       | 0.535| 0.417 | 0.557       |
> |              | 720                | 0.903        | 1.167  | 0.714 | 0.822   | 3.940 | 4.023  | 1.687  | 1.697    | 3.860       | 3.865| 1.684| 1.685   | 0.856       | 1.202| 0.698 | 0.841       |
> |              | **Avg. Drop**      | -            | 47.89% | -     | 28.80%  | -     | 6.38%  | -      | 1.85%    | -           | -0.06%|-     |-0.63%   | -           | 63.33%|-     | 35.89%      |
>
> *Note: O.MSE and O.MAE are evaluated in the original test set. S.MSE and S.MAE are evaluated in the shuffling test set.*
>
> >[W3-2-4] The authors should provide a detailed explanation of the notations and abbreviations used in the Tables.
>
> **[RW3-2-4]** Thank you for the valuable suggestion. We revise the tables to provide detailed explanations of the symbols and abbreviations used.

---

> ### Author Response · Authors · 2024-11-21
> **Response to Reviewer y2RJ [Part 5]**
>
> >[W3-3-1] Although the authors add the efficiency comparison between SAMBA and the baseline models, there is no clarification on which dataset(s) the comparison is conducted.
>
> **[RW3-3-1]** Thanks for the reviewer's valuable suggestion. Your feedback helps improve the completeness of our experiments. The results in the paper were measured on the ETTm1 dataset. **To provide a more comprehensive evaluation, we further measured the results on the Traffic dataset.** The results are shown below and have been included in the $\underline{\text{Appendix G of revised paper}}$. SAMBA achieves both a faster training speed and a smaller memory usage compared to many SOTA transformer-based models, such as PatchTST and Crossformer, which also employ attention mechanisms in temporal dimensions.
>
> Table : Efficiency Analysis: The GPU memory (MiB) and speed (running time, s/iter) of each model on the Traffic dataset. Mem means memory footprint.
>
>
> |  Input Length   |         96 |   | 336   |    | 720   |   |
> |----------------|--------------|-------|---------|-------|---------|-------|
> |  Models         |    Mem    |  Speed   | Mem   |  Speed       | Mem   | speed        |
> | SAMBA          |    2235  | 0.0403  | 2275  | 0.0711  | 2311  | 0.1232  |
> | PatchTST       |    3065  | 0.0658  | 12299 | 0.2382  | 25023 | 0.4845  |
> | iTransformer   |   3367  | 0.0456  | 3389  | 0.0465  | 3411  | 0.0482  |
> | DLinear        |    579   | 0.0057  | 619   | 0.0082  | 681   | 0.0139  |
> | TimesNet       |   6891  | 0.2492  | 7493  | 0.4059  | 8091  | 0.6289  |
> | Crossformer    |  21899 | 0.1356  | 40895 | 0.1369  | 69711 | 0.1643  |
> | FEDFormer      |   1951  | 0.1356| 1957  | 0.1369   | 2339  | 0.1643   |
> | Autoformer     |    1489  | 0.0309  | 1817  | 0.0362  | 2799  | 0.0457  |
>
>
>
> >[W3-3-2] In addition, the term 'Mem' in the efficiency comparison is not explained, which makes the experiments regarding efficiency comparison confusing. Please specify which dataset(s) are used for the efficiency comparison, and provide an explanation for the 'Mem' metric used for the efficiency comparison.
>
> **[RW3-3-2]** Thank you for the valuable suggestion to avoid misunderstandings. The term Mem refers to the GPU memory utilized during model training. We have added this explanation to the revised paper.
>
> >[RW4] There are many typos and writing mistakes in the manuscript. For example, on page 22, “Solar-Energy)or” should be "Solar-Energy) or" and "Exchange ) overall" should be "Exchange) overall". The manuscript requires thorough proofreading.
>
> **[RW4]** Thanks for pointing out these problems. We have proofread our paper to examine the typos and revise them.
>
> Overall, we sincerely thank the reviewer for the valuable suggestions mentioned above. Your feedback has significantly contributed to improving the quality of our manuscript. We hope that we have addressed your concerns and questions. We look forward to further discussions with you and hearing your new evaluations.
>
> [1] MambaTS: Improved Selective State Space Models for Long-term Time Series Forecasting, Arxiv
>
> [2] C-Mamba: Channel Correlation Enhanced State Space Models for Multivariate Time Series Forecasting, Arxiv
>
> [3] Time series modeling and forecasting with sample convolution and interaction, NeurIPS 2022
>
> [4] MICN:Multi-scale local and global context modeling for long-term series forecasting, ICLR 2023

---

> ### Author Response · Authors · 2024-11-25
> **[**Gentle Reminder**]: Kind Request for Reviewers' Feedback**
>
> Dear Reviewer y2RJ,
>
> Thank you once again for your valuable and constructive review, which has helped us refine our contribution and clarify its strengths.
>
> We would like to kindly remind you that the discussion deadline is approaching. After this deadline, we may not have the opportunity to respond to your comments.
>
> Additionally, during the rebuttal period, we have supplemented our study with **over 250 experimental results** to enhance the comprehensiveness of our experiments and the reliability of our conclusions. These results address your concerns regarding the experiments and are all included in $\underline{\text{the revised paper, highlighted in blue}}$.
>
> We sincerely appreciate your dedication and look forward to your feedback.
>
> Sincerely,
> ICLR 2025 Conference Submission 767 Authors,

---

> > ### Comment · Reviewer_y2RJ · 2024-11-25
> >
> > Thanks for the detailed response. My concerns have been partially addressed, so I have decided to maintain my original score. Below are the specific reasons:
> >
> > 1. From the response to question 1, I still do not understand the detailed design of the 'disentangled dependency encoding strategy.' The authors claim that they 'explore the direction of introducing distinct inductive biases...', however, **I could not find any specific design regarding 'inductive biases' in the paper** (by searching the keyword 'inductive biases'). In addition, there are some other Disentangled methods towards the time dimension and the variate dimension (e.g., TImeDRL [1]), the authors should demonstrate the advantages or differences of their methods compared to these existing methods.
> >
> > 2. I carefully read the comparison experiments regarding the nonlinear activation function, including the results in $\underline{\text{Appendix F.3 of revised paper}}$. First, the Exchange dataset is challenging, and it seems that even naive methods (without any parameters) perform better than deep learning-based methods [2]. I would argue that **we cannot be confident that experiments on the Exchange dataset can validate the effectiveness of removing nonlinear activation functions**. Second, the experimental results on Traffic and ETTm1 datasets show that **removing nonlinear activation functions in many variants leads to performance degradation**. This raises concerns about the necessity of removing nonlinear activation functions. In addition, the difference between SAMBA and the existing works is that the authors only remove nonlinear activation functions of Mamba, which further raises concerns about the novelty of the proposed method.
> >
> > 3. It seems that most of the experimental results are based on the reproduced results from the Time Series Library. I wonder why are the results for other methods almost identical to those in the paper, while the results for CARD differ significantly. Why didn’t the author directly use the results from the CARD paper? In addition, **the experimental results need to be carefully verified**. For example, the MSE results of iTransformer on ETTm2 dataset should change from 0.246 to 0.250, when the prediction length is 192.
> >
> >
> > [1] Chang C, Chan C T, Wang W Y, et al. TimeDRL: Disentangled Representation Learning for Multivariate Time-Series. ICDE, 2024.
> >
> > [2] Hewamalage H, Ackermann K, Bergmeir C. Forecast Evaluation for Data Scientists: Common Pitfalls and Best Practices. DMKD, 2023.

---

> ### Author Response · Authors · 2024-11-27
> **Thanks for your response [Part 1]**
>
> We appreciate the reviewer’s feedback, and we will address their points individually.
>
> >[W1-1]  I could not find any specific design regarding 'inductive biases' in the paper
>
> **[RW1-1]** In fact, **inductive bias is a fundamental concept in machine learning and academic terminology.** The inductive bias (also known as learning bias) of a learning algorithm refers to the set of assumptions that the learner uses to predict the outputs of given inputs it has not encountered before [1]. For instance, the inductive bias of CNNs includes translation invariance, as well as the assumption that local regions in the input data typically contain meaningful information. In our case, we used the term "inductive bias" to more clearly illustrate our architectural design principle in contrast to DRL, which we will ensure to include in the revised version of our paper. Similar to GNNs, which prioritize neighboring nodes for target node prediction, the two encoders in our strategy assume that **cross-time or cross-variate dependencies are more prominent, which naturally leads to their respective encoding dimensions.** Incorporating these biases enables the encoders to concentrate on distinct dependencies without mutual interference.
>
> >[W1-2] there are some other Disentangled methods towards the time dimension and the variate dimension (e.g., TImeDRL [1]), the authors should demonstrate the advantages or differences of their methods compared to these existing methods.
>
> **[RW1-2]** In fact, **the paper provided by the reviewer does not focus on the decoupling of temporal and variable dimensions.** Instead, it introduces a concept inspired by the [CLS] token in BERT to design a timestamp-level embedding and an instance-level embedding, thereby achieving the decoupling of local representation and global representation. This is entirely different from the disentangling of temporal and variable dimensions that we proposed. The difference between our work and** the most closely related CARD **has already been discussed in the $\underline{\text{Related Work section of original paper}}$.
>
> >[W2-1] I would argue that we cannot be confident that experiments on the Exchange dataset can validate the effectiveness of removing nonlinear activation functions. Second, the experimental results on Traffic and ETTm1 datasets show that removing nonlinear activation functions in many variants leads to performance degradation.
>
>  **[RW2]** First, **I did not find any reference in the materials [2] suggesting that non-deep learning methods perform better than deep learning methods in Exchange dataset.** Secondly, we provided the simplest linear layer approach, which is essentially a linear estimation model. The experimental results show that its performance is worse than that of the Patch+Transformer and Patch+Mamba, which contradicts the conclusion the reviewer has suggested. Third, the Exchange dataset is already widely used in the LTSF field, indicating that the dataset's value has been recognized in this area. We have provided an explanation for the performance decline in many variants. Furthermore, the continuous performance improvement observed with Mamba after removing the activation functions underscores the importance of this approach.
>
> >[W2-2]  In addition, the difference between SAMBA and the existing works is that the authors only remove nonlinear activation functions of Mamba, which further raises concerns about the novelty of the proposed method.
>
> **[RW2-2]** As noted by the reviewer in [W1-2] that **we also proposed a model-agnostic disentangled encoding method with theoretical guarantees, which is one of the key contributions of our work.** The reviewer's assessment of the novelty of our approach should not overlook this method. Unlike previous methods for introducing cross-variable dependencies that lacked theoretical explanation, we provide theoretical proof showing that our disentangled encoding approach is more effective. This lays a theoretically grounded path for future research in this area, encouraging further developments along this line.
>
> [1] The Need for Biases in Learning Generalizations

---

> ### Author Response · Authors · 2024-11-27
> **Thanks for your response [Part 2]**
>
> >[W3-1]  I wonder why are the results for other methods almost identical to those in the paper, while the results for CARD differ significantly. Why didn’t the author directly use the results from the CARD paper?
>
> **[RW3-1]** As stated in $\underline{\text{Section B.2 of original paper}}$, **CARD uses a different loss function during training compared to other baselines.** Comparing baselines with different loss functions in the same table is not fair. To ensure fairness, we utilized the official implementation of CARD and trained and evaluated it under the same time series setting to maintain consistency.
>
> >[W3-2] In addition, the experimental results need to be carefully verified. For example, the MSE results of iTransformer on ETTm2 dataset should change from 0.246 to 0.250, when the prediction length is 192.
>
> **[RW3-2]** We appreciate the reviewer for pointing out the typo. We conducted a thorough review of the experimental results. **The error only affects the results of iTransformer on the ETTm2 dataset,** but the average results for this dataset remain correct. Therefore, **this does not affect the validity of our conclusion that SAMBA achieved state-of-the-art performance.**

---

### Official Review · Reviewer_WkCP · 2024-11-03

**Soundness:** 2
**Presentation:** 1
**Contribution:** 3
**Rating:** 6
**Confidence:** 2

**Summary:**

This paper:

1. Identifies three types of dependencies in multivariate time series data.
2. Simplifies the Mamba activation functions to eliminate non-linearity.
3. Proposes a dependency encoding strategy to disentangle these dependencies, minimizing interference between the time and variate dimensions.

**Strengths:**

1. The authors provide a clear and comprehensive description of the key dependencies in LTSF modeling, including order dependency, semantic dependency, and cross-variate dependency. The model architecture is well-designed to capture these dependencies.

2. The architecture introduced in this paper to capture relationships across both temporal and variate dimensions enhances Mamba's ability to effectively model cross-variate relationships.

3. The experiments conducted demonstrate that the proposed architecture outperforms previous methods and can be effectively transferred to other models.

**Weaknesses:**

1. The paper contains several spelling and formatting errors. The authors should carefully check them before submission. Examples include:
   a) Inconsistent abbreviations, with “LTSF” sometimes written as “LTST”;
   b) Formatting issues in Theorem 1;
   c) Various spelling errors.

2. The title and model name "Simplified Mamba, SAMBA" emphasize a simplification of Mamba, specifically through the removal of non-linearity. However, according to the paper, the performance improvements from this regularization method are not particularly significant compared to other strategies, such as patch tokenization. Additionally, the authors mention that the elimination is not fully implemented, as non-linear activation function remains in the gating mechanism. Furthermore, the overall SAMBA architecture actually employs Mamba along the temporal dimension and bidirectional Mamba along the variate dimension, making it more complex rather than simplified, compared to the original Mamba for sequence modeling. Thus, the authors might consider rebranding their methods to avoid misleading readers.

**Questions:**

1. About order dependency:

The statement and evidences to prove that Transformer-based models are unsuitable for modeling order information seems insufficient.

First, improving a model's perception of order information can depend on two factors: the tokenization and the training method. In tokenization, if different lag timesteps are treated as distinct features inside a token (e.g., series embedding as in Linear models and iTransformer, or patch embedding as in PatchTST), shuffling values across timesteps will clearly impact performance, as the linear projection layer automatically considers them as separate input features. However, if timestep values are projected into the same latent space during tokenization, the model must rely on additional structures (such as attention + positional encoding) to learn this order information. Can this order be learned with pure Transformers? The success of LLMs proves that models can learn precise positional information. LLMs accurately identify and utilize context positions for next-token prediction without making sequencing errors, thanks to the learning of mechanisms like induction heads, which has been widely validated. Autoregressive training method enforces models to learn these temporal ordering algorithms.

The choice of tokenization and training method is largely independent of the model architecture. Both Transformers and Mamba can use these methods to improve the capture of positional information in LTSF tasks. Experimental results in the paper also show that patch tokenization enhances performance across different architectures. It separates timesteps within a local period into distinct features inside a token, rather than treating all timesteps as equivalent tokens in the attention mechanism.

Constructing algorithms between tokens in attention (or SSM in Mamba) involves learning the sequence's semantic information. The richer the semantic information, the more we need to model algorithms at a finer granularity (shorter patch sizes). The paper suggests that patching enhances semantic dependency learning, but this seems contradictory. Patching likely reduces the need for complex semantic relationship modeling, functioning as a form of attention regularization that improves performance on datasets with simpler semantics, contrary to Assumption 2. Correspondingly, Linear models treat the entire series as embeddings and all timesteps as lag features, omitting the need to construct semantic relationships between timesteps, which aligns with the statement in the paper. However, the analysis of patching seems inconsistent. The authors could experiment on datasets with richer semantic information (e.g., ODE-based datasets with underlying dynamics) to see if pure timestep embedding outperforms patch embedding.

In summary, does Mamba have an advantage over Transformer-based models in modeling order information as claimed? It seems so, but this advantage may come from patching tokenization and possibly from Mamba's SSM handling of sequential/causal token information. Previous LTSF encoder-only Transformers may lack the sequential structure. The authors could compare Mamba to Transformers using causal masks/training methods to further establish Mamba's structural superiority in LTSF. Additionally, more discussion is needed to support the definitions and statements of semantic dependency.


2. About the dataset selection：

The experiments in Tables 1-3 and Figure 1 are conducted solely on the ETTm1 dataset, which seems insufficient to draw general conclusions. The nature of the datasets significantly impacts the performance of different models with varying degrees of regularization.

The performance comparison in Table 1 using shuffling on a single dataset appears insufficient (actually, I hold a similar view regarding these experiments conducted in the DLinear paper). Here's a simple counterexample:

Consider an invertible MA(1) process, $ X_t = \mu + \varepsilon_t + \theta \varepsilon_{t-1} $, with two sub-optimal predictors: one using the global average of the input and the other using the last value as the prediction. It can be shown that the first predictor is better or equal to the second one under invertible condition. However, if the input is shuffled, the first predictor’s performance does not deteriorate, while the second, less optimal predictor's MSE decreases when $ \theta > 0 $, remains unchanged at $ \theta = 0 $, and increases when $ \theta < 0 $. This indicates that comparing performance drops due to shuffling may not be sufficient to prove a model’s sensitivity to order, as better models may also learn predictors insensitive to position.

The conclusions from Table 2 may also be dataset-sensitive. For datasets with underlying dynamics and shifting multivariate effects, exposing more temporal tokens for algorithm construction in attentions could be more advantageous. The authors could extend experiments to datasets like Solar and Exchange to strengthen these claims.

Table 3 should be a foundational point of the paper, but it requires validation across more datasets. The choice between model linearity and complexity largely depends on the dataset. When clear non-linear relationships exist, the model may need to build more complex algorithms between temporal tokens, leading to different conclusions.

Figure 1 shows that removing activation functions increases the model’s regularization. The authors could consider testing this on more complex datasets, such as Traffic (or maybe PEMS?), to confirm that stronger regularization is indeed beneficial.


### Conclusion

Even though I have some concerns about the authors’ claims and experimental methods, the authors have put in a substantial amount of work throughout the paper. Therefore, I look forward to further discussion with the authors on these points before giving a final score.

---

> ### Author Response · Authors · 2024-11-21
> **Response to Reviewer WkCP [Part 1]**
>
> Dear Reviewer WkCP,
>
> We would like to sincerely thank you for providing a detailed review and insightful suggestions, particularly on the discussion about patches and suggestions regarding experimental setups. **Following your suggestions, we conducted over 500 additional experiments and reported new results for more than 250 of them. We hope these efforts lead to a reassessment of our manuscript.** Below, we provide detailed, point-by-point responses to your concerns.
>
> >[W1] The paper contains several spelling and formatting errors. The authors should carefully check them before submission. Examples include: a) Inconsistent abbreviations, with “LTSF” sometimes written as “LTST”; b) Formatting issues in Theorem 1; c) Various spelling errors.
>
> **[RW1]**  Thanks for pointing out these problems. We have revised them, and We have proofread our paper to examine the typos and revise them.
>
> >[W2-1] However, according to the paper, the performance improvements from removal of non-linearity regularization method are not particularly significant compared to other strategies, such as patch tokenization.
>
> **[RW2-1]** Thank you for the reviewers' valuable suggestions. Eliminating non-linear activation functions in Mamba is an effective and simple regularization method that also yields significant performance improvements. As shown in $\underline{\text{Table 3}}$, directly removing non-linear activation functions results in a 5.79% improvement in the MSE metric. Supplementary experiments measured on more datasets, provided in $\underline{\text{Appendix F of revised paper}}$, are consistent with this conclusion. Moreover, even with the use of patch operations, our ablation experiments demonstrate that this is an effective regularization method for Mamba.
>
> >[W2-2]  Elimination is not fully implemented, as non-linear activation function remains in the gating mechanism.
>
> **[RW2-2]** Thank you for pointing out the confusion. As stated in $\underline{\text{Lines 351 and 352}}$, the non-linearity in the gating mechanism is designed to **maintain learning stability and robustness, rather than to learn complex representations. Removing the non-linear activation function here would disrupt the gating mechanism.** Therefore, we retained this non-linearity.
>
> >[W2-3]  the overall SAMBA architecture it more complex rather than simplified, compared to the original Mamba for sequence modeling. Thus, the authors might consider rebranding their methods to avoid misleading readers.
>
> **[RW2-3]** Thank you for the reviewers' valuable suggestions. SAMBA refers to the Simplified Mamba Block. Regarding the structure of the overall model, we will consider renaming it to avoid misleading interpretations.
>
> >[Q1] About order dependency:
>
> **[RW1]** We deeply appreciate and thank the reviewer for raising this intriguing perspective for discussion. Let us elaborate and discuss further:
>
> >[Q1-1] Can this order be learned with pure Transformers? The success of LLMs proves that models can learn precise positional information.
>
> **[RQ1-1]**  Indeed, existing works have shown that LLMs can learn the order of textual sequences, but no work has yet demonstrated that LLMs can capture order dependency in time series. A recent study [1] suggests that **"LLMs do not have unique capabilities for representing sequential dependencies in time series."** Furthermore, they discovered that "LLMs fail to convincingly improve time series forecasting." **These findings indicate that the success of LLMs learning order dependency on text cannot be directly transferred to time series data.** Similarly, our experimental results in $\underline{\text{Table 3}}$ and the $\underline{\text{Appendix F.1 of revised paper}}$ show that transformers struggle to effectively learn the order dependencies in time series data. This may be due to the low semantic density of time series data. However, we believe that exploring how LLMs can leverage the characteristics of time series effectively is a promising research direction.

---

> ### Author Response · Authors · 2024-11-21
> **Response to Reviewer WkCP [Part 2]**
>
> >[Q1-2] Constructing algorithms between tokens in attention (or SSM in Mamba) involves learning the sequence's semantic information. The richer the semantic information, the more we need to model algorithms at a finer granularity (shorter patch sizes). The paper suggests that patching enhances semantic dependency learning, but this seems contradictory.
>
> **[RQ1-2]** The reviewer raised a valuable discussion regarding patching. We agree with the notion that richer semantics require finer-grained modeling. **This statement does not conflict with our Assumption 2.** The former refers to data that inherently has strong semantic information, such as text, where its rich semantics require fine-grained processing. In contrast, the time series data we study are semantically sparse [2][3]. As noted in PatchTST [3], "A single time step does not have semantic meaning like a word in a sentence; thus, extracting local semantic information is essential in analyzing their connections." **Hence, Patches in time series data enhance locality and capture comprehensive semantic information that cannot be achieved at the point level by aggregating time steps into subseries-level patches [3].** Therefore, our Assumption 2 is reasonable.
>
> >[Q1-3] The authors could experiment on datasets with richer semantic information (e.g., ODE-based datasets with underlying dynamics) to see if pure timestep embedding outperforms patch embedding.
>
> **[RQ1-3]** Regarding the comparison of patch embeddings versus pure time steps on datasets with strong ODE dynamics, this is an intriguing discussion, and we appreciate the reviewer's deep insights into this area. **Existing work [4] demonstrates that even on datasets with strong ODE dynamics, patch embeddings still achieve significant performance improvements.** This aligns with our previous discussions on the semantic sparsity of time series. However, it is worth noting that this work also shows that **datasets with strong ODE dynamics are sensitive to patch size**. Larger patches may harm performance, which is consistent with the reviewer's speculation. Overall, this work supports findings from previous studies and aligns with our Assumption 2 while reflecting the reviewer's valuable insight: **the semantics of individual time series points are sparse, and using appropriately sized patches can enhance the semantic richness of the data.** However, excessively large patches may disrupt the sequence's semantic information.
>
> >[Q1-4] In summary, does Mamba have an advantage over Transformer-based models in modeling order information as claimed? It seems so, but this advantage may come from patching tokenization and possibly from Mamba's SSM handling of sequential/causal token information. The authors could compare Mamba to Transformers using causal masks/training methods to further establish Mamba's structural superiority in LTSF.
>
> **[RQ1-4]** Thank you for the reviewer's discussion on whether Mamba can better model order dependencies. When conducting the experiments in Table 1, we ensured that both Mamba and the Transformer used the same tokenization method as suggested by the Time Series Library GitHub repository. **This ensured fairness in the experiments** and avoided any advantage of Mamba in capturing order dependencies stemming from unfair tokenization methods. As a result, the advantage is attributable to the recursive processing of the SSM compared to attention mechanisms. We also tested the performance of a transformer trained using a causal-masked autoregressive approach on the Exchange dataset. **The experimental results show that causal transformer is more sensitive to order than a standard transformer but still underperforms Mamba.**
>
> | Datasets     | Prediction Length | Mamba  |        |      |        | Transformer| |       |           | Causal Transformer|    |    |      |
> |--------------|--------------------|-------|-------|-------|--------|------|-------|-------|----------|------|------|-------|---------------|
> |              |                    | O.MSE | S.MSE | O.MAE | S.MAE | O.MSE | S.MSE | O.MAE | S.MAE  | O.MSE | S.MSE | O.MAE | S.MAE  |
> | **Exchange** | 96                 | 1.260 | 1.401 | 0.915 | 0.943 | 0.730 | 0.738 | 0.782 | 0.722  | 0.570 | 0.584 | 0.610 | 0.629  |
> |              | 192                | 1.398 | 1.626 | 1.040 | 1.060 | 1.304 | 1.284 | 0.913 | 0.949  | 1.182 | 1.259 | 0.918 | 0.938  |
> |              | 336                | 1.835 | 1.921 | 1.111 | 1.141 | 1.860 | 1.862 | 1.090 | 1.085  | 1.405 | 1.445 | 0.943 | 0.945  |
> |              | 720                | 3.940 | 4.023 | 1.687 | 1.697 | 3.860 | 3.865 | 1.684 | 1.685  | 3.532 | 3.605 | 0.698 | 0.837  |
> |              | **Avg. Drop**      | -     | 6.38% | -     | 1.85% | -     | -0.06% | -     | -0.63% | -     | 3.05% | -     | 1.35% |

---

> ### Author Response · Authors · 2024-11-21
> **Response to Reviewer WkCP [Part 3]**
>
> >[Q1-5] Additionally, more discussion is needed to support the definitions and statements of semantic dependency.
>
> **[RQ1-5]**  Thank you for the suggestions. We have included additional discussions on the differences between order dependency and semantic dependency in the $\underline{\text{Appendix P of revised paper}}$.
>
> >[Q2] About the dataset selection
>
> **[RQ2]** Thank you for your suggestions on improving the comprehensiveness of our experiments and for providing clear and intuitive examples to illustrate the limitations. Following your recommendations, we have extended the experiments in Section 4 to include the Solar, Exchange, and Traffic datasets. Below, we address your concerns and suggestions point by point:
>
> >[Q2-1] The performance comparison in Table 1 using shuffling on a single dataset appears insufficient (actually, I hold a similar view regarding these experiments conducted in the DLinear paper).
>
> **[RQ2-1]** We present the results of shuffling on the Exchange dataset. The results indicate that, compared to Linear and Mamba, which show significant performance degradation, the Transformer even exhibits a slight performance improvement. **This suggests that Linear and Mamba are capable of capturing order dependencies, while Transformers struggle in this regard.** The results of other datasets and more detailed discussions can be found in $\underline{\text{Appendix F.1 of revised paper}}$.
>
> | Datasets     | Prediction Length | Linear Model |        |       |         | Mamba |        |        |          | Transformer |      |      |         |
> |--------------|--------------------|--------------|--------|-------|---------|-------|--------|--------|----------|-------------|------|------|---------|
> |              |                    | O.MSE        | S.MSE  | O.MAE | S.MAE   | O.MSE | S.MSE  | O.MAE  | S.MAE    | O.MSE       | S.MSE| O.MAE| S.MAE   |
> | **Exchange** | 96                 | 0.0832       | 0.210  | 0.201 | 0.332   | 1.260 | 1.401  | 0.915  | 0.943    | 0.730       | 0.738| 0.782| 0.722   |
> |              | 192                | 0.179        | 0.325  | 0.299 | 0.414   | 1.398 | 1.626  | 1.040  | 1.060    | 1.304       | 1.284| 0.913| 0.949   |
> |              | 336                | 0.338        | 0.521  | 0.418 | 0.534   | 1.835 | 1.921  | 1.111  | 1.141    | 1.860       | 1.862| 1.090| 1.085   |
> |              | 720                | 0.903        | 1.167  | 0.714 | 0.822   | 3.940 | 4.023  | 1.687  | 1.697    | 3.860       | 3.865| 1.684| 1.685   |
> |              | **Avg. Drop**      | -            | 47.89% | -     | 28.80%  | -     | 6.38%  | -      | 1.85%    | -           | -0.06%|-     |-0.63%   |
>
> *Note: O.MSE and O.MAE are evaluated in the original test set. S.MSE and S.MAE are evaluated in the shuffling test set.*

---

> ### Author Response · Authors · 2024-11-21
> **Response to Reviewer WkCP [Part 4]**
>
> >[Q2-2] The conclusions from Table 2 may also be dataset-sensitive. For datasets with underlying dynamics and shifting multivariate effects, exposing more temporal tokens for algorithm construction in attentions could be more advantageous. The authors could extend experiments to datasets like Solar and Exchange to strengthen these claims.
>
> **[RQ2-2]** Following your suggestion, we measured the results on the Solar and Exchange datasets. **On the Exchange dataset, patching improves model performance,** aligning with the conclusions in Table 2. However, on the Solar dataset, patching leads to a performance decline, **possibly due to the high proportion of zero values in Solar,** which disrupts the intrinsic semantic information of individual data points when patching is applied. The results of other datasets and more detailed discussions can be found in the $\underline{\text{Appendix F.2 of revised paper}}$.
> Overall, the additional experimental results support our conclusions in Table 2 and align with discussions in [WQ1-3], demonstrating that on datasets with ODE dynamics, **patching enhances the semantic richness of the data, enabling semantic-aware models like Transformers and Mamba to achieve better performance.**
>
> | Dataset       | Prediction Length   | Linear Model |      | Patch+Linear Model |      | Mamba |      | Patch+Mamba |      | Transformer |      | Patch+Transformer |      |
> |---------------|--------------|--------------|------|---------------------|------|-------|------|--------------|------|-------------|------|-------------------|------|
> |               |              | MSE          | MAE  | MSE                 | MAE  | MSE   | MAE  | MSE          | MAE  | MSE         | MAE  | MSE               | MAE  |
> | **Exchange**  | 96           | 0.0832       | 0.201| 0.0823              | 0.207| 1.260 | 0.915| 0.0871       | 0.207| 0.989       | 0.782| 0.0861            | 0.204|
> |               | 192          | 0.179        | 0.299| 0.165               | 0.302| 1.398 | 1.040| 0.176        | 0.298| 1.265       | 0.913| 0.183             | 0.303|
> |               | 336          | 0.338        | 0.418| 0.285               | 0.401| 1.835 | 1.111| 0.327        | 0.413| 1.860       | 1.090| 0.332             | 0.417|
> |               | 720          | 0.903        | 0.714| 0.799               | 0.685| 3.940 | 1.687| 0.853        | 0.694| 3.860       | 1.684| 0.854             | 0.697|
> |               | **Avg**      | 0.376        | 0.408| 0.333               | 0.399| 2.254 | 1.188| 0.361        | 0.403| 1.993       | 1.117| 0.364             | 0.405|
> | **Solar-Energy** | 96        | 0.326        | 0.346| 0.357               | 0.439| 0.190 | 0.248| 0.204        | 0.243| 0.201       | 0.269| 0.218             | 0.264|
> |               | 192          | 0.363        | 0.364| 0.373               | 0.446| 0.224 | 0.292| 0.237        | 0.265| 0.233       | 0.289| 0.250             | 0.284|
> |               | 336          | 0.402        | 0.378| 0.395               | 0.454| 0.315 | 0.354| 0.254        | 0.277| 0.232       | 0.294| 0.271             | 0.300|
> |               | 720          | 0.402        | 0.368| 0.393               | 0.445| 0.293 | 0.295| 0.254        | 0.278| 0.216       | 0.280| 0.271             | 0.295|
> |               | **Avg**      | 0.373        | 0.364| 0.380               | 0.446| 0.217 | 0.288| 0.237        | 0.266| 0.220       | 0.283| 0.252             | 0.286|

---

> ### Author Response · Authors · 2024-11-21
> **Response to Reviewer WkCP [Part 5]**
>
> >[Q2-3] Table 3 should be a foundational point of the paper, but it requires validation across more datasets.
>
> **[RQ2-3]**  Following your suggestion, we expanded the scope of our experiments to include the Exchange and Traffic datasets. **The table below demonstrates that removing non-linear activation functions benefits both Mamba and Transformer models** (more results are included in the $\underline{\text{Appendix F.3 of revised paper}}$). However, the limited improvements on the Traffic dataset, along with the performance decline of MLP, are attributed to the stronger semantic complexity and the involvement of more non-linear relationships in the Traffic dataset. The simplified architecture of MLP, without non-linear activation functions, is unable to effectively handle such complexity. In contrast, the performance improvements observed for Transformer and Mamba suggest that their inherently complex architectures are already capable of handling these relationships, making non-linear activation functions redundant. Training curves in $\underline{\text{Appendix F.3 of revised paper}}$ further support this conclusion. **The ablation experiments in Section 6.2 further support the effectiveness of removing the nonlinear activation function in Mamba.**
> | Datsets     | Model        | MLP    |        | Mamba  |        | Transformer |        |
> |--------------|---------------|--------|--------|--------|--------|-------------|--------|
> |              |               | MSE    | MAE    | MSE    | MAE    | MSE         | MAE    |
> | **Exchange** | Original      | 0.398  | 0.419  | 2.255  | 1.189  | 1.994       | 1.117  |
> |              | Original-n    | 0.374  | 0.407  | 2.122  | 1.143  | 1.194       | 0.895  |
> |              | Improvement   | 6.03%  | 2.86%  | 5.90%  | 3.79%  | 40.13%      | 19.87% |
> | **Traffic**  | Original      | 0.554  | 0.366  | 0.669  | 0.385  | 0.833       | 0.480  |
> |              | Original-n    | 0.621  | 0.400  | 0.658  | 0.381  | 0.829       | 0.479  |
> |              | Improvement   | -12.27%| -9.28% | 1.57%  | 0.98%  | 0.42%       | 0.16%  |
>
> >[Q2-4] Figure 1 shows that removing activation functions increases the model’s regularization. The authors could consider testing this on more complex datasets, such as Traffic (or maybe PEMS?), to confirm that stronger regularization is indeed beneficial.
>
> **[RQ2-4]** Our Training curves in $\underline{\text{Appendix F.3 of revised paper}}$ confirm that removing non-linear activation functions is beneficial, further validating the proposed approach.
>
> Overall, we sincerely thank the reviewer for the valuable suggestions mentioned above. Your feedback has significantly contributed to improving the quality of our manuscript. We hope that we have addressed your concerns and questions. We look forward to further discussions with you and hearing your new evaluations.
>
> - [1] Are Language Models Actually Useful for Time Series Forecasting?, NeurIPS 2024
> - [2] Are Transformers Effective for Time Series Forecasting?, AAAI 2023
> - [3] A Time Series is Worth 64 Words: Long-term Forecasting with Transformers, ICLR 2023
> - [4] Irregular Multivariate Time Series Forecasting: A Transformable Patching Graph Neural Networks Approach, ICML 2024

---

> ### Author Response · Authors · 2024-11-25
> **[**Gentle Reminder**]: Kind Request for Reviewers' Feedback**
>
> Dear Reviewer WkCP,
>
> Thank you once again for your valuable and constructive review, which has helped us refine our contribution and clarify its strengths.
>
> We would like to kindly remind you that the discussion deadline is approaching. After this deadline, we may not have the opportunity to respond to your comments.
>
> Additionally, during the rebuttal period, we have supplemented our study with ***over 250 experimental results*** to enhance the comprehensiveness of our experiments and the reliability of our conclusions. These results address your concerns regarding the experiments and are all included in $\underline{\text{the revised paper, highlighted in blue}}$.
>
> We sincerely appreciate your dedication and look forward to your feedback.
>
> Sincerely,
> ICLR 2025 Conference Submission 767 Authors

---

> > ### Comment · Reviewer_WkCP · 2024-11-27
> >
> > Thanks to the authors for their detailed responses. The authors have conducted sufficient experiments and further analyses. From the beginning, I acknowledged the amount of work put into the paper. Even though we might have differing opinions on the practical value of Mamba in time series applications, I believe that the wealth of information and insights provided by these extensive experiments can be further assessed by our community. Therefore, I have decided to raise the score to 6.

---

> > > ### Author Response · Authors · 2024-11-27
> > > **Thanks for your response**
> > >
> > > We sincerely appreciate your positive evaluation and recognition of the contributions of this work. This study represents our assessment of Mamba's potential in the LTSF domain, and we are grateful for the reviewers' acknowledgment of our efforts. We would also like to extend our thanks once again for the many constructive questions raised during the rebuttal process, which have significantly contributed to improving the quality of this work. If there are any remaining concerns or aspects where further clarification could enhance your evaluation, we would be more than happy to address them in greater detail.

---

### Official Review · Reviewer_qdrT · 2024-11-04

**Soundness:** 2
**Presentation:** 3
**Contribution:** 1
**Rating:** 3
**Confidence:** 4

**Summary:**

This paper investigates the adaptation of a recently emerged  Mamba architecture to long-term time-series forecasting. It highlights three key aspects: order, semantic, and cross-variate dependencies. The derived Samba architecture encompasses two branches for temporal and variable dependency encoding, of which the encoded representations are concatenated to produce forecasts.

**Strengths:**

It is interesting to investigate the pros and cons of the Mamba architecture, or more broadly, structured state-space models (SSMs), on long-term time-series forecasting.

**Weaknesses:**

Concerns Regarding the Soundness and Contributions of the Paper

1. Overlooking the Unique Advantage of Mamba: Modeling Long Sequences

Mamba, a recent and popular instantiation of state-space models (SSMs), is renowned for its efficiency in processing long sequences. It significantly reduces the computational overheads of self-attention in Transformers while seemingly maintaining long-term dependencies in various real-world datasets. However, this paper primarily compares time-series representation learning among MLP, Transformer, and Mamba, fixing the lookback length at T=96. When modeling relatively shorter sequences, the necessity of introducing SSMs diminishes significantly, which in turn reduces the impact of this research.

2. Inappropriate and Biased Expressions/Claims

Lines 014-016: The claim, "However, most approaches still struggle to comprehensively capture reliable and informative dependencies inherent in time series data," lacks rigor. Existing time-series forecasting models have made significant progress in learning effective time-series representations and deliver commendable forecasts in many scenarios. The paper's experiments do not show a substantial difference in model performance to support this claim.

Lines 052-054: The assertion that "they struggle with perceiving temporal order due to the permutation-invariant nature of self-attention, even with positional encodings," is debatable. It is not clear how sensitivity to sequence order impacts the capability of learning effective temporal representations. Extensive research on position encoding for Transformers has already led to the success of modern large language models. The experiments in Table 1 only demonstrate that Transformers are less sensitive to order permutation than Linear and Mamba, which is expected. However, it is unclear how this capability affects temporal representation learning. In fact, being less sensitive to order permutation might be advantageous in sequence modeling. Furthermore, the use of Mamba to model cross-variate dependency seems flawed, as time-series variates lack a ground-truth order. Following the author's logic, using an order-sensitive model like Mamba for cross-variate dependency might not be appropriate.

Lines 058-061: The statement, "existing approaches that utilize cross-variate dependency (Channel Dependent, CD) frequently underperform compared to methods that treat each variate independently (Channel-Independent, CI)," is not universally applicable. The performance of CD and CI approaches highly depends on specific cases. Studies, such as iTransformer, have demonstrated the benefits of cross-variate modeling.

3. Insufficient Experiments Leading to Unreliable Conclusions

The analyses in Section 4, which inform the design of Samba, are primarily based on experiments conducted on the small ETTm1 dataset and compared across basic neural architectures like Linear, vanilla MLP, and Transformer. These results can be interpreted in multiple ways, not solely as the authors suggest. For instance, Table 1 shows Transformers' reduced sensitivity to order perturbation, which may actually be beneficial for encoding effective temporal representations. Table 2 investigates patching effectiveness, originally introduced in PatchTST, yet PatchTST itself is not included in the analysis. Additionally, the impact of patch size on results and whether all datasets lead to the same conclusion are unexplored. Other factors, such as TSMixer extending MLP-based architectures on time-series and PatchTST's designs to prevent overfitting, are not considered in Table 3. Consequently, the analyses in Section 4 are unconvincing regarding the necessity of introducing Mamba.

4. Seemingly Downgraded Performance of Baselines

For example, in Table 4, PatchTST on ETTh1 produces an MSE of 0.469 and an MAE of 0.454. However, the original paper reported much lower errors (https://arxiv.org/pdf/2211.14730).

5. Lack of Unique Contributions

In addition to studying Mamba for time series, this paper appears to offer few unique contributions or novel insights. Concepts such as patching, cross-variate and cross-time dependency, and normalization to prevent overfitting have been extensively explored in prior research. Furthermore, Transformer variants in time series have effectively addressed challenges in these areas. Therefore, without the focus on modeling very long sequences, the necessity of adapting Mamba for time series remains questionable.

**Questions:**

See weaknesses.

---

> ### Author Response · Authors · 2024-11-21
> **Response to Reviewer qdrT [Part 1]**
>
> Dear Reviewer qdrT,
>
> We sincerely appreciate the reviewer’s valuable feedback. **Following your suggestions, we conducted over 500 additional experiments and reported new results for more than 250 of them. We hope these efforts lead to a reassessment of our manuscript.** Below, we provide detailed, point-by-point responses to your concerns.
>
> >[W1] Overlooking the Unique Advantage of Mamba: Modeling Long Sequences.
>
> **[RW1]**
> - First, we want to emphasize that **lookback window 96 is a common setting in the LTSF domain,** widely adopted by most works [1][2], and are also a standard setting for many Mamba for LTSF studies [3][4].
> - Second, rather than directly leveraging the advantages of Mamba already proven in other domains, **our work reveals an unexplored advantage of Mamba in the LTSF domain,** namely its ability to simultaneously capture order dependency and semantic dependency, which adds greater value to our research.
> - Moreover, **we do not deny Mamba's strengths in long-sequence modeling,** and we have already demonstrated in $\underline{\text{Appendix I of original paper}}$ that Mamba can effectively leverage longer lookback windows from 48 to 720.
>
> >[W2-1] The claim, "However, most approaches still struggle to comprehensively capture reliable and informative dependencies inherent in time series data," lacks rigor.
>
> **[RW2-1]** Based on the three types of dependencies we defined, we empirically find that existing Transformer and Linear models are unable to simultaneously capture both order dependency and semantic dependency (see Section 4 and the additional results on more datasets provided in the $\underline{\text{Appendix F of revised paper}}$). Therefore, models based on Transformers and Linear architectures cannot effectively capture these two dependencies simultaneously.
>
> >[W2-2] The assertion that "they struggle with perceiving temporal order due to the permutation-invariant nature of self-attention, even with positional encodings," is debatable.
>
> **[RW2-2]** The conclusion that "they struggle with perceiving temporal order due to the permutation-invariant nature of self-attention, even with positional encodings" is a viewpoint we cited from DLinear. In the DLinear abstract, the original statement is: "While employing positional encoding and using tokens to embed sub-series in Transformers facilitate preserving some ordering information, the nature of the permutation-invariant self-attention mechanism inevitably results in temporal information loss." Therefore, **this perspective is not a novel claim we are proposing but rather a conclusion that has been discovered by others.**
>
> >[W2-3]  Experiments in Table 1 only demonstrate that Transformers are less sensitive to order permutation than Linear and Mamba, which is expected. It is not clear how sensitivity to sequence order impacts the capability of learning effective temporal representations.
>
> **[RW2-3]** We appreciate the reviewers' possible inference regarding this conclusion. However, existing studies [5][6] suggest that **this effect is not unclear but rather explicit:** a good time series predictor should be sensitive to the order of the sequence. **This perspective is clearly stated in Section 5.3 of DLinear [5]: "in time series forecasting, the sequence order often plays a crucial role."** This view has also been acknowledged in recent work [6], which uses it to demonstrate that LLMs are unable to effectively learn the order dependency in time series.
>
> >[W2-4] use of Mamba to model cross-variate dependency seems flawed, as time-series variates lack a ground-truth order. Following the author's logic, using an order-sensitive model like Mamba for cross-variate dependency might not be appropriate.
>
> **[RW2-4]** As stated in $\underline{\text{Appendix C of original paper, "Why Choose Mamba To Encode Cross-variate Dependency?"}}$, using sequential models to process unordered sequences is feasible, as evidenced by using LSTM or Mamba to encode unordered graph neighbors [7][8]. Additionally, considering the unordered nature of variable sequences, we designed a bidirectional Mamba to encode them effectively.
>
> >[W2-5] Statement "existing approaches that utilize cross-variate dependency (Channel Dependent, CD) frequently underperform compared to methods that treat each variate independently (Channel-Independent, CI)," is not universally applicable. The performance of CD and CI approaches highly depends on specific cases. Studies, such as iTransformer, have demonstrated the benefits of cross-variate modeling.
>
> **[RW2-5]** We kindly request reviewers not to overlook the context surrounding this statement. In the following sentence, we emphasize that iTransformer largely addresses this issue but still fails to account for the interplay between the temporal and variable dimensions. Therefore, we propose a disentangled encoding approach to model cross-variable dependencies more effectively, which is also theoretically validated.

---

> ### Author Response · Authors · 2024-11-21
> **Response to Reviewer qdrT [Part 2]**
>
> >[w3] Insufficient Experiments Leading to Unreliable Conclusions
>
> >[W3-1] The analyses in Section 4, which inform the design of SAMBA, are primarily based on experiments conducted on the small ETTm1 dataset and compared across basic neural architectures like Linear, vanilla MLP, and Transformer.
>
> **[RW3-1]** Thank you for the valuable suggestion regarding the comprehensiveness of our experiments. To address the reviewer’s concerns, we have extended the $\underline{\text{Section 4}}$ experiments to include the more datasets. According to your suggestions, we included the latest SOTA model, iTransformer, which uses a Linear Model to encode temporal relationships and is therefore expected to be sensitive to time order.  **The following experimental results demonstrate that both Linear and Mamba effectively capture order dependency.** However, the following results reveal that even Transformer exhibits improved performance on the Exchange dataset after shuffling, indicating that Transformer models fail to learn the critical order dependency in time series. The results of other datasets and more detailed discussions can be found in $\underline{\text{Appendix F of revised paper}}$.
>
> | Datasets     | Prediction Length | Linear Model |        |       |         | Mamba |        |        |          | Transformer |      |      |         | iTransformer|      |       |             |
> |--------------|--------------------|--------------|--------|-------|---------|-------|--------|--------|----------|-------------|------|------|---------|-------------|------|-------|-------------|
> |              |                    | O.MSE        | S.MSE  | O.MAE | S.MAE   | O.MSE | S.MSE  | O.MAE  | S.MAE    | O.MSE       | S.MSE| O.MAE| S.MAE   | O.MSE       | S.MSE| O.MAE | S.MAE       |
> | **ETTm1**    | 96                 | 0.383        | 0.988  | 0.400 | 0.697   | 0.517 | 0.922  | 0.508  | 0.688    | 0.643       | 0.884| 0.575| 0.643   | 0.345       | 0.892| 0.378 | 0.610       |
> |              | 192                | 0.413        | 0.986  | 0.415 | 0.697   | 0.575 | 0.931  | 0.546  | 0.699    | 0.805       | 1.01 | 0.664| 0.730   | 0.383       | 0.903| 0.395 | 0.617       |
> |              | 336                | 0.441        | 0.987  | 0.435 | 0.698   | 0.730 | 0.957  | 0.634  | 0.703    | 0.882       | 1.12 | 0.737| 0.817   | 0.423       | 0.923| 0.420 | 0.630       |
> |              | 720                | 0.497        | 0.992  | 0.469 | 0.704   | 0.873 | 0.973  | 0.704  | 0.723    | 0.928       | 1.12 | 0.752| 0.800   | 0.489       | 0.932| 0.456 | 0.641       |
> |              | **Avg. Drop**      | -            | 127.97%| -     | 62.55%  | -     | 40.37% | -      | 17.60%   | -           | 22.40%|-     | 6.55%   | -           | 122.56%|-     | 0.515%      |
> | **Exchange** | 96                 | 0.0832       | 0.210  | 0.201 | 0.332   | 1.260 | 1.401  | 0.915  | 0.943    | 0.730       | 0.738| 0.782| 0.722   | 0.0869      | 0.242| 0.207 | 0.358       |
> |              | 192                | 0.179        | 0.325  | 0.299 | 0.414   | 1.398 | 1.626  | 1.040  | 1.060    | 1.304       | 1.284| 0.913| 0.949   | 0.179       | 0.374| 0.301 | 0.450       |
> |              | 336                | 0.338        | 0.521  | 0.418 | 0.534   | 1.835 | 1.921  | 1.111  | 1.141    | 1.860       | 1.862| 1.090| 1.085   | 0.331       | 0.535| 0.417 | 0.557       |
> |              | 720                | 0.903        | 1.167  | 0.714 | 0.822   | 3.940 | 4.023  | 1.687  | 1.697    | 3.860       | 3.865| 1.684| 1.685   | 0.856       | 1.202| 0.698 | 0.841       |
> |              | **Avg. Drop**      | -            | 47.89% | -     | 28.80%  | -     | 6.38%  | -      | 1.85%    | -           | -0.06%|-     |-0.63%   | -           | 63.33%|-     | 35.89%      |
>
> *Note: O.MSE and O.MAE are evaluated in original test set. S.MSE and S.MAE are evaluated in shuffling test set.*
>
> >[W3-2] For instance, Table 1 shows Transformers' reduced sensitivity to order perturbation, which may actually be beneficial for encoding effective temporal representations
>
> **[RW3-2]** We appreciate the reviewers for presenting an interesting perspective, **but this perspective contradicts the current work [5][6].** In $\underline{\text{Section 4.1}}$, our work adopts the same settings as DLinear [5] to demonstrate that Mamba effectively captures order dependency. **The importance of effectively capturing order dependency for accurate time series forecasting has already been proven by DLinear.** This view has also been acknowledged in recent work [6], which uses it to demonstrate that LLMs are unable to effectively learn the order dependency in time series.
>
> >[W3-3] Table 2 investigates patching effectiveness, originally introduced in PatchTST, yet PatchTST itself is not included in the analysis.
>
> **[RW3-3]** We have already clarified in $\underline{\text{Appendix B.4 (Implementation) of original paper}}$  that "Patch + Transformer" is PatchTST.

---

> ### Author Response · Authors · 2024-11-21
> **Response to Reviewer qdrT [Part 3]**
>
> >[W3-4] Additionally, the impact of patch size on results and whether all datasets lead to the same conclusion are unexplored.
>
> **[RW3-4]**  Thank you for the valuable suggestion regarding the comprehensiveness of our experiments. We follow your suggestion to analyze the impact of patch size. **The results are shown in the table below, and our conclusion is that removing nonlinear activation functions helps mitigate overfitting and is robust to patch size.** Results from additional datasets also align with our conclusion; please refer to $\underline{\text{Appendix F of revised paper}}$ for more details.
> | Dataset       | patch size   | Patch+MLP         |            | Patch+MLP-n           |            |Patch+Mamba            |            | Patch+Mamba-n       |            | Patch+Transformer           |   |Patch+ Transformer-n            |            |
> |---------------|-----|----------------|------------|------------|----------------|------------|------------|----------------|------------|------------|----------------|------------|------------|
> |  |     | MSE | MAE  | MSE | MAE            | MSE        | MAE | MSE            | MAE        | MSE | MAE            | MSE        | MAE |
> | **ETTm1**   | 2  | 0.429      | 0.431      | 0.413       | 0.415          | 0.410      | 0.415      | 0.400         | 0.414      | 0.418       | 0.424          | 0.407      | 0.418       |
> |           | 8 | 0.427         | 0.430      | 0.413       | 0.414          | 0.402      | 0.412      | 0.397          | 0.412      | 0.417       | 0.424          | 0.407      | 0.416      |
> |               | 16 | 0.424        | 0.427     | 0.411      | 0.412          | 0.402      | 0.412      | 0.399          | 0.414      | 0.414      | 0.422          | 0.406      | 0.417      |
> |               | 32 | 0.425          | 0.427      | 0.410      | 0.411          | 0.405      |0.414      | 0.398          | 0.414      | 0.415      | 0.422         | 0.410      | 0.419      |
> | **Exchange** | 2 | 0.401       | 0.445      | 0.327       | 0.401          | 0.365      | 0.405      | 0.361          | 0.401     | 0.370       | 0.408          | 0.364      | 0.405       |
> |               | 8 | 0.399          | 0.444      | 0.319      | 0.397          | 0.360      | 0.402  | 0.355          | 0.398      | 0.369       | 0.407          | 0.363      | 0.403      |
> |               | 16 | 0.403      | 0.448      | 0.321      | 0.399          | 0.361      | 0.403      | 0.357          | 0.400      | 0.364     | 0.405          | 0.362     | 0.404      |
> |               | 32 | 0.402          | 0.447     | 0.321      | 0.398          | 0.362      | 0.403    | 0.358          | 0.401      | 0.365      | 0.407          | 0.362      | 0.404      |
>
> *Note: `-n` indicates the removal of nonlinear activation functions. Reported results are the averages over four prediction horizons: 96, 192, 336, and 720.*
>
> >[W3-5] Other factors, such as TSMixer extending MLP-based architectures on time-series and PatchTST's designs to prevent overfitting, are not considered in Table 3. Consequently, the analyses in Section 4 are unconvincing regarding the necessity of introducing Mamba.
>
> **[RW3-5]**  $\underline{\text{Section 4}}$ focuses on the potential of vanilla Linear, Transformer, and Mamba architectures as backbones for time series forecasting. **Introducing too many new techniques would increase the complexity of the analysis.** In $\underline{\text{Section 4}}$, to evaluate semantic dependencies and verify the role of nonlinear activation functions, we introduced the Patching technique. **As answered in $\underline{\text{[W2-2]}}$, Patch+Transformer is PatchTST.** Our $\underline{\text{Section 4}}$ results show that Mamba is the only model among the three that can simultaneously capture both order dependency and semantic dependency without additional techniques, demonstrating its potential and superiority as a backbone for time series modeling.
>
> >[W4] Seemingly Downgraded Performance of Baselines
>
> **[RW4]**  In $\underline{\text{lines 432–433 of the paper}}$, we have already clarified in the introduction of baselines: “We carefully use 13 popular LTSF forecasting models as our baselines and we cite their performance from Liu et al. (2023) if applicable.” Therefore, **the reported results for PatchTST are based on the results provided in iTransformer.**
> According to the explanation in the iTransformer paper, this difference from:
>
> - **Enlarged lookback window**: The PatchTST paper adopts tunable lookback lengths (336, 512), while our method uniformly uses a length of 96, following the unified long-term forecasting protocol of TimesNet.
> - **More epochs to train**: The PatchTST paper trains the PatchTST model with 100 epochs, whereas we train all models with only 10 epochs.
> - **Learning rate**: The PatchTST paper adopts a carefully designed learning rate strategy.

---

> ### Author Response · Authors · 2024-11-21
> **Response to Reviewer qdrT [Part 4]**
>
> >[W5] Lack of Unique Contributions
>
> **[RW5]** We sincerely hope that the reviewers can reassess the contributions of this work. This paper provides four contributions that promote the development of the LTSF community along the lines of this research. The contributions of our work compared to previous studies are as follows:
>
> 1. **Identification and formal definition of three critical dependencies in time series data:**
>    We define these dependencies to guide the design of future LTSF models effectively.
>
> 2. **In-depth analysis of Mamba’s advantages:**
>    Compared to existing Mamba-based LTSF studies [3][4][9][10], we are the first to analyze the advantages of Mamba relative to Transformer and Linear models. We explain why Mamba is a promising backbone, which facilitates a reassessment of its potential and advantages in LTSF. At the same time, we explore the limitations of Transformer and Linear models in capturing order dependency and semantic dependency.
>
> 3. **Addressing overfitting due to non-linearities:**
>    We find that directly applying MLP, Transformer, and Mamba to LTSF can lead to overfitting issues caused by non-linearities. Removing non-linear activation functions yields performance improvements.
>
> 4. **Proposal of a disentangled encoding approach for cross-variable and cross-temporal dependencies:**
>    Unlike previous methods for introducing cross-variable dependencies that lacked theoretical explanation, we provide theoretical proof showing that our disentangled encoding approach is more effective. This lays a theoretically grounded path for future research in this area, encouraging further developments along this line.
>
> Overall, we sincerely thank the reviewer for the valuable suggestions mentioned above. Your feedback has significantly contributed to improving the quality of our manuscript. We hope that we have addressed your concerns and questions. We look forward to further discussions with you and hearing your new evaluations.
>
> - [1] Autoformer: Decomposition Transformers with Auto-Correlation for Long-Term Series Forecasting, NeurIPS 2021
> - [2] iTransformer: Inverted Transformers Are Effective for Time Series Forecasting, ICLR 2024
> - [3] Is Mamba Effective for Time Series Forecasting?, Arxiv
> - [4] TimeMachine: A Time Series is Worth 4 Mambas for Long-term Forecasting, Arxiv
> - [5] Are Transformers Effective for Time Series Forecasting?, AAAI 2023
> - [6] Are Language Models Actually Useful for Time Series Forecasting?, NeurIPS 2024
> - [7] Inductive Representation Learning on Large Graphs, NeurIPS 2017
> - [8] Graph Mamba: Towards Learning on Graphs with State Space Models, KDD 2024
> - [9] Bi-Mamba4TS: Bidirectional Mamba for Time Series Forecasting, Arxiv
> - [10] MambaTS: Improved Selective State Space Models for Long-term Time Series Forecasting , Arxiv

---

> ### Author Response · Authors · 2024-11-25
> **[**Gentle Reminder**]: Kind Request for Reviewers' Feedback**
>
> Dear Reviewer qdrT,
>
> Thank you once again for your valuable and constructive review, which has helped us refine our contribution and clarify its strengths.
>
> We would like to kindly remind you that the discussion deadline is approaching.  After this deadline, we may not have the opportunity to respond to your comments.
>
> Additionally, during the rebuttal period, we have supplemented our study with ***over 250 experimental results***  to enhance the comprehensiveness of our experiments and the reliability of our conclusions.  These results address your concerns regarding the experiments and are all included in $\underline{\text{the revised paper, highlighted in blue}}$.
>
> We sincerely appreciate your dedication and look forward to your feedback.
>
> Sincerely,
> ICLR 2025 Conference Submission 767 Authors

---

> > ### Comment · Reviewer_qdrT · 2024-11-25
> > **Thank you for your response**
> >
> > I appreciate the authors' considerable efforts in providing a thorough response.
> >
> > However, I do not believe my critical concerns have been adequately addressed, particularly regarding the necessity of adapting Mamba to time-series data.
> >
> > The primary demonstrated strength of Mamba lies in its efficiency when processing long contexts, rather than in improved expressiveness or learning capabilities (https://arxiv.org/abs/2406.07887). Consequently, I feel the authors may have overlooked the best opportunity for adapting Mamba to time-series applications.
> >
> > Regarding the claimed superiority of preserving order dependency (citing arguments based on DLinear, a paper published over two years ago), I do not see a clear necessity for preserving order dependency to achieve accurate forecasts. For instance, there have been significant advancements in improving time-series Transformers since then, such as reversible instance normalization (https://openreview.net/forum?id=cGDAkQo1C0p) and PatchTST (https://arxiv.org/abs/2211.14730), which substantially enhance vanilla Transformer performance in time-series forecasting without relying on order sensitivity. As such, I remain unconvinced by the designed experiments perturbing input order to justify whether a model is better suited for time-series forecasting. A simple recurrent neural network (e.g., GRU, LSTM) may show higher sensitivity to order perturbations, but this does not imply superior learning or generation capabilities compared to Transformers.
> >
> > Furthermore, while cross-variate dependency does not inherently involve order dependency, the design of a bidirectional Mamba raises additional concerns. How would the model handle other perturbed orders? Would an exponentially growing number of Mamba branches be required to encode various order permutations? I fail to see the necessity or rationale for using Mamba to model variate dependency effectively. In contrast, self-attention could be a perfect fit for modeling cross-variate depdency.
> >
> > In summary, I believe this work misses the key strength of Mamba, and its claimed properties, contributions, and novel designs appear unnecessary and unsubstantiated.

---

> ### Author Response · Authors · 2024-11-27
> **Thank you for your response [Part 1]**
>
> We appreciate the reviewer’s feedback. Before addressing the reviewer’s comments point by point, we would like to emphasize the main claim of this work: the advantage of Mamba lies in its ability to **simultaneously capture order dependency and semantic dependency.** The coexistence of these two capabilities enables Mamba's superiority in time series tasks. Below, we provide detailed responses to the reviewer’s comments:
>
> >[W1] The primary demonstrated strength of Mamba lies in its efficiency when processing long contexts, rather than in improved expressiveness or learning capabilities
>
> **[RW1]** The reviewer has provided an interesting finding regarding Mamba in the NLP domain. Similarly, we have demonstrated Mamba's advantage in handling long sequences through experiments with increasing lookback windows in the $\underline{\text{Appendix I of original paper}}$. We reiterate that there is currently a lack of research on Mamba's unique advantages in the time series domain, rather than merely transferring the already validated advantages of Mamba in NLP. Our study focuses on Mamba's distinctive strengths in time series, and our experimental results strongly demonstrate that Mamba can effectively capture both order dependency and semantic dependency. This finding highlights the need for the time series field to reassess the potential and advantages of Mamba.
>
> >[W2-1]  I do not see a clear necessity for preserving order dependency to achieve accurate forecasts. For instance, there have been significant advancements in improving time-series Transformers since then, such as reversible instance normalization (https://openreview.net/forum?id=cGDAkQo1C0p) and PatchTST (https://arxiv.org/abs/2211.14730), which substantially enhance vanilla Transformer performance in time-series forecasting without relying on order sensitivity.
>
> **[RW2-1]**  We reiterate that **cross-time dependency encompasses both order and semantic dependencies, and performance analysis should not isolate one from the other.** The reviewer mentioned the success of PatchTST, which highlights the importance of semantic dependency. Meanwhile, **the superior performance of many simpler MLP-based models like TimeMixer [1] underscores the significance of order dependency.** Mamba's ability to simultaneously capture both order and semantic dependencies showcases its potential as a backbone for time series tasks. This also explains why Patch+Mamba outperforms Patch+Transformer.
>
> >[W2-2] A simple recurrent neural network (e.g., GRU, LSTM) may show higher sensitivity to order perturbations, but this does not imply superior learning or generation capabilities compared to Transformers.
>
> **[RW2-2]** The weakness of RNNs in capturing semantic dependency accounts for their underperformance compared to Transformers. In summary, **the success of MLP-based models demonstrates the importance of order dependency, while the strong performance of Transformer-based models highlights the significance of semantic dependency.** The discovery that Mamba can capture both types of dependencies underscores its potential as a backbone for time series tasks.

---

> ### Author Response · Authors · 2024-11-28
> **Thank you for your response [Part 2]**
>
> >[W3-1] Furthermore, while cross-variate dependency does not inherently involve order dependency, the design of a bidirectional Mamba raises additional concerns. How would the model handle other perturbed orders? Would an exponentially growing number of Mamba branches be required to encode various order permutations?
>
> **[W3-1]** We thank the reviewer for pointing out this issue. In fact, it is unnecessary to adopt different Mamba encoding strategies for different orders, as bidirectional Mamba encoding is sufficient to handle various possible variate orders. We conducted experiments by reversing and randomly shuffling the variable order, and the **results demonstrate that bidirectional Mamba is robust to variate order.**
>
> | Dataset       | Predict Length | SAMBA  | MSE   | MAE   | SAMBA_s | MSE   | MAE   | SAMBA_b | MSE   | MAE   |
> |---------------|----------------|--------|-------|-------|---------|-------|-------|---------|-------|-------|
> | **ETTm1**     | 96             |        | 0.315 | 0.357 |         | 0.316 | 0.357 |         | 0.314 | 0.356 |
> |               | 192            |        | 0.360 | 0.383 |         | 0.362 | 0.383 |         | 0.361 | 0.383 |
> |               | 336            |        | 0.389 | 0.405 |         | 0.387 | 0.405 |         | 0.389 | 0.406 |
> |               | 720            |        | 0.448 | 0.440 |         | 0.448 | 0.441 |         | 0.445 | 0.439 |
> | **Traffic**   | 96             |        | 0.388 | 0.261 |         | 0.389 | 0.262 |         | 0.388 | 0.262 |
> |               | 192            |        | 0.411 | 0.271 |         | 0.410 | 0.271 |         | 0.409 | 0.270 |
> |               | 336            |        | 0.428 | 0.278 |         | 0.431 | 0.278 |         | 0.427 | 0.278 |
> |               | 720            |        | 0.461 | 0.297 |         | 0.461 | 0.299 |         | 0.462 | 0.297 |
>
> *Note: SAMBA_s is trained and tested on the dataset with shuffled variate order, while SAMBA_b is trained and tested on the dataset with reversed variate order.*
>
> >[W3-2]  I fail to see the necessity or rationale for using Mamba to model variate dependency effectively. In contrast, self-attention could be a perfect fit for modeling cross-variate depdency.
>
> **[W3-2]** Regarding whether Mamba or Transformer performs better, **we have already demonstrated in $\underline{\text{Appendix C of original paper}}$ that using Mamba alone to encode cross-variate dependency achieves superior performance and efficiency compared to Transformer.** This conclusion is further supported by the experiments in $\underline{\text{Appendix J of original paper}}$ and other studies on Mamba [2].
>
> - [1] TimeMixer: Decomposable Multiscale Mixing for Time Series Forecasting, ICLR 2024
> - [2] Is Mamba Effective for Time Series Forecasting?, Arxiv

---

### Meta-Review · Area_Chair_x68S · 2024-12-19

**Metareview:**

The reviewers raised multiple concerns, including the modeling, experiments and novelty. Although the author feedback has largely improved the manuscript, some concerns have not been resolved yet, which suggests a reject. The paper does not manage to clearly differentiate itself from existing research in the field and fails to fully address the concerns raised by multiple reviewers regarding fundamental aspects like the choice of Mamba and the true novelty of the proposed techniques.  The authors are suggested to include all the revisions and present a better future version of this paper.

**Additional Comments On Reviewer Discussion:**

Diverged reviews and some reviewers pointed out that their concerns were not resolved yet. The positive reviewers does not argue for accept.

---

### Decision · Program_Chairs · 2025-01-22

Reject